# Scaling laws for learning with real and surrogate data

**Ayush Jain[1]**
ayush.jain@granica.ai

**Andrea Montanari[1,2]**
andrea.montanari@granica.ai

**Eren Sasoglu[1]**
eren.sasoglu@granica.ai

[1] Granica Computing Inc. — granica.ai
[2] Stanford University

## Abstract

Collecting large quantities of high-quality data can be prohibitively expensive or impractical, and a bottleneck in machine learning. One may instead augment a small set of $n$ data points from the target distribution with data from more accessible sources, e.g. data collected under different circumstances or synthesized by generative models. We refer to such data as 'surrogate data'. We study a weighted empirical risk minimization (ERM) approach for integrating surrogate data into training. We analyze mathematically this method under several classical statistical models, and validate our findings empirically on datasets from different domains. Our main findings are: $(i)$ Integrating surrogate data can significantly reduce the test error on the original distribution. Surprisingly, this can happen *even when the surrogate data is unrelated to the original ones*. We trace back this behavior to the classical Stein's paradox. $(ii)$ In order to reap the benefit of surrogate data, it is crucial to use optimally weighted ERM. $(iii)$ The test error of models trained on mixtures of real and surrogate data is approximately described by a scaling law. This scaling law can be used to predict the optimal weighting scheme, and to choose the amount of surrogate data to add.

## 1 Introduction and overview

### 1.1 Motivation and formulation

Consider a standard learning setting where we are given $n$ i.i.d. points $z_i$ from a target distribution $\mathcal{D}$. Given a family of parametric models governed by the parameters' vector $\boldsymbol{\theta}$, the goal is to find $\boldsymbol{\theta}$ that minimizes the expected test loss $R_{\text{test}}(\boldsymbol{\theta})$ incurred by the model predictions, where expectation is taken over the distribution $\mathcal{D}$. In many application domains, the available data $\boldsymbol{Z} = (z_i)_{i \leq n}$ from the target distribution, referred to as either *real* or *original* data, may be difficult or expensive to acquire. One may then attempt to supplement these data with a different, cheaper source. Examples of such cheaper sources are $(i)$ publicly available datasets; $(ii)$ datasets owned by the same research group or company but acquired in different circumstances, e.g. in a different location; $(iii)$ synthetic data produced by a generative model.

We will denote the data points obtained from this source by $z_i^s$, and assume we have $m$ of them. We assume the 'surrogate' data $\boldsymbol{Z}^s = (z_i^s)_{i \leq m}$ to be i.i.d. samples with distribution $\mathcal{D}^s$. In general, we will not assume the distribution $\mathcal{D}^s$ of synthetic data to be close to the original data distribution $\mathcal{D}$. However we assume that these distributions are over the same domain. A number of questions arise: $(i)$ How should we use the surrogate data in training? $(ii)$ How many surrogate samples should we add to the original data? $(iii)$ Can we predict the improvement in test error achieved by adding surrogate samples to the training?

A natural approach would be to add the surrogate data to the original one in the usual training procedure, and indeed many authors have explored this approach (see Section 1.3). Namely, one attempts to minimize the overall empirical risk $\widehat{R}_{n+m}^{\text{naive}}(\boldsymbol{\theta}) = \sum_{i=1}^{n} \ell(\boldsymbol{\theta}; \boldsymbol{z}_i) + \sum_{i=1}^{m} \ell(\boldsymbol{\theta}; \boldsymbol{z}_i^s)$, where $\ell(z, \theta)$ is a train loss function.

However, a moment of reflection reveals that this approach has serious shortcomings. Consider a simple mean estimation problem, whereby $\boldsymbol{z}_i \sim \mathsf{N}(\boldsymbol{\theta}_*, \boldsymbol{I}_d)$, $\boldsymbol{z}_i^s \sim \mathsf{N}(\boldsymbol{\theta}_*^s, \boldsymbol{I}_d)$, $\ell(\boldsymbol{\theta}; \boldsymbol{z}) = \|\boldsymbol{\theta} - \boldsymbol{z}\|^2$, and $R_{\text{test}}(\boldsymbol{\theta}) = \|\boldsymbol{\theta} - \boldsymbol{\theta}_*\|^2$. A straightforward calculation yields that the test error of the empirical risk minimizer $\hat{\boldsymbol{\theta}}_{n+m}^{\text{naive}} := \arg\min \widehat{R}_{n+m}^{\text{naive}}(\boldsymbol{\theta})$ is

$$R_{\text{test}}(\hat{\boldsymbol{\theta}}_{n+m}^{\text{naive}}) = \left(\frac{m}{n+m}\right)^2 \|\boldsymbol{\theta}_*^s - \boldsymbol{\theta}_*\|^2 + \frac{d}{n+m} \ . \tag{1}$$

As $m$ increases the variance (the second term) decreases, but the bias due to the difference $\|\boldsymbol{\theta}_*^s - \boldsymbol{\theta}_*\|$ increases, and the error approaches $\|\boldsymbol{\theta}_*^s - \boldsymbol{\theta}_*\|^2$, i.e. the model will be only as good as if training only on surrogate data.

In order to overcome these limitations, we study a weighted ERM approach, and will show that the weight plays a crucial role. Namely, we consider the following regularized empirical risk:

$$\widehat{R}_{n,m}(\boldsymbol{\theta}; \alpha) := \frac{1-\alpha}{n} \sum_{i=1}^{n} \ell(\boldsymbol{\theta}; \boldsymbol{z}_i) + \frac{\alpha}{m} \sum_{i=1}^{m} \ell(\boldsymbol{\theta}; \boldsymbol{z}_i^s) + \Omega(\boldsymbol{\theta}) \ , \tag{2}$$

where $\alpha \in [0, 1]$ is the weight of the surrogate dataset and $\Omega : \mathbb{R}^d \to \mathbb{R}_{\geq 0}$ is a regularizer, e.g. a ridge $\Omega(\boldsymbol{\theta}) = \lambda \|\boldsymbol{\theta}\|_2^2$. We denote by

$$\hat{\boldsymbol{\theta}}_{n,m}(\alpha) := \arg\min_{\boldsymbol{\theta}} \widehat{R}_{n,m}(\boldsymbol{\theta}; \alpha) \tag{3}$$

the corresponding empirical risk minimizer, and by $R_{\text{test}}(\hat{\boldsymbol{\theta}}_{n,m}(\alpha))$ the corresponding test error.

For supervised learning tasks, a sample $\boldsymbol{z}$ is represented as $\boldsymbol{z} = (y, \boldsymbol{x})$, where $\boldsymbol{x} \in \mathbb{R}^d$ is covariate vector and $y \in \mathbb{R}$ is response variable and $\boldsymbol{\theta}$ parametrizes a family of models $f(\boldsymbol{x}; \boldsymbol{\theta})$ that predict the response $y$ given covariate vector $\boldsymbol{x}$. We consider losses of the form $\ell(\boldsymbol{\theta}, \boldsymbol{z}) = L(y, f(\boldsymbol{x}; \boldsymbol{\theta}))$ and $R_{\text{test}}(\boldsymbol{\theta}) := \mathbb{E}_{z \sim \mathcal{D}} L_{\text{test}}(y, f(\boldsymbol{x}; \boldsymbol{\theta}))$ for some functions $L$ and $L_{\text{test}}$. We allow for the test loss $L_{\text{test}}$ to be different from the train loss $L$, but we will omit the subscript 'test' from the risk $R$ and the loss $L$ whenever clear from the context.

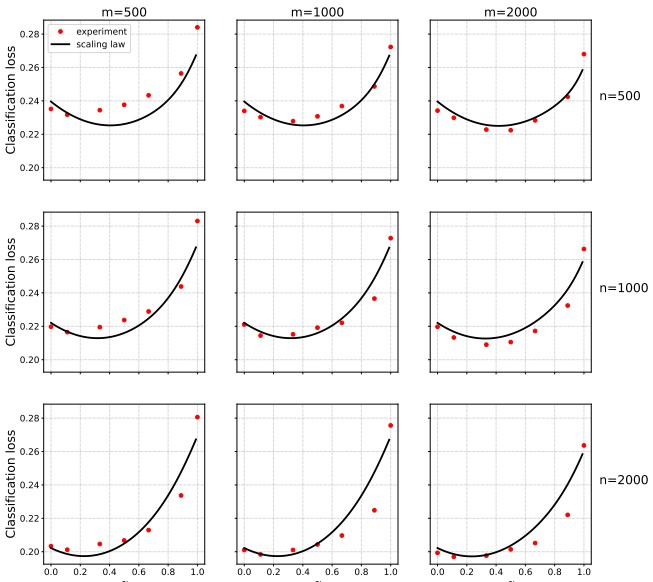

Figure 1: IMDB and Rotten Tomatoes data and neural networks. Test error when trained on mixtures of original and surrogate data. Black curves: prediction from Eq. (4).

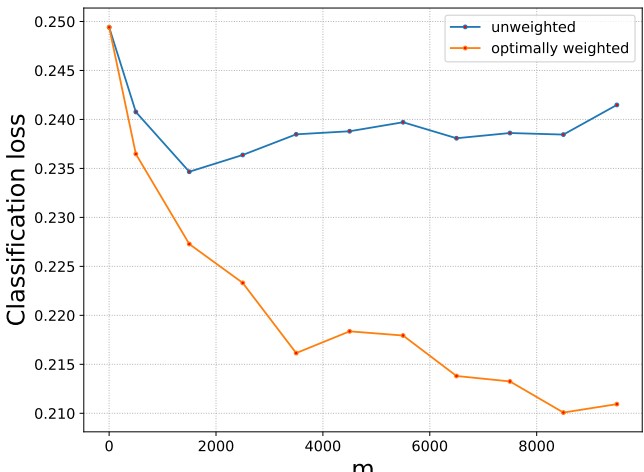

Figure 2: Performance of unweighted vs weighted ERM approach for the setting in Figure 1

Figure 1 provides a preview of our results, for a sentiment analysis task. (Technical details provided in Section 4 and Appendix A.3). Each frame corresponds to a different combination of $n$ and $m$, and we report the test error of our approach as a function of the weight parameter $\alpha$ (red circles). Solid lines report the prediction of a scaling law that will be one of the main results presented below.

We observe that the weighted ERM approach systematically achieves better test error than either training only on original data ($\alpha \to 0$) or on surrogate data ($\alpha \to 1$). Further the error for optimal $\alpha$ is always monotone decreasing both in $m$ and $n$, and the approach outperforms the naive unweighted approach. This is shown more clearly in Figure 2, which also shows that the performance of unweighted ERM can degrade with more surrogate data. Also, while scaling laws typically do not capture the dependence on hyperparameters, the scaling law presented below predicts the dependence on $\alpha$ reasonably well. This is particularly useful, because such a scaling law can be used to tune $\alpha$ optimally and to predict the amount of surrogate data needed.

## 1.2 Summary of results

We study the method outlined above both mathematically and via numerical experiments. Our mathematical results are developed in four different settings: $(i)$ The Gaussian sequence model (Section 3.1); $(ii)$ A non-parametric function estimation setting (Section 3.2); $(iii)$ Low-dimensional empirical-risk minimization (Section 3.3); $(iv)$ High dimensional ridge regression (Section 3.4);

We carry out experiments with the following data sources. (1) Simulated data from linear or Gaussian mixture models: this allows us to explicitly control the distribution shift between the original and surrogate datasets, as well as check our theoretical results in a controlled setting. (2) Real natural language processing (NLP) data for sentiment analysis, with the role of original dataset played by IMDB reviews and the role of surrogate datasets played respectively by Rotten Tomatoes review and Goodreads book reviews. (3) Progression-free survival analysis using Lasso on TCGA PanCancer dataset with female patients data and male patients data as original and surrogate data, respectively. (4) Real image classification data, with CIFAR-10 and CIFAR-100 datasets respectively playing the role of original and surrogate data. Our results support the following conclusions:

**Surrogate data improve test error.** Including surrogate data in training generally improves the test error on the original data, *even if the surrogate data distribution is far from the original one*. In agreement with the interpretation of surrogate data as a regularizer (see also Sec. 2), the improvement is generally positive, although its size depend on the data distributions.

**Tuning of $\alpha$.** The above conclusion holds under the condition that $\alpha$ can be tuned (nearly) optimally. For each of the theoretical settings already mentioned, we characterize this optimal value. We verify

that nearly optimal $\alpha$ can be effectively selected by minimizing the error on a validation split of the original data. An attractive alternative is to use the scaling law we discuss next.

**Scaling law.** We propose a scaling law that captures the behavior of the test error with $n, m, \alpha$:

$$R(\hat{\boldsymbol{\theta}}_{n,m}(\alpha)) - R_* \approx \alpha^2 R_{\mathsf{su}}^{\mathsf{ex}}(\infty) + \left[\alpha^2\left(R_{\mathsf{su}}^{\mathsf{ex}}(m) - R_{\mathsf{su}}^{\mathsf{ex}}(\infty)\right)^{1/\beta} + (1-\alpha)^2 R_{\mathsf{or}}^{\mathsf{ex}}(n)^{1/\beta}\right]^{\beta}. \quad (4)$$

Here $R_*$ is the minimal (Bayes) error, $R_{\mathsf{su}}^{\mathsf{ex}}(m) := R(\hat{\boldsymbol{\theta}}_{0,m}(1)) - R_*$ is the excess test error when training on the surrogate data (and testing on original), $R_{\mathsf{or}}^{\mathsf{ex}}(n) := R(\hat{\boldsymbol{\theta}}_{n,0}(0)) - R_*$ is the excess test error[1] when training on original data (and testing on original), and $\beta$ is a scaling exponent as described in Section 4. The above scaling admits natural generalizations; see Section 5.

**Practical uses of the scaling law.** Given data $\{z_i\}_{i \leq n}$ and a source of surrogate data, we would like to predict how much the test error can be decreased by including any number $m$ of surrogate samples in the mix. The scaling law (4) suggests a simple approach: (1) Learn models on purely original data to extract the behavior of test loss $R(\hat{\boldsymbol{\theta}}_{n,0}(0))$.; (2) Learn models on purely surrogate data to extract the behavior of $R(\hat{\boldsymbol{\theta}}_{0,m}(1))$. (A relatively small sample is sufficient for this step.) (3) Use the minimum over $\alpha$ of Eq. (4) to predict the test error at any given pair $n, m$.

We can further leverage the scaling law to achieve the desired error by: $(I)$ Using the scaling law to determine the number of surrogate samples needed to achieve the desired performance. $(II)$ Acquiring the surrogate samples and train the model using weighted ERM with optimal weighting predicted by scaling law.

### 1.3 Related work

The use of surrogate data to enhance training has attracted increasing research effort, also because of the recent progresses in generative modeling.

This line of work has largely focused on the techniques to generate synthetic data that are well suited for training. A wide variety of methods have been demonstrated to be useful in generating data for computer vision tasks, ranging from object classification to semantic segmentation [RSM+16, JRBM+17, AAMM+18, TPA+18, CLCG19, HSY+22, MPT+22, YCFB+22]. We refer to [SLW20] for a review. More recently, synthetic data have been used for training in natural language processing [HNK+22, MHZH22].

Scaling laws have been broadly successful in guiding the development of large machine learning models [HNA+17, RRBS19, HKK+20, KMH+20, TDR+21, HKHM21, HBM+22, ANZ22, MRB+23]. We expect them to be similarly useful for integrating heterogeneous data into training. The change in scaling laws when training on synthetic data was the subject of a recent empirical study [FCK+23]. On the other hand, no systematic attempt was made at integrating real and synthetic data.

In data augmentation [KSH12, SK19], the original samples are supplemented with transformed or noisy version of the same. In contrast, we assume that surrogate data is obtained from a different source than the original one, and the surrogate samples are independent of the original samples.

The problem we consider was also studied within 'domain adaptation', a subarea of transfer learning [MPRP16, TJJ20]. Among others, [BDBC+10] establishes bounds on the generalization error of weighted ERM via uniform convergence. However these bounds do not reveal the full advantage achieved by this approach and are not precise enough to justify the scaling laws that we derive. Recent works in domain adaptation study the behavior of test error error [Has21, KJSJ24, YLS+24] and its scaling laws [Has21], but only consider vanilla ERM, a special of weighted ERM considered here.

## 2 Regularization, Gaussian mean estimation, Stein paradox

The role of the parameter $\alpha$ can be understood by considering the limit $m \to \infty$:

$$\widehat{R}_{n,\infty}(\boldsymbol{\theta}; \alpha) = \frac{1-\alpha}{n} \sum_{i=1}^{n} \ell(\boldsymbol{\theta}; z_i) + \alpha R^s(\boldsymbol{\theta}) + \Omega(\boldsymbol{\theta}),$$

---

[1]We assume here that $\lim_{n \to \infty} R(\hat{\boldsymbol{\theta}}_{n,0}(0)) = R_*$, i.e. that we achieve Bayes risk with infinitely many original samples. See Section 5.

and $R^s(\boldsymbol{\theta}) = \mathbb{E}_{\boldsymbol{z}^s \sim \mathcal{D}^s} \ell(\boldsymbol{\theta}; \boldsymbol{z}^s)$ is the population risk for surrogate data. This suggests to think of the surrogate data as an additional (highly non-trivial) regularizer, with parameter $\alpha$. This leads to a simple yet important insight: adding surrogate data to the original data is beneficial if $\alpha$ is chosen optimally, and large $m$ reduces statistical fluctuations in this regularizer. This contrasts with the unweighted approach whose test error in general deteriorates for large $m$.

As a toy example, reconsider the mean estimation problem mentioned in the introduction: $\boldsymbol{z}_i \sim \mathsf{N}(\boldsymbol{\theta}_*, \boldsymbol{I}_d)$ and $\boldsymbol{z}_i^s \sim \mathsf{N}(\boldsymbol{\theta}_*^s, \boldsymbol{I}_d)$, $\ell(\boldsymbol{\theta}; \boldsymbol{z}) = \|\boldsymbol{\theta} - \boldsymbol{z}\|^2$ and $R_{\text{test}}(\boldsymbol{\theta}) = \|\boldsymbol{\theta} - \boldsymbol{\theta}_*\|^2$. We have $\hat{\boldsymbol{\theta}}_{n,m}(\alpha) = (1-\alpha)\sum_{i\leq n} \boldsymbol{z}_i/n + \alpha \sum_{i\leq m} \boldsymbol{z}_i^s/m$. In other words, the weighted ERM shrinks the mean of the original data towards the mean of the surrogate data. For a given $\alpha$, the resulting test errors are

$$R(\hat{\boldsymbol{\theta}}_{n,m}(\alpha)) = \alpha^2 R_{\text{su}}^{\text{ex}}(\infty) + \left(\frac{\alpha^2}{m} + \frac{(1-\alpha)^2}{n}\right) d, \quad R_{\text{su}}^{\text{ex}}(\infty) = \|\boldsymbol{\theta}_* - \boldsymbol{\theta}_*^s\|^2, \tag{5}$$

and for the optimum value $\alpha_* = \arg\min_\alpha R(\hat{\boldsymbol{\theta}}_{n,m}(\alpha))$, this yields

$$R(\hat{\boldsymbol{\theta}}_{n,m}(\alpha_*)) = \left(\frac{R_{\text{su}}^{\text{ex}}(\infty) + d/m}{R_{\text{su}}^{\text{ex}}(\infty) + d/m + d/n}\right) \cdot \frac{d}{n}. \tag{6}$$

Note that $1/n$ is the error of training only on original data and the prefactor is always strictly smaller than one. Hence, weighted ERM always achieves better error than training only on original data, *regardless of the distance between original and surrogate data*, although the improvement is larger for small $R_{\text{su}}^{\text{ex}}(\infty)$. This might seem paradoxical at first. As mentioned above, we are shrinking towards an arbitrary point given by the empirical mean of the surrogate data: how can this help?

In fact, this is a disguised version of the celebrated Stein paradox [EM77, Ste81]: in estimating a Gaussian mean, a procedure that shrinks the empirical mean *towards an arbitrary point* by a carefully chosen amount outperforms the naive empirical mean. In our toy example, the naive empirical mean corresponds to estimation purely based on the original data, and we shrink it towards the mean of the surrogate data. Of course, the improvement over empirical mean is only possible if $\alpha$ is chosen optimally. Equation (6) assumes $\alpha = \alpha_*$ is chosen by an oracle that knows the value of $R_{\text{su}}^{\text{ex}}(\infty)$. Stein's analysis implies that in the Gaussian mean problem, $\alpha$ can be chosen empirically as long as the dimension of $\boldsymbol{\theta}$ is $d \geq 3$. In the settings we are interested in, $\alpha$ can be chosen via cross-validation.

## 3 Theoretical results

### 3.1 Gaussian sequence model

The sequence model captures the behavior of many models in non-parametric statistics while being simpler to analyze [Tsy09, GN21]. It is also known to approximate the behavior of overparametrized linear regression [CM22]. The unknown target is $\boldsymbol{\theta}_* \in \mathbb{R}^d$ (with potentially $d = \infty$), and we observe

$$\boldsymbol{y}_i = \boldsymbol{\theta}_* + \sigma \boldsymbol{g}_i, \; i \leq n, \quad \boldsymbol{y}_i^s = \boldsymbol{\theta}_*^s + \sigma_s \boldsymbol{g}_i^s, \; i \leq m, \tag{7}$$

where $\boldsymbol{\theta}_*^s$ is also unknown, and $\boldsymbol{g}_i, \boldsymbol{g}_i^s \sim \mathsf{N}(0, \boldsymbol{I}_d)$ are i.i.d. We study the penalized estimator

$$\hat{\boldsymbol{\theta}}_{n,m}(\alpha) := \arg\min_{\boldsymbol{\theta}} \left\{ \frac{(1-\alpha)}{n} \sum_{i=1}^n \|\boldsymbol{y}_i - \boldsymbol{\theta}\|_2^2 + \frac{\alpha}{m} \sum_{i=1}^m \|\boldsymbol{y}_i^s - \boldsymbol{\theta}\|_2^2 + \lambda \|\boldsymbol{\theta}\|_{\boldsymbol{\Omega}}^2 \right\}, \tag{8}$$

where $\|\boldsymbol{\theta}\|_{\boldsymbol{\Omega}}^2 = \langle \boldsymbol{\theta}, \boldsymbol{\Omega}\boldsymbol{\theta} \rangle$ and $\boldsymbol{\Omega} \succeq 0$ is a regularization weight matrix. We will be concerned with the expected risk

$$R_{n,m}(\alpha, \lambda) = \mathbb{E}\left\{ \|\hat{\boldsymbol{\theta}}_{n,m}(\alpha) - \boldsymbol{\theta}_*\|^2 \right\}. \tag{9}$$

The proof of the next result is presented in Appendix C.

**Theorem 1.** *Let $\omega_1 \leq \omega_2 \leq \cdots$ be the ordered eigenvalues of $\boldsymbol{\Omega}$, and denote by $\boldsymbol{v}_i$ the corresponding eigenvectors. Further denote by $\boldsymbol{\theta}_{*,>k}$, $\boldsymbol{\theta}_{*,>k}^s$ the projections of $\boldsymbol{\theta}_*$, $\boldsymbol{\theta}_{*,s}$ onto $\text{span}(\boldsymbol{v}_i : i > k)$, and similarly for $\boldsymbol{\theta}_{*,\leq k}$, $\boldsymbol{\theta}_{*,\leq k}^s$. Assume that $\omega_k \asymp k^\mu$, $\mu > 1/2$, $\|\boldsymbol{\theta}_{*,>k}\|^2 \leq C_\theta k^{-2\rho}$, $\rho \neq \mu$, and let $\Delta_k$ be such that (for all $k$): $\Delta_k := \omega_k^{-1} |\langle \boldsymbol{\theta}_{*,\leq k} - \boldsymbol{\theta}_{*,\leq k}^s, \boldsymbol{\theta}_{*,\leq k}\rangle_{\boldsymbol{\Omega}}| \leq C_0 k^{-2(\mu \wedge \rho)}$. Then the following hold:*

    (a) *There exists an explicit $\lambda_*(\alpha)$ such that, letting $\beta := 2(\mu \wedge \rho)/(1 + 2(\mu \wedge \rho))$,*

$$R_{n,m}(\alpha, \lambda_*(\alpha)) \leq \alpha^2 R_{\text{su}}^{\text{ex}}(\infty) + C \cdot \left[(1-\alpha)^2 \frac{\sigma^2}{n} + \alpha^2 \frac{\sigma_s^2}{m}\right]^\beta. \tag{10}$$

(b) *If $\mu > 2\rho - 1/2$, there exists $C' > 0$ and there exist $\boldsymbol{\theta}_*, \boldsymbol{\theta}_*^s$ satisfying the assumptions in point $(a)$, such that,*

$$\min_\lambda R_{n,m}(\alpha, \lambda) \geq \alpha^2 R_{\mathsf{su}}^{\mathsf{ex}}(\infty) + C' \cdot \left[(1-\alpha)^2 \frac{\sigma^2}{n} + \alpha^2 \frac{\sigma_s^2}{m}\right]^\beta . \tag{11}$$

Note that since the theorem also implies $R_{\mathsf{su}}^{\mathsf{ex}}(m) - R_{\mathsf{su}}^{\mathsf{ex}}(\infty) \asymp (\sigma_s^2/m)^\beta$ and $R_{\mathsf{or}}^{\mathsf{ex}}(m) \asymp (\sigma^2/n)^\beta$, this result confirms the scaling law (4).

## 3.2 Non-parametric regression in Sobolev classes

In this section we consider the classic non-parametric regression model. We assume that $n = Q^d$ for some integer $Q \geq 2$, and the original data $(\boldsymbol{x}_i, y_i)_{i \leq n}$ are defined through

$$y_i = f_*(\boldsymbol{x}_i) + \varepsilon_i , \quad \varepsilon_i \sim \mathsf{N}(0, \sigma^2) , \tag{12}$$

where $\varepsilon_i$ are independent of $\boldsymbol{x}_i$ and of each other, and $\{\boldsymbol{x}_i\}_{i \leq n}$ equally spaced grid points in the $d$-dimensional unit-cube, i.e. $\mathcal{X}_n = \{\boldsymbol{q}/Q : \boldsymbol{q} \in [Q]^d\}$. Surrogate data have a similar distribution, with $m = Q_s^d$ equally spaced points $\boldsymbol{x}_i^s$ in the unit cube, and $y_i^s = f_{*,s}(\boldsymbol{x}_i^s) + \varepsilon_i^s$, where $\varepsilon_i^s \sim \mathsf{N}(0, \sigma_s^2)$. We assume that $f_*$ has small Sobolev norm, that is,

$$\|f_*\|_{r,2}^2 := \int_{[0,1]^d} \left(|f_*(t)|^2 + \|f_*^{(r)}(t)\|^2\right) \mathrm{d}t \leq 1 .$$

Recall that $\|f\|_{r,2}^2$ is a special reproducing kernel Hilbert space (RKHS) norm: we expect some of the considerations below to generalize to other RKHS norms.

Following our general methodology, we use the estimator

$$\hat{f}_{n,m,\alpha} = \arg\min_f \left\{ \frac{1-\alpha}{n} \sum_{i=1}^n \left(y_i - f(\boldsymbol{x}_i)\right)^2 + \frac{\alpha}{m} \sum_{i=1}^m \left(y_i^s - f(\boldsymbol{x}_i^s)\right)^2 + \lambda\|f\|_{p,2}^2 \right\}. \tag{13}$$

We are interested in $R(f) = \mathbb{E}\{(f(\boldsymbol{x}) - f_*(\boldsymbol{x}))^2\}$, which is the excess squared loss for a test point $\boldsymbol{x} \sim \mathsf{Unif}([0,1]^d)$.

In order to avoid technical burden we will carry out the analysis for a continuous model, the so-called white noise model, where we observe the function $f$ at all points $\boldsymbol{x} \in [0,1]^d$, perturbed by $d$-dimensional white noise:

$$\mathrm{d}Y = f_*(\boldsymbol{x}) \, \mathrm{d}\boldsymbol{x} + \frac{\sigma}{\sqrt{n}} \mathrm{d}B(\boldsymbol{x}) , \tag{14}$$

and similarly for $Y^s$. We use an estimator that naturally generalizes (13) to the continuous case. Our results for the white noise model are as follows.

**Theorem 2.** *Let $\beta = (2p \wedge 4r)/(d + (2p \wedge 4r))$. If $r > d/4$ and $\lambda = (\delta K_{n,m}\sigma^2)^{2r/(d+(2p\wedge 4r))}$, then for every $\delta \in (0,1)$ there exists a constant $C = C(d, \delta)$ such that*

$$R(\hat{f}_{n,m,\alpha}) \leq (1+\delta)\alpha^2 R_{\mathsf{su}}^{\mathsf{ex}}(\infty) + C\left\{(1-\alpha)^2 \cdot \frac{\sigma^2}{n} + \alpha^2 \cdot \frac{\sigma_s^2}{m}\right\}^\beta \tag{15}$$

*with high probability, where $K_{n,m} := (1-\alpha)^2/n + \alpha^2/m$.*

**Remark 3.1.** The white noise model (14) is known to be equivalent to the original model (12) (with deterministic equispaced designs) in the sense of Le Cam, for $r > d/2$ [BL96, Rei08]. While suggestive, this equivalence does not allow us to formally deduce results for the data (12), because it does not apply to the specific estimators of interest here.

With the given choice of $\lambda, r$, the derivation of (15) also implies $R_{\mathsf{su}}^{\mathsf{ex}}(m) - R_{\mathsf{su}}^{\mathsf{ex}}(\infty) \geq C'(\sigma_s^2/m)^\beta$, $R_{\mathsf{or}}^{\mathsf{ex}}(n) \geq C'(\sigma/n)^\beta$ (for the least favorable $f$ [Tsy09]). Hence (15) is consistent with the scaling law (4).

## 3.3 Low-dimensional asymptotics

We study the estimator of Eqs. (2), (3) under the classical asymptotics $n, m \to \infty$ at $d$ fixed. Since this type of analysis is more standard, we defer it to Appendix B. The main result of this analysis is that the scaling law (4) holds in this setting, with the classical parametric exponent $\beta = 1$, for $\alpha \in [0, \alpha_{\max}]$ for a suitable $\alpha_{\max} \in (0, 1)$. Importantly, the interval $[0, \alpha_{\max}]$ includes the optimal choice of the weight $\alpha$.

## 3.4 High-dimensional linear regression

In this section, we study ridge regression in the high-dimensional regime in which the number of samples is proportional to the number of parameters. Denoting the original data by $(\boldsymbol{y}, \boldsymbol{X})$ (with $\boldsymbol{y} \in \mathbb{R}^n$ the vector of responses and $\boldsymbol{X} \in \mathbb{R}^{n \times d}$ the matrix of covariates), and the surrogate data by $(\boldsymbol{y}^s, \boldsymbol{X}^s)$ (with $\boldsymbol{y}^s \in \mathbb{R}^m$ and $\boldsymbol{X}^s \in \mathbb{R}^{m \times d}$), we minimize the regularized empirical risk

$$\widehat{R}_{n,m}(\boldsymbol{\theta}; \alpha) = \frac{1-\alpha}{2n} \|\boldsymbol{y} - \boldsymbol{X}\boldsymbol{\theta}\|_2^2 + \frac{\alpha}{2m} \|\boldsymbol{y}^s - \boldsymbol{X}^s\boldsymbol{\theta}\|_2^2 + \frac{\lambda}{2} \|\boldsymbol{\theta}\|_2^2 , \qquad (16)$$

We assume a simple distribution, whereby the rows of $\boldsymbol{X}$, $\boldsymbol{X}^s$ (denoted by $\boldsymbol{x}_i$, $\boldsymbol{x}_i^s$) are standard normal vectors and

$$\boldsymbol{y} = \boldsymbol{X}\boldsymbol{\theta}_* + \boldsymbol{\varepsilon} , \qquad \boldsymbol{y}^s = \boldsymbol{X}^s\boldsymbol{\theta}_*^s + \boldsymbol{\varepsilon}^s . \qquad (17)$$

for $\boldsymbol{\varepsilon} \sim \mathsf{N}(\boldsymbol{0}, \sigma^2 \boldsymbol{I}_n)$, $\boldsymbol{\varepsilon}^s \sim \mathsf{N}(\boldsymbol{0}, \sigma_s^2 \boldsymbol{I}_m)$. Note that the two data distributions differ in the true coefficient vectors $\boldsymbol{\theta}_*$ versus $\boldsymbol{\theta}_*^s$ as well as in the noise variance. We will denote by $\hat{\boldsymbol{\theta}}_{n,m}(\alpha)$ the ridge estimator, $\hat{\boldsymbol{\theta}}_{n,m}(\alpha) = \arg\min_{\boldsymbol{\theta} \in \mathbb{R}^d} \widehat{R}_{n,m}(\boldsymbol{\theta}; \alpha)$.

The excess test error (for square loss) is given by $R(\hat{\boldsymbol{\theta}}) := \mathbb{E}\{(\langle \boldsymbol{x}, \boldsymbol{\theta}_* \rangle - \langle \boldsymbol{x}, \hat{\boldsymbol{\theta}} \rangle)^2\} = \|\hat{\boldsymbol{\theta}} - \boldsymbol{\theta}_*\|^2$. The next result characterizes this error in the proportional asymptotics.

**Theorem 3.** *Consider the ridge regression estimator $\hat{\boldsymbol{\theta}}_{n,m}(\alpha)$. Let $r := \|\boldsymbol{\theta}_*\|_2$, $r_s := \|\boldsymbol{\theta}_*^s\|_2$ and $\gamma := \cos^{-1}(\langle \boldsymbol{\theta}_*, \boldsymbol{\theta}_*^s \rangle / (\|\boldsymbol{\theta}_*\|_2 \|\boldsymbol{\theta}_*^s\|_2))$. Assume $n, m, d \to \infty$ such that $n/d \to \delta$, $m/d \to \delta_s$, with $\delta + \delta_s > 1$[2]. For $\mathscr{R}(.)$ defined in Appendix E.1, let*

$$\xi^*(\alpha), \xi_\perp^*(\alpha), \omega^*(\alpha) = \operatorname{argmin}_{\xi, \xi_\perp \geq 0, \omega \geq 0} \mathscr{R}(\xi, \xi_\perp, \omega; \alpha),$$

*be the unique minimizer. Then for any $\varepsilon, \varepsilon_0 > 0$, there exist $c > 0$ such that, for all $n$*

$$\mathbb{P}\left( \sup_{\alpha \in [\varepsilon_0, 1-\varepsilon_0]} |R(\hat{\boldsymbol{\theta}}_{n,m}(\alpha)) - \mathscr{R}_{\text{test}}(\alpha)| \leq \varepsilon \right) \geq 1 - 2\, e^{-cn} ,$$

*where $\mathscr{R}_{\text{test}}(\alpha) := (\xi^*(\alpha) - r)^2 + (\xi_\perp^*(\alpha))^2 + (\omega^*(\alpha))^2$. Further, we can take $\varepsilon_0 = 0$ if $\delta, \delta_s > 1$.*

**Remark 3.2** (Optimizing $\alpha$ over the validation set). Note that the concentration of $R(\hat{\boldsymbol{\theta}}_{n,m}(\alpha))$ around the theoretical prediction $\mathscr{R}_{\text{test}}(\alpha)$ in Theorem 3 is uniform over $\alpha \in [\varepsilon_0, 1 - \varepsilon_0]$. This means that we can find the optimal $\alpha$ by computing $\hat{\boldsymbol{\theta}}_{n,m}(\alpha)$ over a grid of $\alpha$ values, estimating $R(\hat{\boldsymbol{\theta}}_{n,m}(\alpha))$ over the validation set and choosing the optimal $\alpha$. The uniform guarantee insures that this procedure will achieve risk $\min_{\alpha \in [0,1]} \mathscr{R}_{\text{test}}(\alpha) + o_P(1)$.

**Remark 3.3** (Relation to scaling laws). An analysis of the equations for $(\xi^*, \xi_\perp^*, \omega^*)$ reveals that, for large $\delta, \delta_s$, the predicted excess risk behaves as $\mathscr{R}_{\text{test}}(\alpha) = \alpha^2 \mathscr{R}_{s,\infty}^* + \alpha^2 C_1/\delta_s + (1-\alpha)^2 C_2/\delta + o(1/\delta, 1/\delta_s)$ (for some constants $\mathscr{R}_{s,\infty}^*, C_1, C_2$). This matches the low-dimensional asymptotics and our scaling law (4) with $\beta = 1$. In practice, we find that, for moderate $\delta, \delta_s$, the behavior of $\mathscr{R}_{\text{test}}(\alpha)$ is better approximated by a different value of $\beta$ (see Appendix A.)

## 4 Empirical results

In this section, we present experiments validating that the scaling law (4) is a good approximation both for simulated and real-world data. For simulated data, we select two different distributions for the original and surrogate datasets. The test and validation sets are generated from the same distribution as the original dataset. In case of real-world data, we choose two different datasets as the original and surrogate datasets. We split the original dataset into train, test, and validation sets, while all examples in the surrogate datasets are allocated solely to the train split.

For each dataset and model discussed in this section, we carry out the same experiment: $(i)$ We use models trained on original data to fit the scaling curve $R(\hat{\boldsymbol{\theta}}_{n,0}(0)) = A_{\text{or}} + B_{\text{or}} n^{-\beta_{\text{or}}}$ and obtain $A_{\text{or}}$ and $\beta_{\text{or}}$ $(ii)$ We use models trained on purely surrogate data to fit the scaling curve $R(\hat{\boldsymbol{\theta}}_{0,m}(1)) = A_{\text{su}} + B_{\text{su}} m^{-\beta_{\text{su}}}$ to obtain $A_{\text{su}}$ and $\beta_{\text{su}}$. $(iii)$ Since assume $R_* = R(\hat{\boldsymbol{\theta}}_{\infty,0}(0))$, we let

---

[2]The same proof, with some additional technical work, yields a characterization for $\delta + \delta_s \leq 1$ as well. We omit it here for brevity.

$R_* = A_{\text{or}}$ and excess risk estimates $R_{\text{or}}^{\text{ex}}(n) = R(\hat{\boldsymbol{\theta}}_{n,0}(0)) - A_{\text{or}}$, $R_{\text{su}}^{\text{ex}}(m) = R(\hat{\boldsymbol{\theta}}_{0,m}(1)) - A_{\text{or}}$ and $R_{\text{su}}^{\text{ex}}(\infty) = A_{\text{su}} - A_{\text{or}}$, and we use $\beta = \beta_{\text{or}}$, the fit exponent obtained from original data); $(iv)$ For each combination of $n, m$, we use our estimates of $R_{\text{su}}^{\text{ex}}(m)$, $R_{\text{or}}^{\text{ex}}(n)$ (as measured empirically on the test set), $\beta$, $R_{\text{su}}^{\text{ex}}(\infty)$, and $R_*$ to plot the predicted $R(\hat{\boldsymbol{\theta}}_{n,m}(\alpha))$ as a function of $\alpha$ using scaling law (4). $(v)$ We then train the model using $n$ original and $m$ surrogate examples with weights $(1 - \alpha)$ and, $\alpha$ for the two datasets, respectively. We average the results of 10 independent runs to compare it against those predicted by the scaling law. For ridge regression, we also compare with exact high-dimensional asymptotics from Theorem 3.

*Let us emphasize that these plots probe the dependence on the hyperparameter $\alpha$. These are much more demanding tests that the usual ones in scaling laws. We generally observe that the scaling law captures well the behavior of the test error for data mixtures. Furthermore, we perform experiments for variety of loss functions to show these scaling laws hold more widely than the theoretical settings we considered.*

**Binary classification with Gaussian mixture data**   This is a simple simulated setting. The original dataset consists of independent and identically distributed examples $(y_i, \boldsymbol{x}_i) \in \mathbb{R} \times \mathbb{R}^d$, $d = 200$, where $y_i$ is uniform over $\{+1, -1\}$, and $\boldsymbol{x}_i\big|_{y_i} \sim \mathsf{N}(y_i\boldsymbol{\theta}_*, \boldsymbol{I}_d)$, where $\boldsymbol{\theta}_* \in \mathbb{R}^d$, $\|\boldsymbol{\theta}_*\| = 1$. Surrogate data have the same distribution, with a different unit vector $\boldsymbol{\theta}_{*,s}$. This data distribution is parametrized by $d$ and the angle $\gamma$ between the original and surrogate parameters, $\cos\gamma := \langle \boldsymbol{\theta}_*, \hat{\boldsymbol{\theta}}_{*,s} \rangle$. We use $\gamma = \pi/10$ in our experiments. For each $(n, m, \alpha)$, we averaged the results over 10 independent runs.

We use two different models for classification: (1) Logistic regression; (2) A one-hidden layer neural network with 32 hidden ReLU neurons. The results for both models are presented in Appendix A.1.

**Linear regression with Gaussian mixture data**   For the Gaussian mixture data generation setup described above, we also perform ridge regression. The results (presented in Appendix A.2) demonstrate that classification loss and square loss often have a similar qualitative behavior as a function of weight $\alpha$, as seen by comparing the classification loss in Figure 7 and the squared loss in Figure 11 for the same setup. Although our theoretical results do not apply directly to classification loss, we believe that our qualitative conclusions generalize. This is confirmed by the similar behavior between the two losses and the successful prediction of actual risk by scaling laws in our classification experiments.

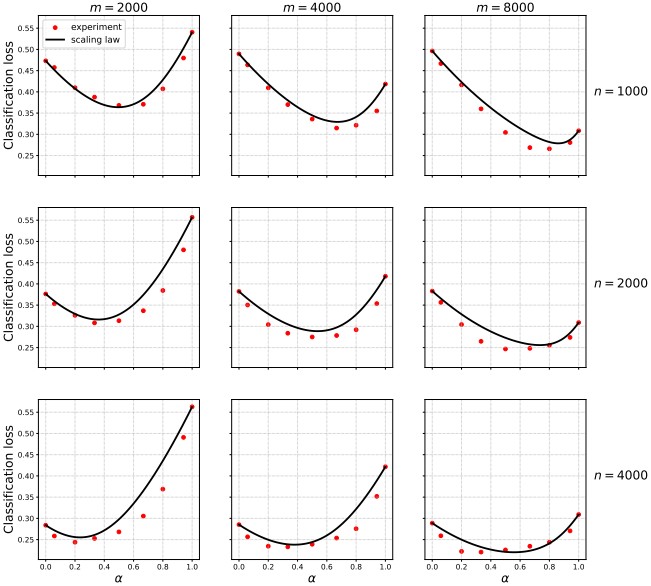

Figure 3: CIFAR10 and CIFAR100 data. Test error when trained on mixtures of original and surrogate data. Black curves: prediction from Eq. (4).

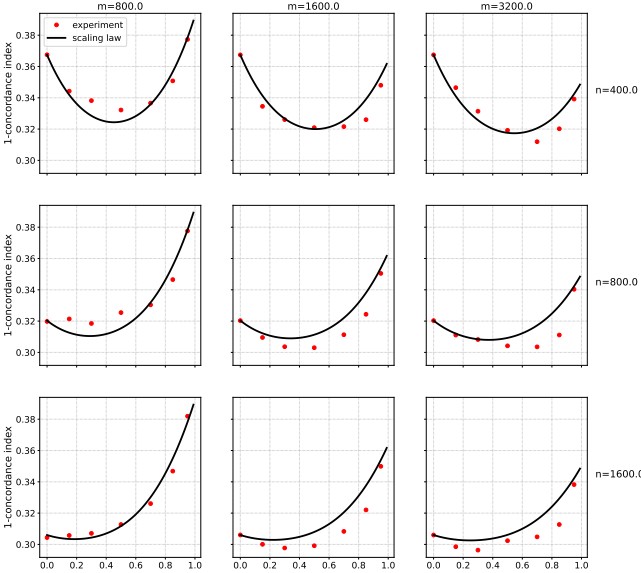

Figure 4: Lasso-based Cox regression on TCGA PanCancer dataset. Test error when trained on mixtures of original and surrogate data. Black curves: prediction from Eq. (4).

**Sentiment analysis in movie reviews**    As original data, we use the IMDB dataset (link) which has 25k reviews for training, each labeled as positive or negative. For validation and testing, we split the IMDB test dataset of 25k reviews into a validation set of 10k reviews and test set of 15k reviews.

We experiment with two different surrogate datasets: 1) Rotten Tomatoes dataset of movie reviews (link): these are data with different distribution but within the same domain. This dataset contains movie reviews and the corresponding sentiments, 2) Goodreads book reviews (link): these are data from a substantially different domain. This dataset has reviews and their ratings. We choose 10k reviews each with a rating of 5 and 1, and label them as positive and negative, respectively.

We convert reviews into feature vectors with $d = 884$ dimensions as explained in Appendix A.3. We use logistic regression and neural network models with the same set of parameters as in the Gaussian mixture experiments (except for the input dimension).

Results with neural nets and Rotten Tomatoes as synthetic dataset are presented in Figure 1 and the remaining results are in Appendix A.3.

**Image classification with CIFAR10 and CIFAR100**    We use 50,000 CIFAR10 training images as original data, its 10 classes for the classification task, and test on the 10,000 CIFAR10 test images. We use 50,000 CIFAR100 training images as surrogate data. We train a 9-layer ResNet model for classification. Appendix A.4 presents details on the data pre-processing and mapping of labels. Results are shown in Figure 3. Note that CIFAR10 and CIFAR100 datasets are quite different from each other, as they have no overlap either in the images or in their label sets. Yet, the test error on training on their mixture is well predicted by the scaling law (4).

**Lasso-based Cox regression on TCGA PanCancer dataset**    We use the public domain TCGA pancancer dataset [GCH+20] (link), with gene expressions as covariates and progression-free survival (PFS) as response. After filtering and feature selection, we are left with 3580 female patients, which we use as original data, and 3640 male patients, which we use as surrogate data. We fit CoxPHFitter model (link) with 500 selected genes and use "1-concordance score" as our loss function. The results are shown in Figure 4. The details of pre-processing and experiment parameters[3] are in Appendix A.5.

**High-dimensional ridge regression**    We simulate the data distribution in Section 3.4, i.e., $y_i = \langle \boldsymbol{\theta}_*, \boldsymbol{x}_i \rangle + \varepsilon_i$, $i \leq n$; $y_i^s = \langle \boldsymbol{\theta}_{*,s}, \boldsymbol{x}_i^s \rangle + \varepsilon_i^s$, $i \leq m$; with $\boldsymbol{x}_i, \boldsymbol{x}_i^s \sim \mathsf{N}(\boldsymbol{0}, \boldsymbol{I}_d)$, $\varepsilon_i \sim \mathsf{N}(0, \sigma^2)$,

---

[3]We observe that training at $\alpha = 1$ yields a somewhat singular behavior: we use a $\alpha = 0.95$ as a proxy of $\alpha = 1$, see appendices.

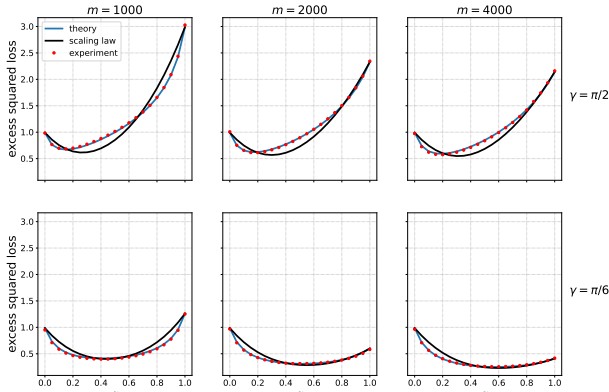

Figure 5: Ridge regression on simulated data. Here $d = 500$, $n = 1000$, $\sigma^2 = \sigma_s^2 = 1$, $\|\boldsymbol{\theta}_*\| = \|\boldsymbol{\theta}_{*,s}\| = 1$, regul. par. $\lambda = 2^{-10}$, and $m$ varies by column. Top row $\gamma = \pi/2$, bottom row $\gamma = \pi/6$.

$\varepsilon_i^s \sim \mathsf{N}(0, \sigma_s^2)$, and fit a simple linear model using ridge regression. The results are shown in Figure 5. In our experiments, we use $d = 500$, $\sigma^2 = \sigma_s^2 = 1$, $\|\boldsymbol{\theta}_*\| = \|\boldsymbol{\theta}_{*,s}\| = 1$ and regularization parameter $\lambda = 2^{-10}$. Under these settings, the model is parametrized by the angle $\gamma$ between $\boldsymbol{\theta}_*$ and $\boldsymbol{\theta}_{*,s}$, where $\cos\gamma := \langle \boldsymbol{\theta}_*, \boldsymbol{\theta}_{*,s} \rangle$. We used $\gamma = \pi/6$ and $\pi/2$ in our experiments.[4]

The theoretical predictions of Theorem 3 for these curves in high-dimensional asymptotics $n, m, d \to \infty$, with $n/d \to \delta$, $m/d \to \delta_s$ are reported as blue lines, and match remarkably well with the empirical data. The simple scaling law (4) nevertheless provides a good approximation of these (more complicated) theoretical formulas.

Note in particular that in the top row of Figure 5, we have $\langle \boldsymbol{\theta}_*, \boldsymbol{\theta}_{*,s} \rangle = 0$, i.e. the surrogate data are as far as possible from the original ones. Nevertheless, the induced regularization effect leads to smaller test error on the original distribution.

We observe proposed scaling law (4) predicts well the behavior of the experiments, across of the datasets above, and for most combinations of original and surrogate examples we have tested.

Finally, we emphasize that the scaling law is only an empirical approximation of reality. This is clearly illustrated by the example of ridge regression: in this case, we use Theorem 3 to precisely predict the discrepancy between precise asymptotics and scaling law, see Appendix A.6.

## 5 Discussion

We conclude by discussing two possible generalizations of the scaling law (4), and its applicability. *First,* throughout this paper we assumed that $R_{\mathsf{or}}^{\mathsf{ex}}(\infty) = 0$, namely that we can achieve the Bayes error by training on infinitely many original samples. In practice this will not hold because of the limited model complexity. Following standard scaling laws [KMH+20, HBM+22], this effect can be accounted for by an additional term $C \cdot N^{-\omega}$, where $N$ is the model size (number of parameters). *Second,* the scaling law (4) implies as special cases that $R_{\mathsf{or}}^{\mathsf{ex}}(n) \approx A_{\mathsf{or}} n^{-\beta}$, $R_{\mathsf{su}}^{\mathsf{ex}}(m) \approx R_{\mathsf{su}}^{\mathsf{ex}}(\infty) + A_{\mathsf{su}} m^{-\beta}$. In particular, the exponent $\beta$ is the same when training on real or surrogate data. In practice, we observe often two somewhat different exponents $\beta_{\mathsf{or}} \neq \beta_{\mathsf{su}}$. In these cases, we set $\beta = \beta_{\mathsf{or}}$, and this appears to work reasonably well. However, we can imagine cases in which the difference between $\beta_{\mathsf{or}}$ and $\beta_{\mathsf{su}}$ is significant enough (4) will stop being accurate.

## Acknowledgements

We are grateful to Joseph Gardi, Germain Kolossov, Marc Laugharn, Kaleigh Mentzer, Rahul Ponnala, and Pulkit Tandon, for several conversations about this work. This work was carried out while Andrea

---

[4]For ridge regression simulations, we directly plot the excess test risks, as the parameter $\boldsymbol{\theta}$ for original data is known. For any $\hat{\boldsymbol{\theta}}$ the excess test risk in this model is simply $\|\boldsymbol{\theta} - \hat{\boldsymbol{\theta}}\|^2$.

Montanari was on leave from Stanford and a Chief Scientist at Granica (formerly known as Project N). The present research is unrelated to AM's Stanford research.

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

# A Details of empirical results

## A.1 Binary classification with Gaussian mixture data

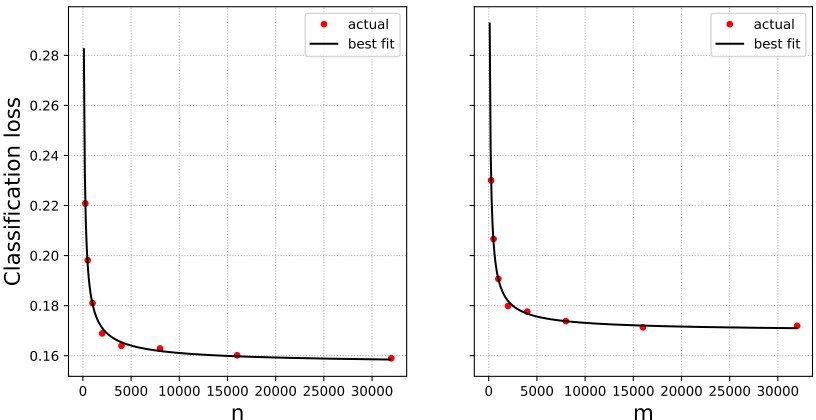

Figure 6: Gaussian mixture data and logistic regression. Test error when trained on original (left plot) and surrogate (right plot) data only (red dots). Best fits are shown in black. These gives the estimates $\beta = 0.72$, $R_* = 0.157$, and $R_{su}^{ex}(\infty) = 0.013$.

We provide details for the models used in the simulations.

**Logistic regression**: We use the scikit-learn implementation with the lbfgs solver, fitting the intercept, with maximum iterations set to 10k. For each run of each $(n, m, \alpha)$ combination, we set the $\ell_2$ penalty (parameter C in scikit-learn) to $2^i, i = -8, ..., 8$ and $10^i, i = -6, -5, -4, -3, 3, 4, 5, 6$, and only report the test result for the value that achieves the best validation error. The results of the individual scaling law estimates and the comparison of joint training results with the scaling law predictions are shown in Figures 6 and 7.

**Neural network**: The network has one hidden layer with 32 ReLU neurons, and an output neuron using sigmoid. For training, we use the binary cross entropy loss, a constant learning rate of 0.05, and batch size 64. We train the network for 1,000 epochs. Similar to the procedure in logistic regression, we use $\ell_2$ regularization (weight decay) and use the validation set to choose the best regularization parameter from the set $\{0, 10^{-5}, 10^{-4}, 10^{-3}, 2 \cdot 10^{-3}, 4 \cdot 10^{-3}, 10^{-2}, 2 \cdot 10^{-2}, 4 \cdot 10^{-2}, 10^{-1}, 2 \cdot 10^{-1}, 4 \cdot 10^{-1}\}$. The results of the individual scaling law estimates and the comparison of joint training results with the scaling law predictions are shown in Figures 8 and 9.

## A.2 Linear regression with Gaussian mixture data

For the Gaussian mixture data, described in the previous section, we perform weighted ridge regression experiments according to equation (16) and plot the square loss. As before, we choose the best regularizer for the ridge regression of the set $2^i, i = -8, ..., 8$ and $10^i, i = -6, -5, -4, -3, 3, 4, 5, 6$, and report the test result for the value that achieves the best validation error. The results are presented in Figures 10 and 11.

## A.3 Sentiment analysis in movie reviews

To convert the movie reviews and book reviews to vectors, we use a combination of two different embedding: We use all the reviews in the training data and then use nltk tagger [Bir06] to find the most frequent 500 adjectives appearing in the samples used for training. Then we use the common Tfidf vectorizer (we used scikit-learn's implementation of tfidf vectorizer) for which we use the list of these most common 500 adjectives as vocabulary. This gives us a vector of length 500 dimension for each review. In addition, we also apply "Paraphrase-MiniLM-L6-v2" sentence transformer which is based on BERT with 6 Transformer Encoder Layers, and return a 384 dimension vector representation of the reviews. For each movie review we concatenate the results of tfidf vectorizer and sentence transformer to get a 884 dimensional representation that we use as our input vector.

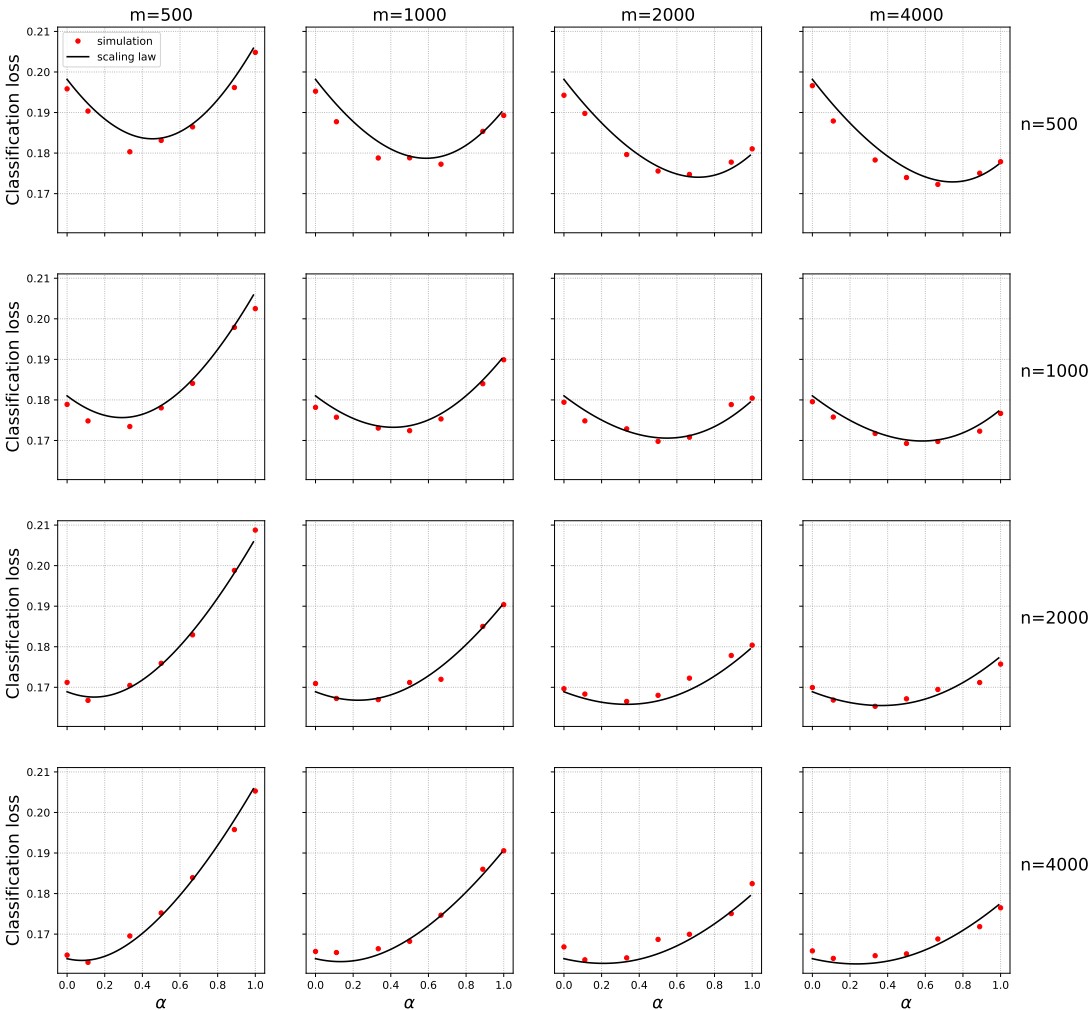

Figure 7: Gaussian mixture data and logistic regression. Test error when trained on mixtures of original ($n$ varying by row) and surrogate ($m$ varying by column) data. Black curves: scaling formula (4).

We use logistic regression and neural networks with the same set of parameters as in the Gaussian mixture experiments (except for the input dimension). We plot the average loss over 10 independent runs.

Results omitted from the main text are presented in Figures 12–16.

### A.4 Image classification with CIFAR10 and CIFAR100

We largely use the model and the training procedure described at https://jovian.ml/aakashns/05b-cifar10-resnet. We normalize the images for mean and standard deviation. We train a 9-layer ResNet model for classification, using Adam for optimization, weight decay, and gradient clipping, trained over 16 epochs with a one-cycle learning rate scheduling policy, minimizing cross entropy loss. For each combination of $m$, $n$, and $\alpha$, we report the average test error over 10 runs. Since there is no overlap between the label sets of CIFAR10 and CIFAR100, the latter dataset needs to be relabeled. We do this by training a separate 9-layer ResNet model on 10,000 randomly chosen CIFAR10 images from the training set of 50,000 examples (without creating a separate split for them), and use its predictions on CIFAR100 images as labels.

Scaling curves are presented in Figure 17 and 3.

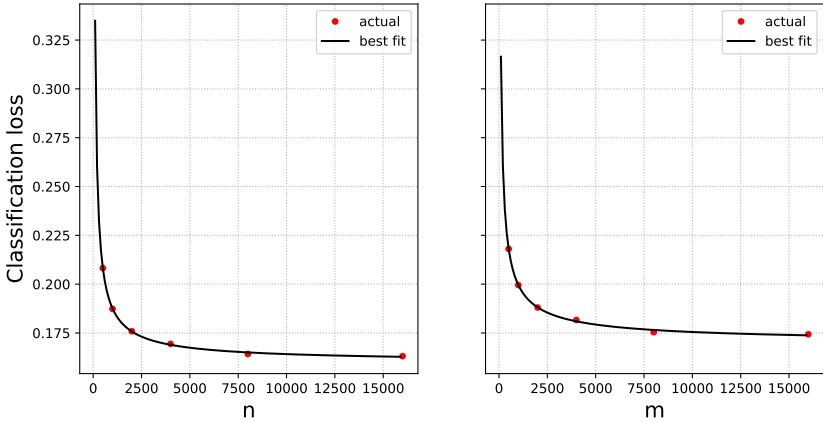

Figure 8: Gaussian mixture data and neural network. Test error when trained on original (left plot) and surrogate (right plot) data only (red dots). Best fits are shown in black. These gives the estimates $\beta = 0.79$, $R_* = 0.160$ and $R_{su}^{ex}(\infty) = 0.010$.

### A.5   Lasso on TCGA PanCancer dataset

We used public domain TCGA pancancer dataset. After, filtering samples with incomplete values we are left with 9220 patients, each having 20,531 gene expression values and the outcome was PFS (progression-free survival). Out of these we used a group of 2000 patients, splitted into train and test set of 1000 each to select 500 genes having the largest absolute Cox PH score. We also used the mean and standard deviation of gene expression values of these 2000 patients to normalize the gene expression columns for the remaining 7220 patients. Among the remaining of 7220 patients 3580 were females. We treated the female patients data as original data, and split them into train (50%), test (25%) and validation split (25%). The remaining 3640 patients data was used as surrogate dataset. We fit CoxPHFitter model (link) with 500 selected genes and use "1-concordance score" as our loss function. We used the validation split to choose best value of $\ell_1$ penalty parameter from $2^i, i = 2, 0, -2, -4, -6, -8, -10, -12, -14, -16$ in the model. We observed discontinuity at $\alpha = 1$. To avoid this discontinuity, we approximated $R(\hat{\boldsymbol{\theta}}_{n,m}(1))$ by $R(\hat{\boldsymbol{\theta}}_{n,m}(1-\epsilon))$ if $n > 0$ and by $R(\hat{\boldsymbol{\theta}}_{m/2,m}(1-\epsilon))$ if $n = 0$, where we choose $\epsilon = 0.05$. We plot the average loss over 10 independent runs. The results are presented in Figures 18 and 4.

### A.6   High-dimensional ridge regression

We present additional ridge regression experiments here in Figs. 19–30. We plot the average loss over 10 independent runs. In these experiments, as in the main paper, we set $d = 500$, $\sigma^2 = \sigma_s^2 = 1$, $\|\boldsymbol{\theta}_*\| = 1$, $\|\boldsymbol{\theta}_{*,s}\| = 1$, except for the last four Figs. 27–30, where we use $\|\boldsymbol{\theta}_{*,s}\| = 1/2$. We used angle $\gamma = \pi/6$ and $\pi/2$ in our experiments.

We consider two methods: (1) Fix $\lambda$ to a very small value $2^{-10}$, and (2) For each random draw of datasets select $\lambda$ that achieves the best validation performance. For the latter method, we try $\lambda = 2^i$, where $i = -10, -8, -6, \ldots, 8, 10$. For ridge regression simulations, we directly plot the excess test risks, as the parameter $\theta$ for original data is known and for any $\hat{\theta}$ the excess test risk in this model is $\|\theta - \hat{\theta}\|^2$.

## B   Low-dimensional asymptotics

### B.1   Formal statements

In this appendix, we present our results on the estimator of Eqs. (2), (3) under the classical asymptotics $n, m \to \infty$ at $d$ fixed. For simplicity, we assume no regularizer is used in this regime.

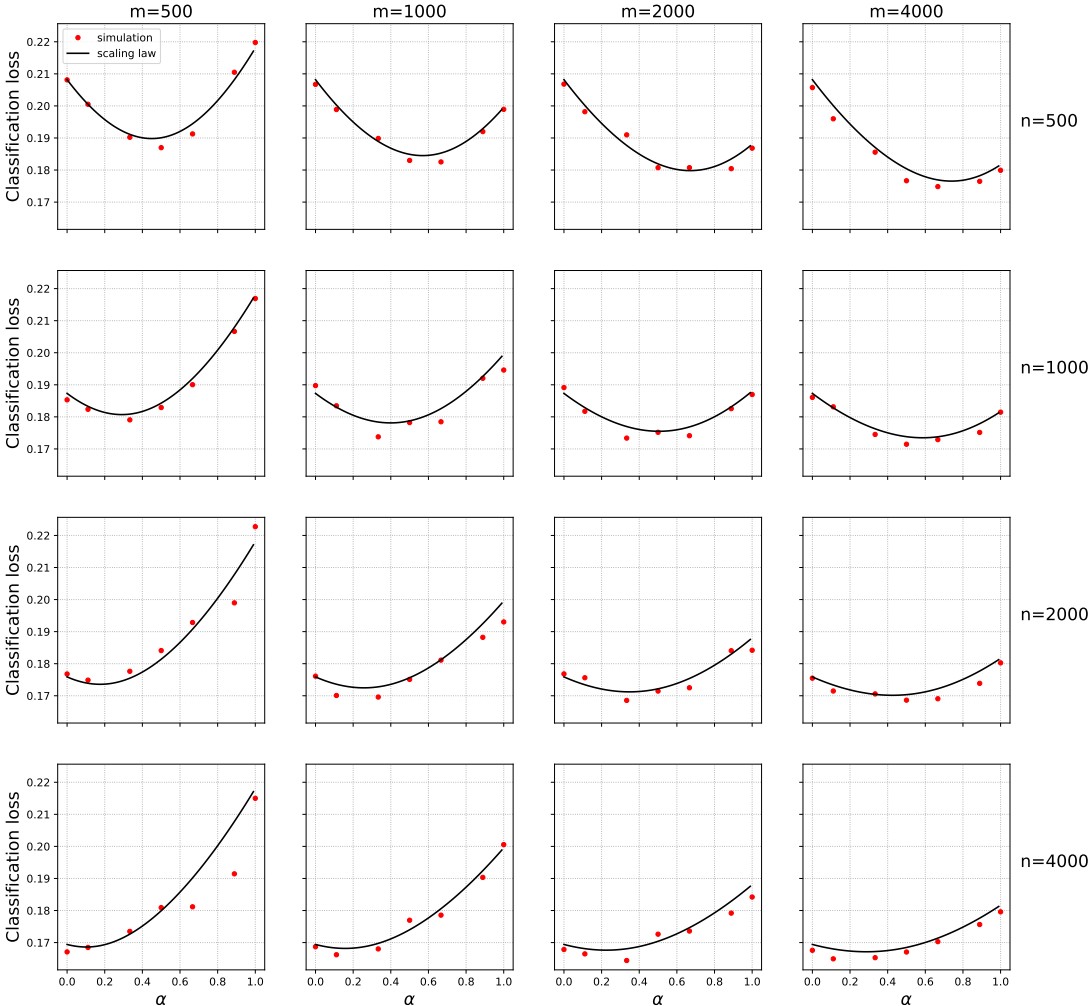

Figure 9: Gaussian mixture data and neural network. Test error when training mixture of original ($n$ varying by row) and surrogate ($m$ varying by column) data. Black curves: scaling law (4).

Beyond classical regularity assumptions of low-dimensional asymptotics, in this section we will make the following assumption which guarantees that original and surrogate distribution are 'not arbitrarily far.' Recall that $R^s(\boldsymbol{\theta})$ denotes the population error on surrogate data.

**Assumption 1** (Distribution shift for low-$d$ asymptotics)**.** *There exists a constant $K_*$ such that for all $\boldsymbol{\theta} \in \mathbb{R}^d$,*

$$\left| R^s(\boldsymbol{\theta}) - R(\boldsymbol{\theta}) \right| \leq K_* \big( 1 + R(\boldsymbol{\theta}) \big). \tag{18}$$

The regularity conditions are similar to the ones in [vdV00]. Here and in the following $\mathsf{B}(\boldsymbol{\theta}_*, r)$ is the ball of radius $r$ centered at $\boldsymbol{\theta}_*$.

**Assumption 2** ('Classical' regularity)**.**

  (a) *The original population risk $R(\boldsymbol{\theta})$ is uniquely minimized at a point $\boldsymbol{\theta}_*$.*

  (b) *$\boldsymbol{\theta} \mapsto \ell(\boldsymbol{\theta}; \boldsymbol{z})$ is non-negative lower semicontinuous. Further, define the following limit in $[0, \infty]$ for $\boldsymbol{u} \in \mathbb{S}^{d-1}$:*

$$\ell_\infty(\boldsymbol{u}; \boldsymbol{z}) := \liminf_{\substack{\boldsymbol{\theta} \to \infty \\ \boldsymbol{\theta}/\|\boldsymbol{\theta}\|_2 \to \boldsymbol{u}}} \ell(\boldsymbol{\theta}; \boldsymbol{z}). \tag{19}$$

  *Then we assume $\inf_{\boldsymbol{u} \in \mathbb{S}^{d-1}} \mathbb{E}\ell_\infty(\boldsymbol{u}; \boldsymbol{z}) \geq R(\boldsymbol{\theta}_*) + c$ for some $c > 0$.*

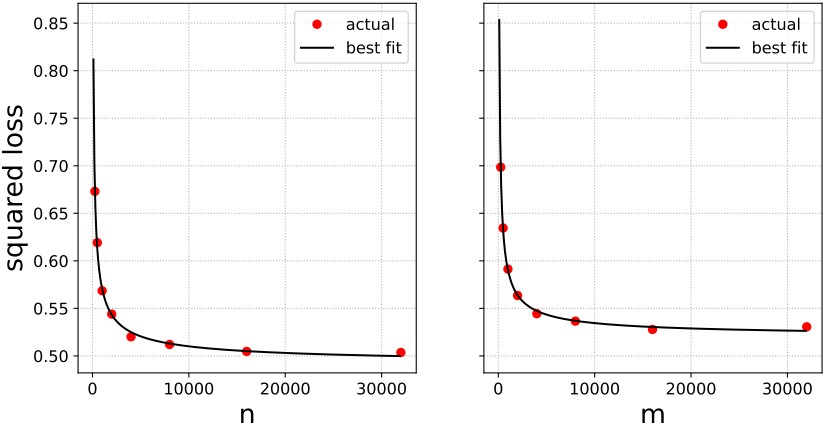

Figure 10: Gaussian mixture data and ridge regression. Test error when trained on original (left plot) and surrogate (right plot) data only (red dots). Best fits are shown in black. These gives the estimates $\beta = 0.60$, $R_* = 0.49$, and $R_{\text{su}}^{\text{ex}}(\infty) = 0.03$.

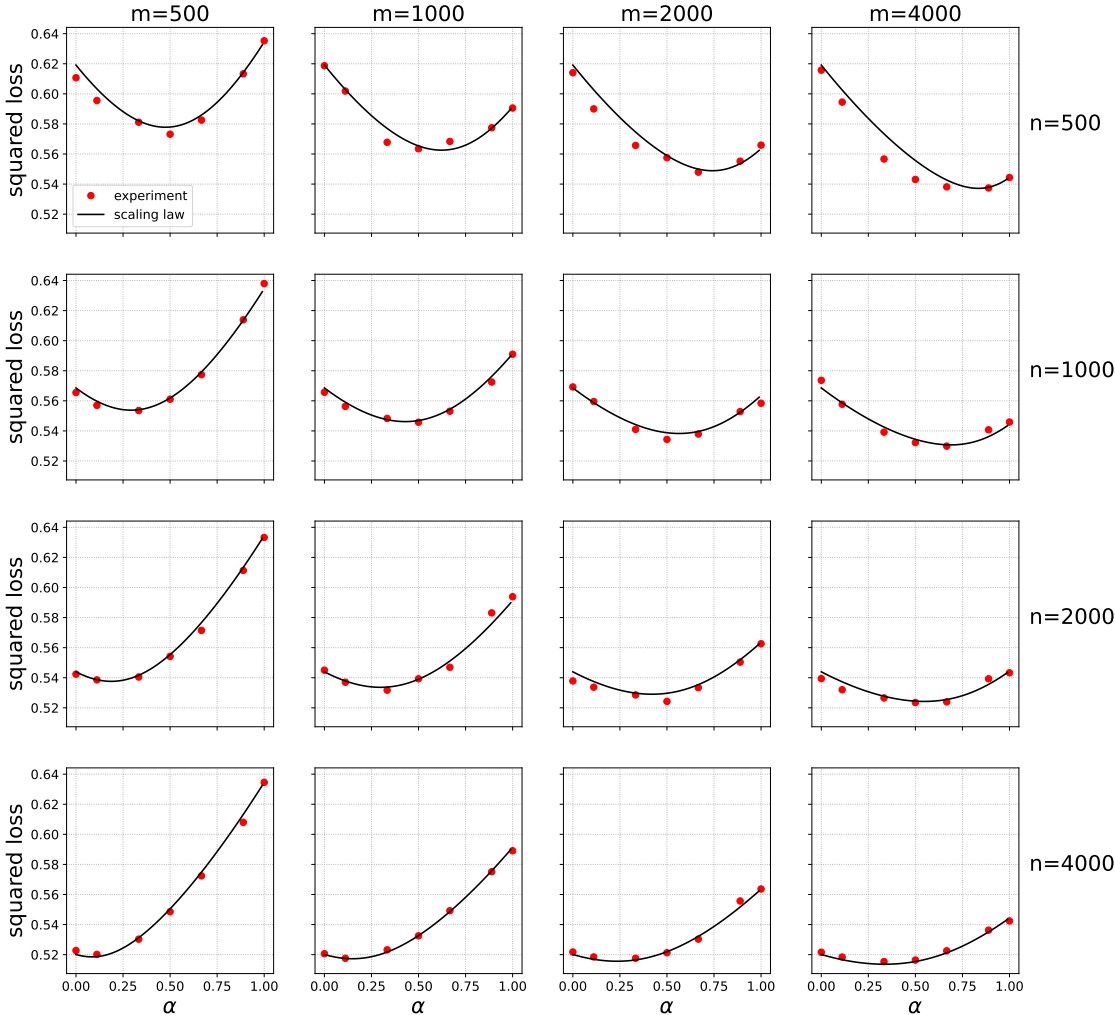

Figure 11: Gaussian mixture data and ridge regression. Test error when trained on mixtures of original and surrogate data. Black curves: prediction from Eq. (4).

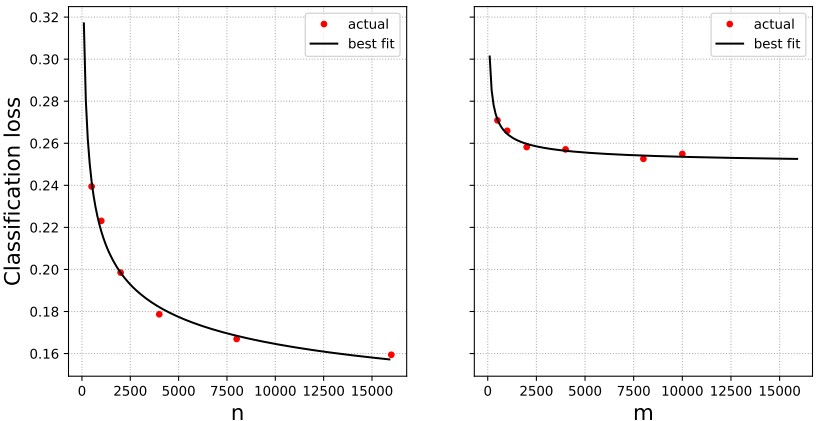

Figure 12: IMDB and Rotten Tomatoes data and logistic regression. Test error when trained on original (left plot) and surrogate (right plot) data only (red dots), together with scaling law fits (black lines). Best fit parameters are $\beta = 0.27$, $R_* = 0.101$ and $R_{\mathsf{su}}^{\mathsf{ex}}(\infty) = 0.148$.

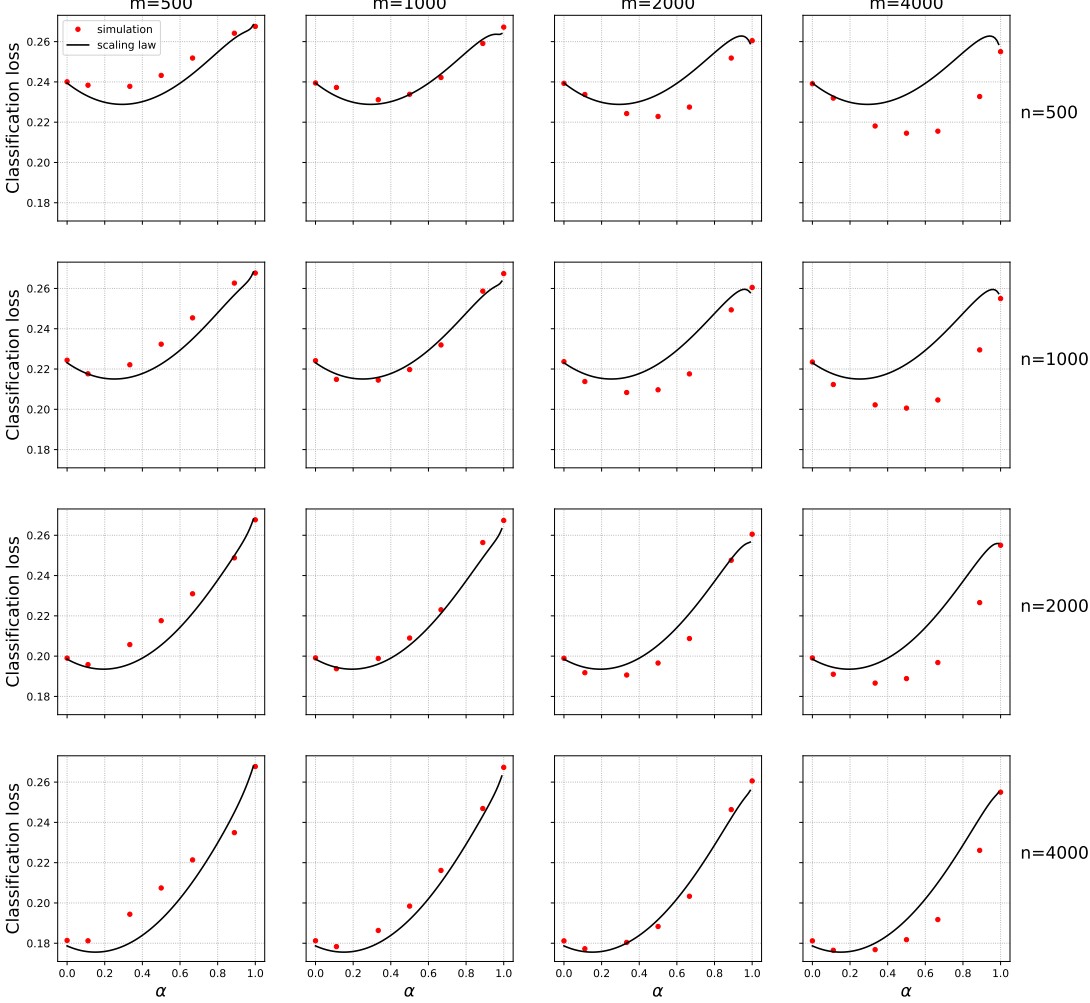

Figure 13: IMDB and Rotten Tomatoes data and logistic regression. Test error when trained on mixtures of original and surrogate data. Black curves: prediction from Eq. (4).

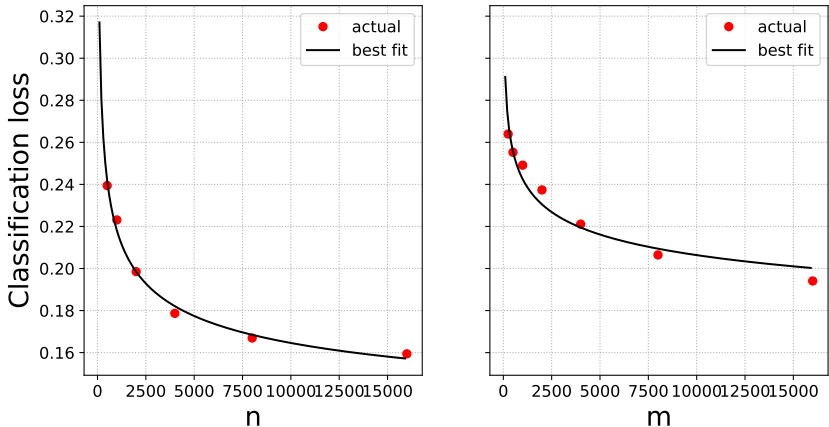

Figure 14: IMDB and Goodreads book reviews (as surrogate dataset) and logistic regression. Test error when trained on original (left plot) and surrogate (right plot) data only (red dots), together with scaling law fits (black lines). Best fit parameters are $\beta = 0.27$, $R_* = 0.101$ and $R_{\mathsf{su}}^{\mathsf{ex}}(\infty) = 0.101$.

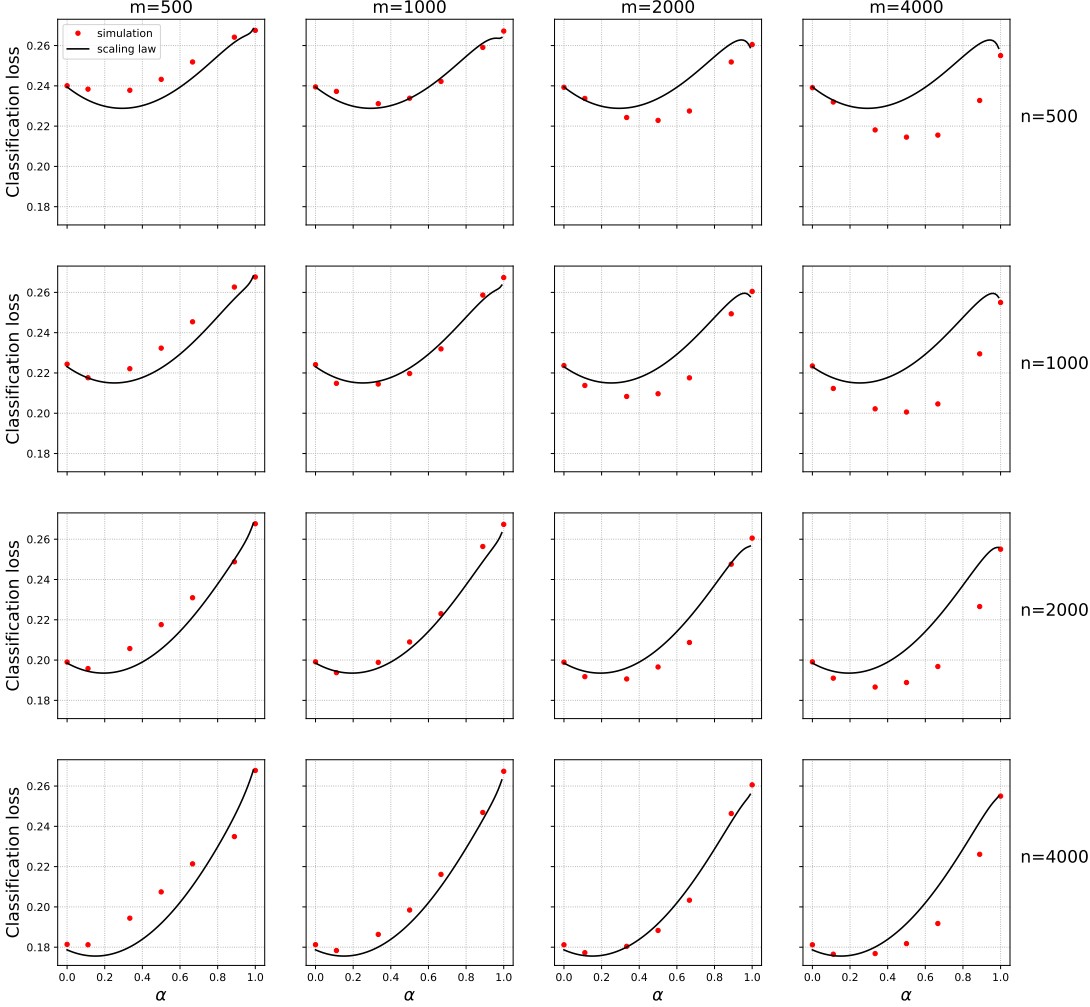

Figure 15: IMDB and Goodreads book reviews and logistic regression. Test error when trained on mixtures of original and surrogate data. Black curves: prediction from Eq. (4).

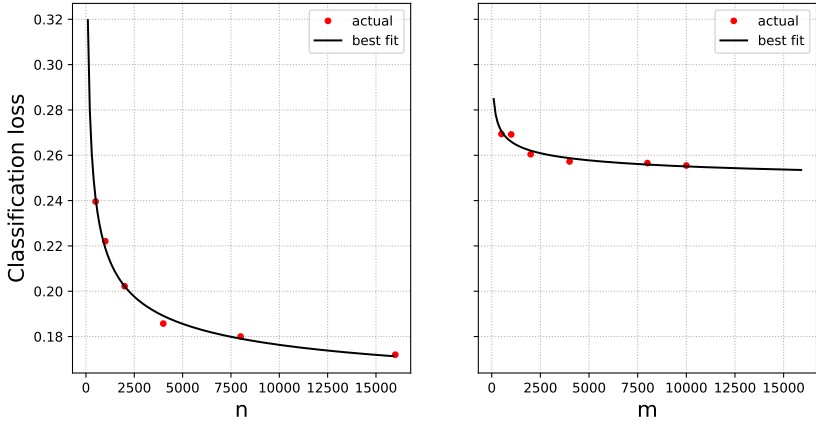

Figure 16: IMDB and Rotten Tomatoes data and neural networks. Scaling law fits for models trained on original (left plot) and surrogate (right plot) data only (red dots)(as in Fig. 12), together with scaling law fits (black lines). Best fit parameters are $\beta = 0.37$, $R_* = 0.145$ and $R_{su}^{ex}(\infty) = 0.095$.

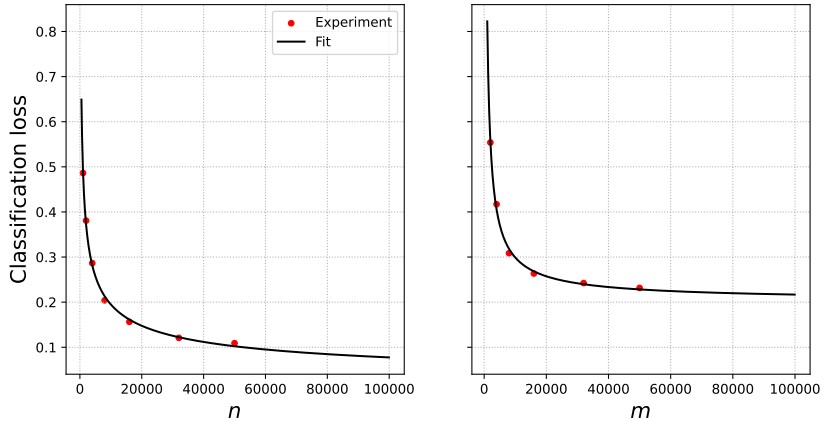

Figure 17: CIFAR10 and CIFAR100 data: (left) Test error scaling of original data (left) and surrogate data (right). Best fit parameters are $\beta = 0.404$, $R_* = 0.0013$, and $R_{su}^{ex}(\infty) = 0.199$.

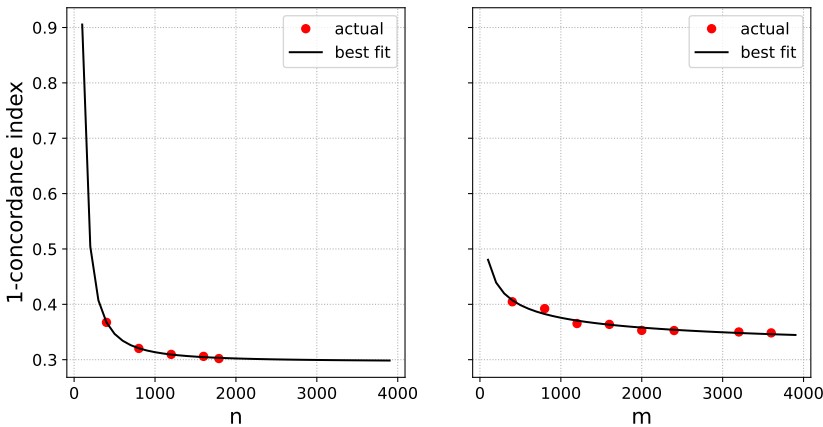

Figure 18: Lasso-based Cox regression on TCGA PanCancer dataset with female patients data as original data and male patients data as surrogate data. Scaling law fits for models trained on original (left plot) and surrogate (right plot) data only (red dots)(as in Fig. 12.) Best fit parameters are $\beta = 1.55$, $R_* = 0.29$ and $R_{su}^{ex}(\infty) = 0.29$.

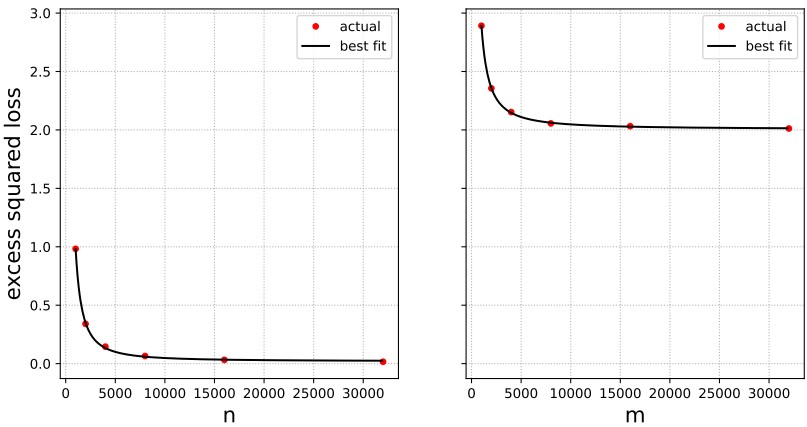

Figure 19: Ridge regression with $\gamma = \pi/2$, and regularization parameter $\lambda = 2^{-10}$: Test error scaling of the original data (left), and surrogate data (right). Best curve fits give the estimates $\beta = 1.57$ and $R_{\mathsf{su}}^{\mathsf{ex}}(\infty) = 2.0$

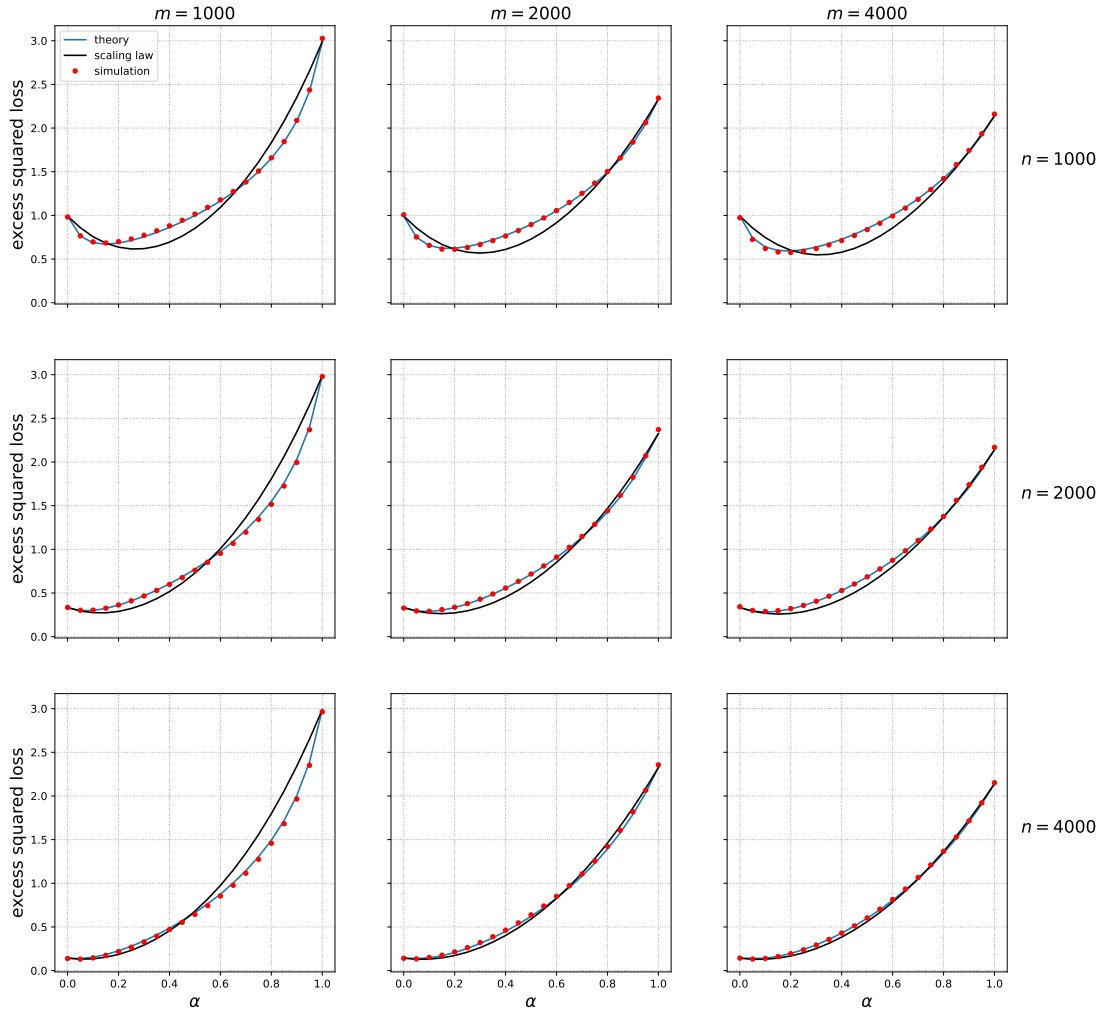

Figure 20: Ridge regression with $\gamma = \pi/2$, and regularization parameter $\lambda = 2^{-10}$

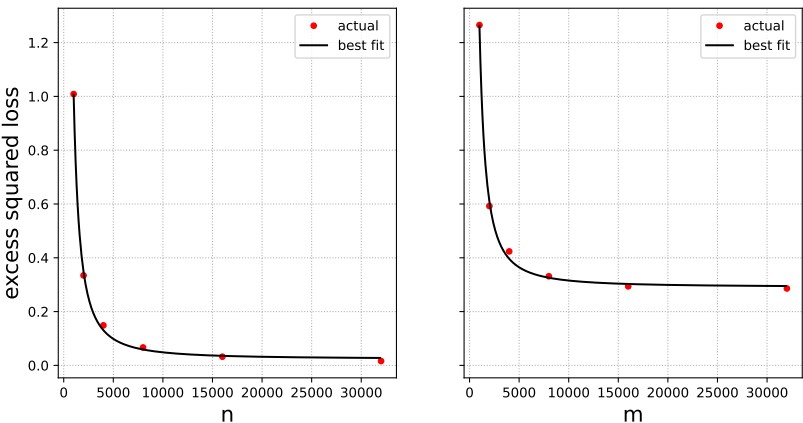

Figure 21: Ridge regression with $\pi/6$ between $\theta$ and $\theta_s$, and regularization parameter $\lambda = 2^{-10}$: Test error scaling of the original data (left), and surrogate data (right). Best curve fits give the estimates $\beta = 1.57$ and $R_{\mathsf{su}}^{\mathsf{ex}}(\infty) = 0.29$

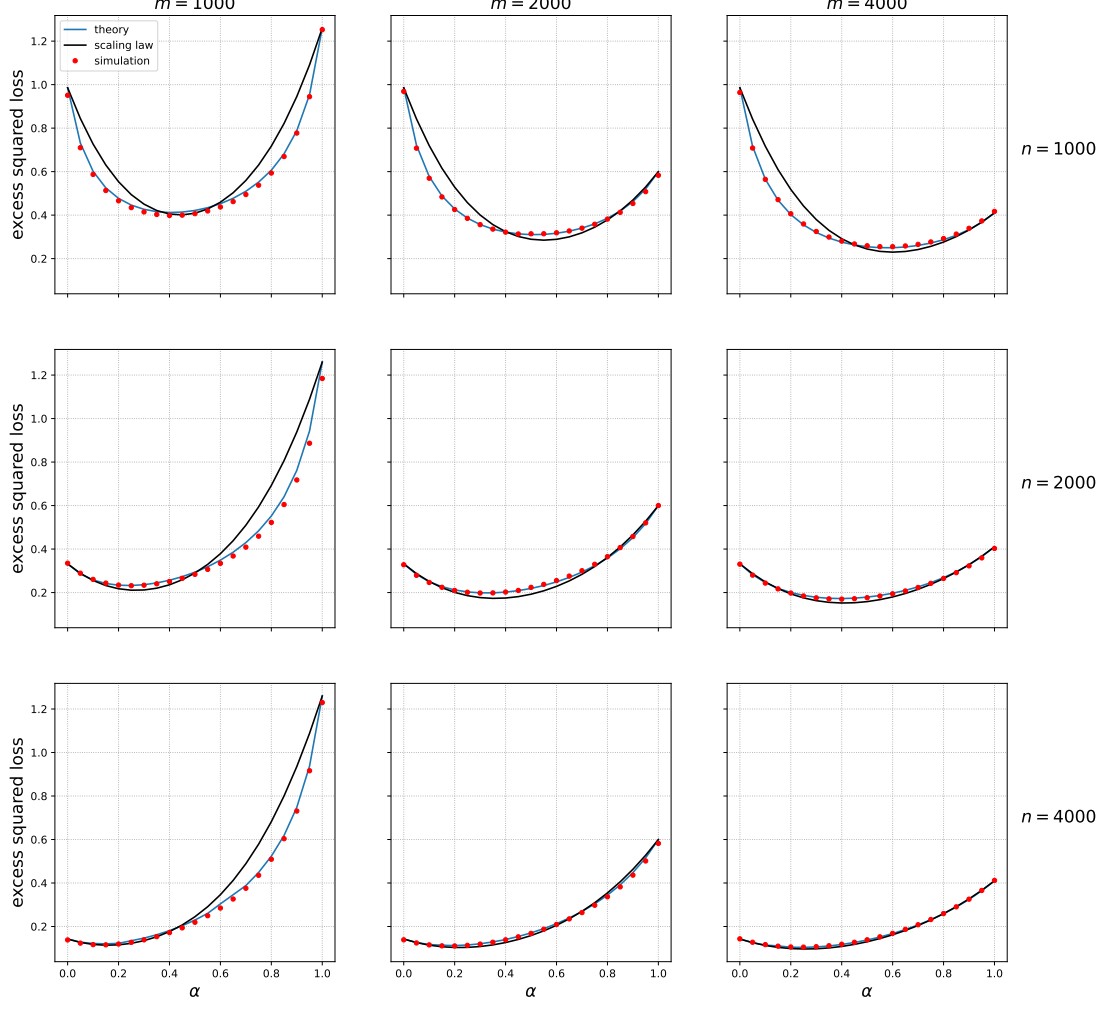

Figure 22: Ridge regression with $\pi/6$ between $\theta$ and $\theta_s$, and regularization parameter $\lambda = 2^{-10}$

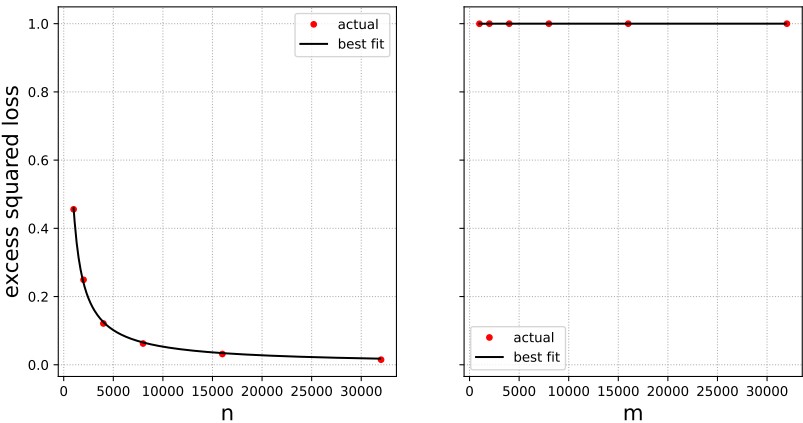

Figure 23: Ridge regression with $\pi/2$ between $\theta$ and $\theta_s$, and the best regularization parameter: Test error scaling of the original data (left), and surrogate data (right). Best curve fits give the estimates $\beta = 0.94$ and $R_{\mathsf{su}}^{\mathsf{ex}}(\infty) = 1.0$

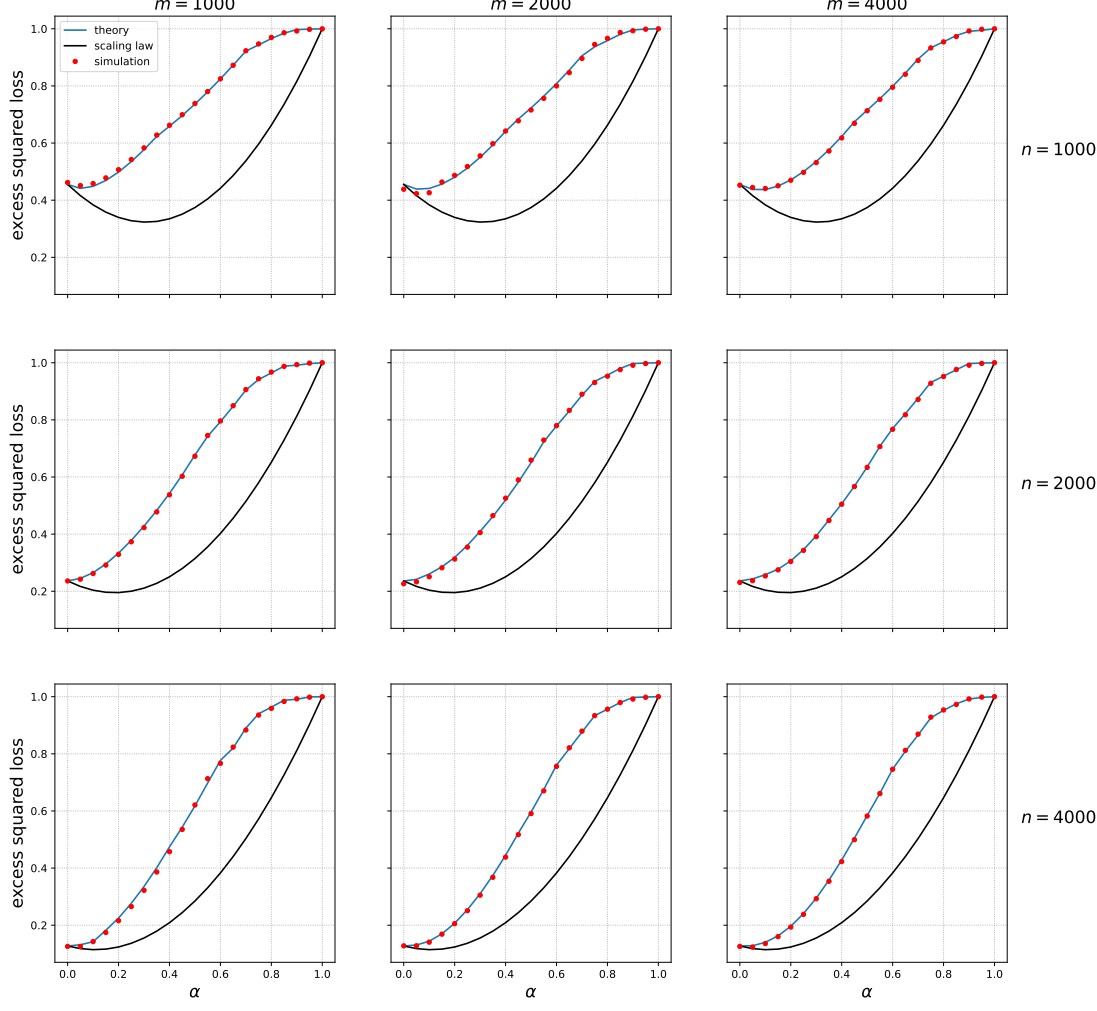

Figure 24: Ridge regression with $\gamma = \pi/2$, and the best regularization parameter

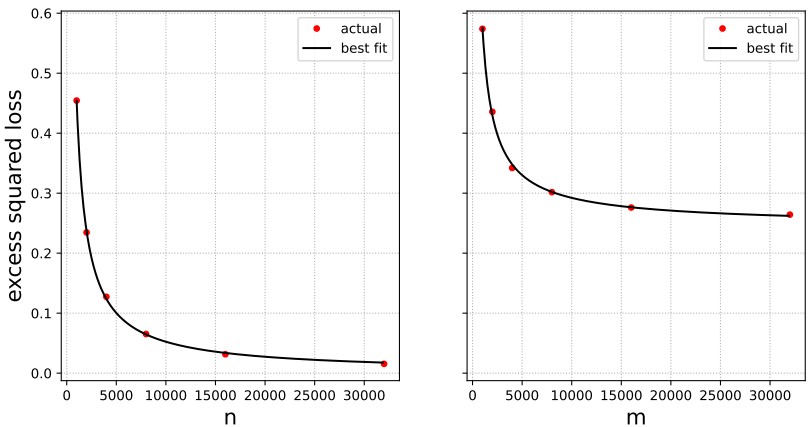

Figure 25: Ridge regression with $\pi/6$ between $\theta$ and $\theta_s$, and the best regularization parameter: Test error scaling of the original data (left), and surrogate data (right). Best curve fits give the estimates $\beta = 0.94$ and $R_{\mathsf{su}}^{\mathsf{ex}}(\infty) = 0.24$

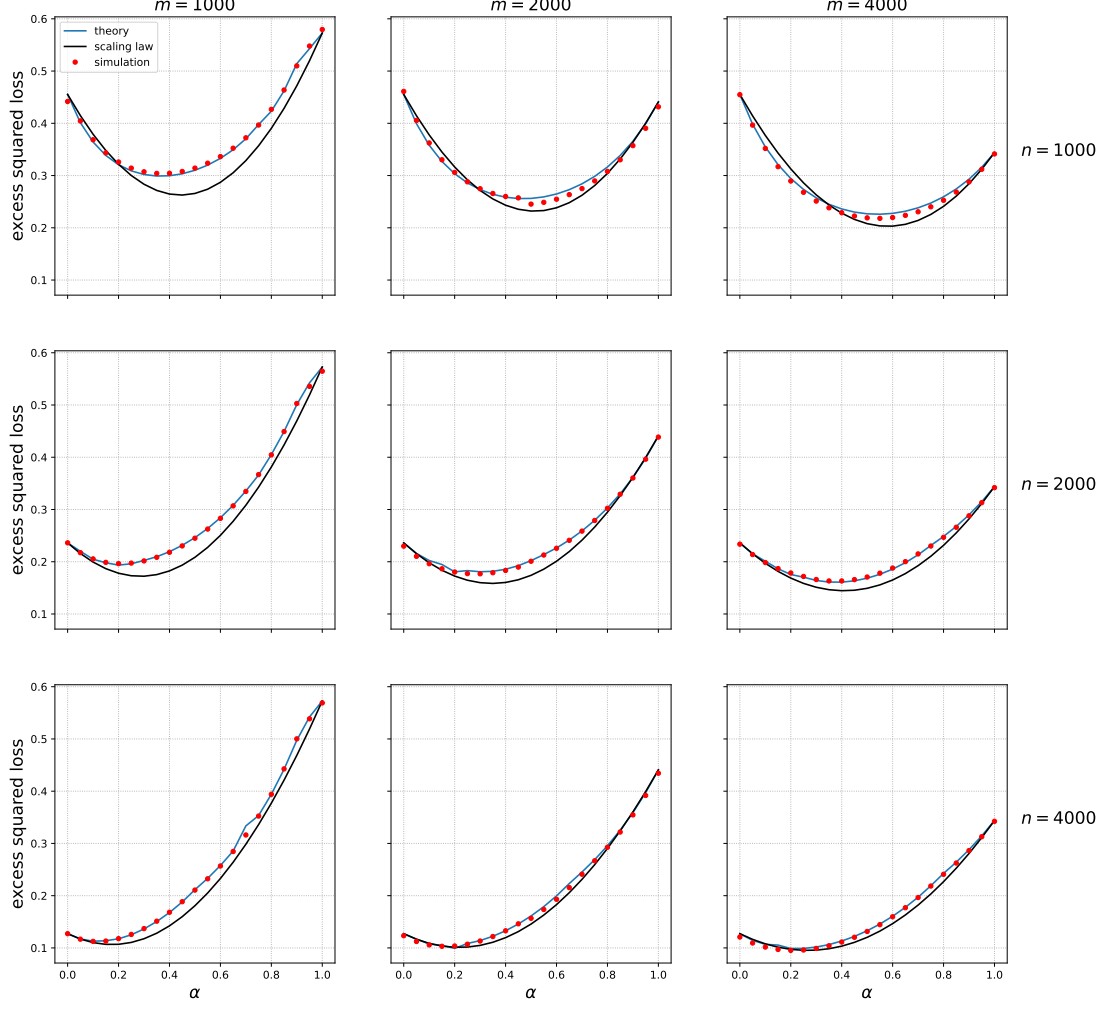

Figure 26: Ridge regression with $\pi/6$ between $\theta$ and $\theta_s$, and the best regularization parameter

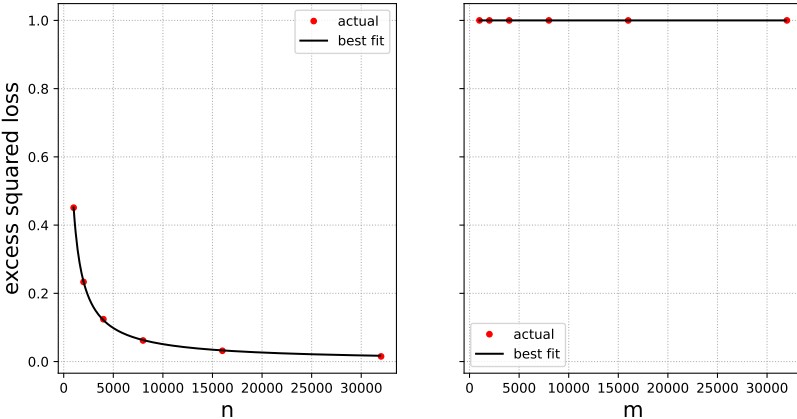

Figure 27: Ridge regression with $\pi/2$ between $\theta$ and $\theta_s$, $\|\theta\| = 1$, $\|\theta_s\| = 1/2$ and the best regularization parameter: Test error scaling of the original data (left), and surrogate data (right). Best curve fits give the estimates $\beta = 0.94$ and $R_{\mathsf{su}}^{\mathsf{ex}}(\infty) = 1.00$

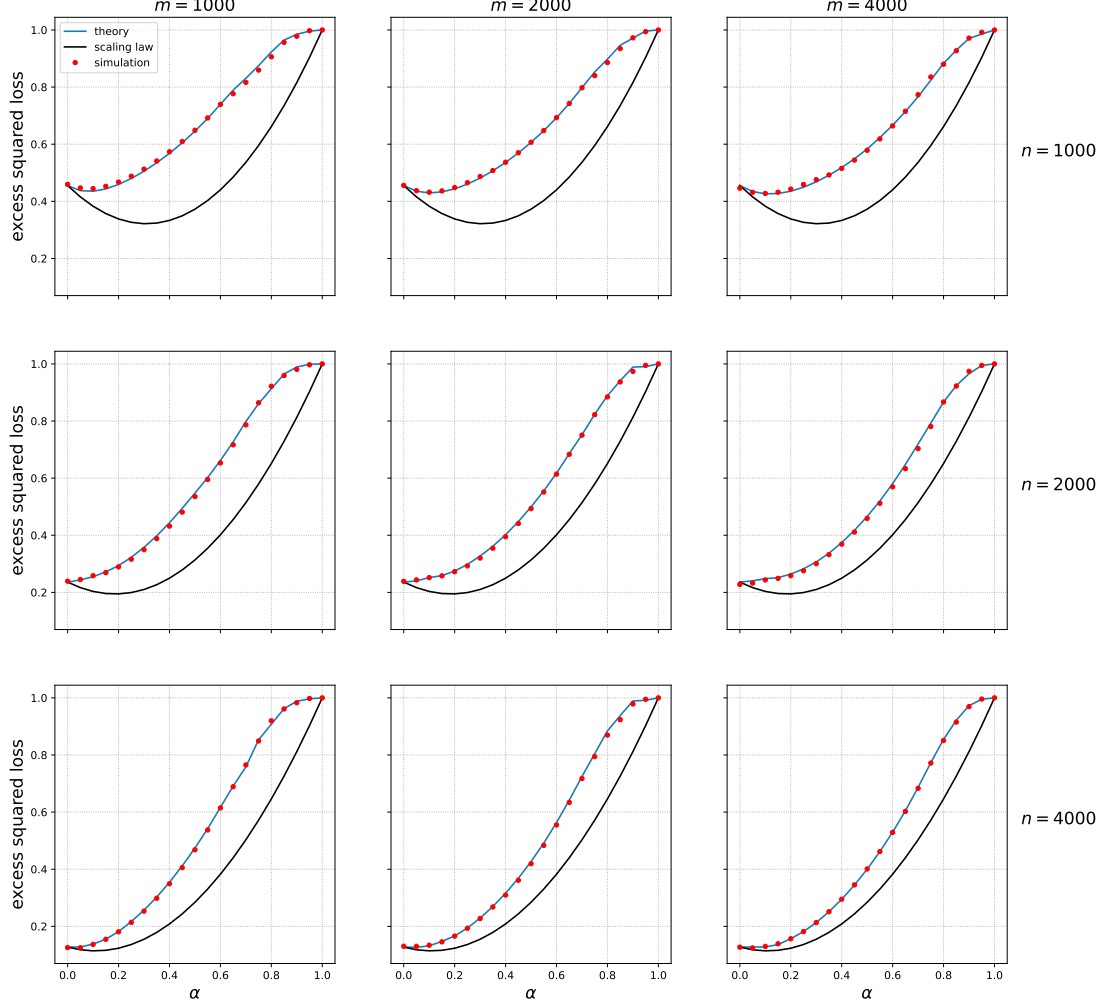

Figure 28: Ridge regression with $\pi/2$ between $\theta$ and $\theta_s$, $\|\theta\| = 1$, $\|\theta_s\| = 1/2$ and the best regularization parameter

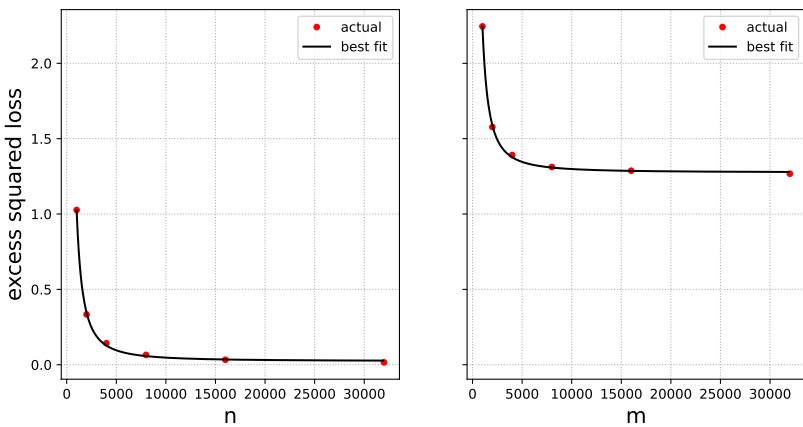

Figure 29: Ridge regression with $\gamma = \pi/2$, $\|\theta\| = 1$, $\|\theta_s\| = 1/2$, and regularization parameter $\lambda = 2^{-10}$: Test error scaling of the original data (left), and surrogate data (right). Best curve fits give the estimates $\beta = 1.57$ and $R_{\text{su}}^{\text{ex}}(\infty) = 1.27$

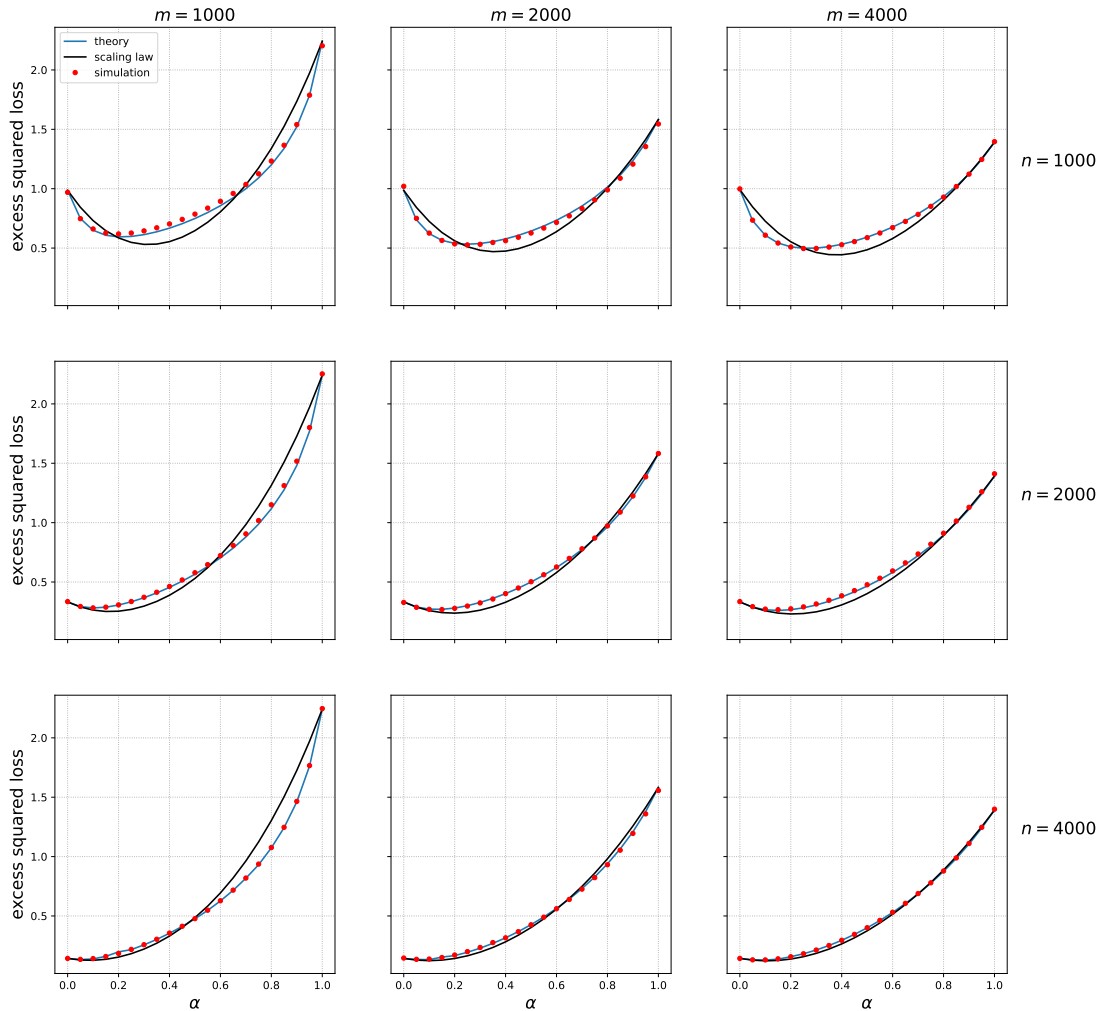

Figure 30: Ridge regression with $\gamma = \pi/2$, $\|\theta\| = 1$, $\|\theta_s\| = 1/2$, and regularization parameter $\lambda = 2^{-10}$

(c) $\boldsymbol{\theta} \mapsto \ell(\boldsymbol{\theta}; \boldsymbol{z})$ is differentiable at $\boldsymbol{\theta}_*$ almost surely, both under $\boldsymbol{z} \sim \mathbb{P}$ and under $\boldsymbol{z} \sim \mathbb{P}^s$. Further, there exists $r > 0$ such that, letting $\mathsf{B} := \mathsf{B}(\boldsymbol{\theta}_*, r)$, the following holds for a constant $C$:

$$\mathbb{E} \sup_{\boldsymbol{\theta}_1 \neq \boldsymbol{\theta}_2 \in \mathsf{B}} \left\{ \frac{|\ell(\boldsymbol{\theta}_1; \boldsymbol{z}) - \ell(\boldsymbol{\theta}_2; \boldsymbol{z})|^2}{\|\boldsymbol{\theta}_1 - \boldsymbol{\theta}_2\|_2^2} \right\} \leq C < \infty. \tag{20}$$

(d) The functions $\boldsymbol{\theta} \mapsto R(\boldsymbol{\theta})$, $\boldsymbol{\theta} \mapsto R^s(\boldsymbol{\theta})$, are twice differentiable in a neighborhood of $\boldsymbol{\theta}_*$, with Lipschitz continuous Hessian. Further $\nabla^2 R(\boldsymbol{\theta}_*) \succ \mathbf{0}$ (strictly positive definite).

**Proposition B.1.** *Under Assumption 1 and Assumption 2, define the following $d \times d$ matrices*

$$\boldsymbol{H} := \nabla^2 R(\boldsymbol{\theta}_*) = \mathbb{E}[\nabla^2 \ell(\boldsymbol{\theta}_*; \boldsymbol{z})], \tag{21}$$

$$\boldsymbol{K} := \mathrm{Cov}(\nabla \ell(\boldsymbol{\theta}_*; \boldsymbol{z}); \nabla \ell(\boldsymbol{\theta}_*; \boldsymbol{z})), \tag{22}$$

$$\boldsymbol{K}_s := \mathrm{Cov}_s(\nabla \ell(\boldsymbol{\theta}_*; \boldsymbol{z}^s); \nabla \ell(\boldsymbol{\theta}_*; \boldsymbol{z}^s)), \tag{23}$$

*where $\mathrm{Cov}$, $\mathrm{Cov}_s$ denote the covariances, respectively, with respect to the original data (i.e., with respect to $\boldsymbol{z} \sim \mathbb{P}$), and with respect to the surrogate data (i.e., with respect to $\boldsymbol{z}^s \sim \mathbb{P}_s$). Further define the $d$-dimensional vector*

$$\boldsymbol{g}^s := \nabla R^s(\boldsymbol{\theta}_*) - \nabla R(\boldsymbol{\theta}_*). \tag{24}$$

*Then there exists $\alpha_{\max} \in (0, 1]$ (depending only on the constants in the assumptions) such that, for all $\alpha \in [0, \alpha_{\max}]$, the excess risk of the estimator $\hat{\boldsymbol{\theta}}_{n,m}(\alpha)$ satisfies (for $D := \|\boldsymbol{g}^s\|$ bounded by a constant)*

$$R(\hat{\boldsymbol{\theta}}_{n,m}(\alpha)) - R(\boldsymbol{\theta}_*) = \alpha^2 \langle \boldsymbol{g}^s, \boldsymbol{H}^{-1} \boldsymbol{g}^s \rangle + \frac{(1-\alpha)^2}{n} \cdot \mathrm{Tr}(\boldsymbol{H}^{-1} \boldsymbol{K}) \tag{25}$$

$$+ \frac{\alpha^2}{m} \cdot \mathrm{Tr}(\boldsymbol{H}^{-1} \boldsymbol{K}_s) + O\left(\left(\frac{1}{m \vee n} + D\alpha^2\right)\left(\frac{1}{(m \vee n)^{1/2}} + D\alpha\right)\right).$$

*(Here the big O hides dependence on the constants in Assumptions 1 and 2.)*

**Remark B.1.** For economy of notation we stated Proposition B.1 in the case in which the excess risk is measured by using the same loss as for training, i.e. $\ell_{\text{test}} = \ell$. However the same result Eq. (25) applies with minor modifications to the case $\ell_{\text{test}} \neq \ell$ (and thus, with $R$ replaced by $R^{\text{test}}$), provided $R^{\text{test}}$ is also twice differentiable with Lipschitz Hessian, and $\nabla R^{\text{test}}(\boldsymbol{\theta}_*) = \mathbf{0}$. In this case, (25) has to be modified replacing $\boldsymbol{H}^{-1}$ by $\boldsymbol{H}^{-1} \nabla^2 R^{\text{test}}(\boldsymbol{\theta}_*) \boldsymbol{H}^{-1}$.

**Remark B.2.** The error terms in Eq. (25) are negligible under two conditions: $(i)$ $m$ and $n$ are large, which is the classical condition for low-dimensional asymptotics to hold; $(ii)$ $\|\boldsymbol{g}^s\|_2 = \|\nabla R^s(\boldsymbol{\theta}_*)\|_2 \alpha$ is small. In particular, the latter condition will hold in two cases. *First,* when $\|\nabla R^s(\boldsymbol{\theta}_*)\|_2$ is of order one (i.e. the distribution shift is large), but $\alpha$ is small (surrogate data are downweighted). Note that, when the distribution shift is large, and the sample size $n$ is large enough, we expect small $\alpha$ to be optimal and therefore Eq. (25) covers the 'interesting' regime.

*Second,* when $\|\nabla R^s(\boldsymbol{\theta}_*)\|_2$ is small (i.e. the shift is small) and $\alpha$ is of order one. If in addition we have $\nabla^2 R^s(\boldsymbol{\theta}_*) \approx \nabla^2 R^s(\boldsymbol{\theta}_*)$, it can be shown that the range of validity of Eq. (25) covers the whole interval $\alpha \in [0, 1]$.

**Remark B.3.** Note that the distribution shift is measured in Eq. (25) by the first term $\langle \boldsymbol{g}^s, \boldsymbol{H}^{-1} \boldsymbol{g}^s \rangle$. The original and surrogate distribution can be very different in other metrics (e.g. in total variation or transportation distance), but as long as $\boldsymbol{g}^s$ is small (as measured in the norm defined by $\boldsymbol{H}^{-1}$), surrogate data will reduce test error.

Note that, within the setting of Proposition B.1, the excess error of training only on original data is $R_{\text{or}}^{\text{ex}}(n) := R(\hat{\boldsymbol{\theta}}_{n,0}(0)) - R(\boldsymbol{\theta}_*) = \mathrm{Tr}(\boldsymbol{H}^{-1} \boldsymbol{K})/n + o(1/n)$, while $R_{\text{su}}^{\text{ex}}(m) := R(\hat{\boldsymbol{\theta}}_{n,m}(0)) - R(\boldsymbol{\theta}_*) = \langle \boldsymbol{g}^s, \boldsymbol{H}^{-1} \boldsymbol{g}^s \rangle + \mathrm{Tr}(\boldsymbol{H}^{-1} \boldsymbol{K}_s)/m + o(1/m)$. Hence Eq. (B.1) can be recast in the form of our general scaling law (4), namely:

$$R(\hat{\boldsymbol{\theta}}_{n,m}(\alpha)) - R(\boldsymbol{\theta}_*) \approx \alpha^2 R_{\text{su}}^{\text{ex}}(\infty) + \left[\alpha^2(R_{\text{su}}^{\text{ex}}(m) - R_{\text{su}}^{\text{ex}}(\infty)) + (1-\alpha)^2 R_{\text{or}}^{\text{ex}}(n)\right],$$

which (as expected) corresponds to the parametric scaling exponent $\beta = 1$.

An immediate consequence of Proposition B.1 is that surrogate data do not hurt, and will help if their distribution is close enough to the original one (under the assumption of optimally chosen $\alpha$).

**Corollary B.2.** *Under the assumptions of Proposition B.1, let* $\overline{R}_{\mathrm{or}}(n) := \mathrm{Tr}\big(\boldsymbol{H}^{-1}\boldsymbol{K}\big)/n$, *and* $\overline{R}_{\mathrm{su}}(m) := \langle \boldsymbol{g}^s, \boldsymbol{H}^{-1}\boldsymbol{g}^s \rangle + \mathrm{Tr}\big(\boldsymbol{H}^{-1}\boldsymbol{K}_s\big)/m$. *For* $\alpha_{n,m}^* = \overline{R}_{\mathrm{or}}(n)/(\overline{R}_{\mathrm{su}}(m) + \overline{R}_{\mathrm{or}}(n))$, *we have*

$$R\big(\hat{\boldsymbol{\theta}}_{n,m}(\alpha_{n,m}^*)\big) - R_* = \big(\overline{R}_{\mathrm{or}}(n)^{-1} + \overline{R}_{\mathrm{su}}(m)^{-1}\big)^{-1} + \Delta_{n,m},$$

*with* $\Delta_{n,m}$ *of the same order as the error in Prop. B.1.*

## B.2 Proofs

**Lemma B.3.** *Under the assumptions of Proposition B.1 (Assumption 1 and Assumption 2) there exists* $\alpha_{\max} \in (0,1]$, *depending only on the constants appearing there such that the following holds:*

(*i*) *The function* $\boldsymbol{\theta} \mapsto R(\boldsymbol{\theta}; \alpha) := (1-\alpha)\,R(\boldsymbol{\theta}) + \alpha\, R^s(\boldsymbol{\theta})$ *has a unique minimizer* $\boldsymbol{\theta}_*(\alpha) \in \mathbb{R}^d$. *Further* $\boldsymbol{\theta}_*(\alpha) \in \mathsf{B}(\boldsymbol{\theta}_*, r)$, *and* $\boldsymbol{\theta}_*(\alpha) \to \boldsymbol{\theta}_*$ *as* $\alpha \downarrow 0$.

(*ii*) *We have* $\hat{\boldsymbol{\theta}}_{n,m}(\alpha) \to \boldsymbol{\theta}_*$ *in probability as* $n, m \to \infty$.

*Proof.* Fix $r_0 \in (0, r]$ By Assumption 2.(a), $\inf_{\boldsymbol{\theta} \notin \mathsf{B}(\boldsymbol{\theta}_*; r_0)} R(\boldsymbol{\theta}) > R(\boldsymbol{\theta}_*) + \delta_0$ for some constant $\delta_0$. Hence, using Assumption 1, for any $\boldsymbol{\theta} \notin \mathsf{B}(\boldsymbol{\theta}_*; r)$

$$\begin{aligned}
R(\boldsymbol{\theta}; \alpha) &\geq R(\boldsymbol{\theta}) - K_*\alpha\big[1 + R(\boldsymbol{\theta})\big] \\
&\geq (1 - K_*\alpha)R(\boldsymbol{\theta}) - K_*\alpha \\
&\geq (1 - K_*\alpha)(R(\boldsymbol{\theta}_*) + \delta_0) - K_*\alpha\,.
\end{aligned}$$

In the other hand $R(\boldsymbol{\theta}_*; \alpha) \leq (1 + K_*\alpha)R(\boldsymbol{\theta}_*) + K_*\alpha$, whence

$$\begin{aligned}
R(\boldsymbol{\theta}; \alpha) - R(\boldsymbol{\theta}_*; \alpha) &\geq (1 - K_*\alpha)\delta_0 - 2K_*\alpha R(\boldsymbol{\theta}_*) \\
&\quad - 2K_*\alpha,,
\end{aligned}$$

which is strictly positive for $\alpha < \alpha_{\max}(r_0) := \delta_0/(4K_*(1 + R(\boldsymbol{\theta}_*)))$. Hence the minimum must be achieved in $\mathsf{B}(\boldsymbol{\theta}_*; r_0)$ (note that since $R(\boldsymbol{\theta})$, $R_s(\boldsymbol{\theta})$ are lower semicontinuous, the minimum is achieved).

By Assumption 2.(d), for $r_0$ sufficiently small, $\boldsymbol{\theta} \mapsto \nabla R(\boldsymbol{\theta}; \alpha)$ is strictly convex in $\mathsf{B}(\boldsymbol{\theta}_*; r_0)$ and therefore the minimizer is unique. This proves point (*i*).

Point (*ii*) follows from a modification of Theorem 5.14 in [vdV00]. Namely, for a diverging sequence $\{(n(k), m(k)) : k \in \mathbb{N}\}$, we consider to $\widehat{R}_{*,k}(\boldsymbol{u}) := \widehat{R}_{n(k),m(k)}(c(\boldsymbol{u})\boldsymbol{u}; \alpha)$, where $c(\boldsymbol{u}) := (1 + \|\boldsymbol{u}\|^2)^{-1/2}$. This function is lower semicontinuous on the compact set $\mathsf{B}(\boldsymbol{0}; 1)$ and converges almost surely to its expectation for every fixed $\boldsymbol{u}$ in this set, and hence the argument of Theorem 5.14 [vdV00] applies here. □

*Proof of Proposition B.1.* By a modification of Theorem 5.39 in [vdV00] (here $\boldsymbol{\theta}_*(\alpha)$ is defined as in Lemma B.3)

$$\hat{\boldsymbol{\theta}}_{n,m}(\alpha) = \boldsymbol{\theta}_*(\alpha) + \frac{1-\alpha}{n}\boldsymbol{H}(\alpha)^{-1}\sum_{i=1}^{n}\big[\nabla\ell(\boldsymbol{\theta}_*(\alpha); \boldsymbol{z}_i) - \mathbb{E}\nabla\ell(\boldsymbol{\theta}; \boldsymbol{z})\big] \tag{26}$$

$$+ \frac{\alpha}{m}\boldsymbol{H}(\alpha)^{-1}\sum_{i=1}^{m}\big[\nabla\ell(\boldsymbol{\theta}_*(\alpha); \boldsymbol{z}_i^c) - \mathbb{E}_s\nabla\ell(\boldsymbol{\theta}; \boldsymbol{z})\big] + O_P(m^{-1} + n^{-1})\,, \tag{27}$$

where $\boldsymbol{H}(\alpha) := (1 - \alpha)\nabla^2 R(\boldsymbol{\theta}_*(\alpha)) + \alpha\nabla^2 R_s(\boldsymbol{\theta}_*(\alpha))$. Note that in the present setting the error is of order $m^{-1} + n^{-1}$ because we assume the Hessian to be Lipschitz continuous.

The population minimizer $\boldsymbol{\theta}_*(\alpha)$ solves

$$\begin{aligned}
\boldsymbol{0} &= \nabla R(\boldsymbol{\theta}_*(\alpha); \alpha) \\
&= \nabla R(\boldsymbol{\theta}_*; \alpha) + \nabla^2 R(\boldsymbol{\theta}_*; \alpha)(\boldsymbol{\theta}_*(\alpha) - \boldsymbol{\theta}_*) + \int_0^1 \big[\nabla^2 R(\boldsymbol{\theta}_t; \alpha) - \nabla^2 R(\boldsymbol{\theta}_*; \alpha)\big](\boldsymbol{\theta}_*(\alpha) - \boldsymbol{\theta}_*)\,\mathrm{d}t\,,
\end{aligned}$$

where $\boldsymbol{\theta}_t = t\,\boldsymbol{\theta}_*(\alpha) + (1-t)\,\boldsymbol{\theta}_*$. Denoting by $L_2$ the Lipschitz constant of the Hessian (in operator norm), and recalling that $\nabla R(\boldsymbol{\theta}_*) = \mathbf{0}$, we have

$$\nabla^2 R(\boldsymbol{\theta}_*;\alpha)(\boldsymbol{\theta}_*(\alpha) - \boldsymbol{\theta}_*) = -\alpha\nabla R_s(\boldsymbol{\theta}_*) + \boldsymbol{u}\,,$$
$$\|\boldsymbol{u}\|_2 \leq L_2\|\boldsymbol{\theta}_*(\alpha) - \boldsymbol{\theta}_*\|^2\,.$$

Recalling that, by Lemma B.3, $\boldsymbol{\theta}_*(\alpha) \to \boldsymbol{\theta}_*$ as $\alpha \to 0$, this implies

$$\boldsymbol{\theta}_*(\alpha) - \boldsymbol{\theta}_* = -\boldsymbol{H}^{-1}\nabla R_s(\boldsymbol{\theta}_*)\alpha + O\big(((\|\nabla R_s(\boldsymbol{\theta}_*)\|_2 \vee \|\nabla R_s(\boldsymbol{\theta}_*)\|_2^2)\alpha^2\big)\,. \tag{28}$$

Substituting in Eq. (26), we get

$$\hat{\boldsymbol{\theta}}_{n,m}(\alpha) - \boldsymbol{\theta}_* = -\,\boldsymbol{H}^{-1}\nabla R_s(\boldsymbol{\theta}_*)\alpha + \frac{1-\alpha}{n}\boldsymbol{H}(\alpha)^{-1}\sum_{i=1}^{n}\big[\nabla\ell(\boldsymbol{\theta}_*(\alpha);\boldsymbol{z}_i) - \mathbb{E}\nabla\ell(\boldsymbol{\theta};\boldsymbol{z})\big] \tag{29}$$

$$+ \frac{\alpha}{m}\boldsymbol{H}(\alpha)^{-1}\sum_{i=1}^{m}\big[\nabla\ell(\boldsymbol{\theta}_*(\alpha);\boldsymbol{z}_i^c) - \mathbb{E}_s\nabla\ell(\boldsymbol{\theta};\boldsymbol{z})\big] + \boldsymbol{\Delta}\,, \tag{30}$$

$$\|\boldsymbol{\Delta}\| \leq C\Big(\|\nabla R_s(\boldsymbol{\theta}_*)\|_2 \vee \|\nabla R_s(\boldsymbol{\theta}_*)\|_2^2\Big)\alpha^2 + \frac{C}{m \wedge n}\,. \tag{31}$$

The claim follows by substituting the above in

$$\mathbb{E}R(\hat{\boldsymbol{\theta}}_{n,m}(\alpha)) - R(\boldsymbol{\theta}) = \mathbb{E}\langle\hat{\boldsymbol{\theta}}_{n,m}(\alpha) - \boldsymbol{\theta}_*, \boldsymbol{H}(\hat{\boldsymbol{\theta}}_{n,m}(\alpha) - \boldsymbol{\theta}_*)\rangle + O\Big(\mathbb{E}\|\hat{\boldsymbol{\theta}}_{n,m}(\alpha) - \boldsymbol{\theta}_*\|^3\Big) \tag{32}$$

and using $\boldsymbol{H}(\alpha) = \boldsymbol{H} + O(\alpha)$. $\qquad\square$

## C  Gaussian sequence model: Proofs for Section 3.1

### C.1  General ridge regression

We define $\hat{\boldsymbol{\Sigma}} = \boldsymbol{X}^\mathsf{T}\boldsymbol{X}/n$, $\hat{\boldsymbol{\Sigma}}_s = \boldsymbol{X}_s^\mathsf{T}\boldsymbol{X}_s/m$, and $\hat{\boldsymbol{\Sigma}}_\alpha = (1-\alpha)\hat{\boldsymbol{\Sigma}} + \alpha\hat{\boldsymbol{\Sigma}}_s$. We then have

$$R_{n,m}(\alpha,\lambda) = B_{n,m}(\alpha,\lambda) + \frac{(1-\alpha)^2\sigma^2}{n}\cdot V_{n,m}(\alpha,\lambda) + \frac{\alpha^2\sigma_s^2}{n}\cdot V_{n,m}^s(\alpha,\lambda)\,, \tag{33}$$

$$B_{n,m}(\alpha,\lambda) := \Big\|\boldsymbol{\Sigma}^{1/2}(\boldsymbol{\Omega} + \hat{\boldsymbol{\Sigma}}_\alpha)^{-1}\big(\boldsymbol{\Omega}\boldsymbol{\theta}_* - \alpha\hat{\boldsymbol{\Sigma}}_s(\boldsymbol{\theta}_*^s - \boldsymbol{\theta}_*)\big)\Big\|^2\,, \tag{34}$$

$$V_{n,m}(\alpha,\lambda) := \mathrm{Tr}\Big((\boldsymbol{\Omega} + \hat{\boldsymbol{\Sigma}}_\alpha)^{-1}\hat{\boldsymbol{\Sigma}}(\boldsymbol{\Omega} + \hat{\boldsymbol{\Sigma}}_\alpha)^{-1}\boldsymbol{\Sigma}\Big)\,, \tag{35}$$

$$V_{n,m}^s(\alpha,\lambda) := \mathrm{Tr}\Big((\boldsymbol{\Omega} + \hat{\boldsymbol{\Sigma}}_\alpha)^{-1}\hat{\boldsymbol{\Sigma}}_s(\boldsymbol{\Omega} + \hat{\boldsymbol{\Sigma}}_\alpha)^{-1}\boldsymbol{\Sigma}\Big) \tag{36}$$

### C.2  Proof of Theorem 1

Without loss of generality, we can assume $\boldsymbol{\Omega} = \mathrm{diag}((\omega_k)_{k\geq 1})$ with $\omega_k$ non-decreasing. A simple calculation gives the following general expression for the test error:

$$R_{n,m}(\alpha,\lambda) = B_{n,m}(\alpha,\lambda) + s_{n,m}(\alpha)\cdot V_{n,m}(\alpha,\lambda)\,, \tag{37}$$

$$B_{n,m}(\alpha,\lambda) := \sum_{k=1}^{\infty}\Big(\frac{1}{1+\lambda\omega_k}\Big)^2\big[(\alpha + \lambda\omega_k)\theta_{*,k} - \alpha\theta_{*,k}^s\big]^2\,, \tag{38}$$

$$V_{n,m}(\alpha,\lambda) := \sum_{k=1}^{\infty}\Big(\frac{1}{1+\lambda\omega_k}\Big)^2\,, \tag{39}$$

$$s_{n,m}(\alpha) := (1-\alpha)^2\frac{\sigma^2}{n} + \alpha^2\frac{\sigma_s^2}{m}\,. \tag{40}$$

We define (with $k_1 = 0$ if the condition is never verified)

$$k_1 := \max\big\{k :\ \lambda\omega_k \leq 1\big\}\,. \tag{41}$$

Note that

$$0 < k \le k_1 \quad \Rightarrow \quad 0 < \lambda\omega_k \le 1 \,, \tag{42}$$

$$k_1 < k \quad \Rightarrow \quad 1 < \lambda\omega_k \,. \tag{43}$$

We now estimate various sums by breaking them by the value of $k$

$$B_{n,m} \le \sum_{k=1}^{k_1} \big[(\alpha + \lambda\omega_k)\theta_{*,k} - \alpha\theta^s_{*,k}\big]^2 + \sum_{k=k_1+1}^{\infty} \frac{1}{(\lambda\omega_k)^2}\big[(\alpha + \lambda\omega_k)\theta_{*,k} - \alpha\theta^s_{*,k}\big]^2$$

$$\le \sum_{k=1}^{k_1} \big[\alpha^2(\theta_{*,k} - \theta^s_{*,k})^2 + 2\alpha(\theta_{*,k} - \theta^s_{*,k})\lambda\omega_k\theta_{*,k} + (\lambda\omega_k)^2\theta^2_{*,k}\big]$$

$$+ \sum_{k=k_1+1}^{\infty} \Big[\frac{\alpha^2}{(\lambda\omega_k)^2}(\theta_{*,k} - \theta^s_{*,k})^2 - \frac{2\alpha}{\lambda\omega_k}(\theta_{*,k} - \theta^s_{*,k})\theta_{*,k} + \theta^2_{*,k}\Big]$$

$$\le \alpha^2\|\boldsymbol{\theta}_{*,\le k_1} - \boldsymbol{\theta}^s_{*,\le k_1}\|^2 + \frac{2\alpha}{\omega_{k_1}}\big|\langle\boldsymbol{\theta}_{*,\le k_1} - \boldsymbol{\theta}^s_{*,\le k_1}, \boldsymbol{\theta}_{*,\le k_1}\rangle_{\boldsymbol{\Omega}}\big| + \frac{1}{\omega^2_{k_1}}\|\boldsymbol{\theta}_{*,\le k_1}\|^2_{\boldsymbol{\Omega}^2}$$

$$+ \alpha^2\omega^2_{k_1+1}\|\boldsymbol{\theta}_{*,>k_1} - \boldsymbol{\theta}^s_{*,>k_1}\|^2_{\boldsymbol{\Omega}^{-2}} + 2\alpha\omega_{k_1+1}\big|\langle\boldsymbol{\theta}_{*,>k_1} - \boldsymbol{\theta}^s_{*,>k_1}, \boldsymbol{\theta}_{*,>k_1}\rangle_{\boldsymbol{\Omega}^{-1}}\big| + \|\boldsymbol{\theta}_{*,>k_1}\|^2 \,,$$

and

$$V_{n,m} \le k_1 + \sum_{k>k_1} \frac{\omega^2_{k_1+1}}{\omega^2_k} \le (k_1 + c_\#) \,,$$

since under the assumption $\omega_k \asymp k^\mu$, $\mu > 1/2$, we have $\sum_{k>k_1}(\omega_{k_1+1}/\omega_k)^2 \le c_\#$.

Recalling the definitions in the theorem, and letting

$$\delta_k := \max\Big(\omega_{k+1}\big|\langle\boldsymbol{\theta}_{*,>k} - \boldsymbol{\theta}^s_{*,>k}, \boldsymbol{\theta}_{*,>k}\rangle_{\boldsymbol{\Omega}^{-1}}\big|; \; \omega^2_{k+1}\|\boldsymbol{\theta}_{*,>k} - \boldsymbol{\theta}^s_{*,>k}\|^2_{\boldsymbol{\Omega}^{-2}}\Big) \,,$$

we have

$$B_{n,m} \le \alpha^2\|\boldsymbol{\theta}_* - \boldsymbol{\theta}^s_*\|^2 + \|\boldsymbol{\theta}_{*,>k_1}\|^2 + \frac{1}{\omega^2_{k_1}}\|\boldsymbol{\theta}_{*,\le k}\|^2_{\boldsymbol{\Omega}^2} + 3\delta_{k_1} + 2\Delta_{k_1} \,,$$

whence

$$R_{n,m}(\alpha, \lambda) \le \alpha^2\|\boldsymbol{\theta}_* - \boldsymbol{\theta}^s_*\|^2 + \|\boldsymbol{\theta}_{*,>k_1}\|^2 + \frac{1}{\omega^2_{k_1}}\|\boldsymbol{\theta}_{*,\le k_1}\|^2_{\boldsymbol{\Omega}^2} + (k_1 + c_\#) \cdot s_{n,m}(\alpha) + 3\delta_{k_1} + 2\Delta_{k_1}$$

$$= \alpha^2 R^{\text{ex}}_{\text{su}}(\infty) + \|\boldsymbol{\theta}_{*,>k_1}\|^2 + \frac{1}{\omega^2_{k_1}}\|\boldsymbol{\theta}_{*,\le k_1}\|^2_{\boldsymbol{\Omega}^2} + (k_1 + c_\#) \cdot s_{n,m}(\alpha) + 3\delta_{k_1} + 2\Delta_{k_1} \,.$$

Next we specialize to the case $\|\boldsymbol{\theta}_{*,>k}\|^2 \le C_\theta k^{-2\rho}$, $\omega_k \asymp k^\mu$ $\mu \ne \rho$. In this case we have $\omega^{-2}_k\|\boldsymbol{\theta}_{*,\le k}\|^2_{\boldsymbol{\Omega}^2} \le Ck^{-2(\mu\wedge\rho)}$, and therefore, by suitably adjusting the constant $C$

$$R_{n,m}(\alpha, \lambda) \le \alpha^2 R^{\text{ex}}_{\text{su}}(\infty) + Ck_1^{-2(\mu\wedge\rho)} + (k_1 + c_\#) \cdot s_{n,m}(\alpha) + 3\delta_{k_1} + 2\Delta_{k_1} \,.$$

We now bound $\delta_k$. By Cauchy-Schwarz and monotonicity of $\omega$,

$$\omega_{k+1}\big|\langle\boldsymbol{\theta}_{*,>k} - \boldsymbol{\theta}^s_{*,>k}, \boldsymbol{\theta}_{*,>k}\rangle_{\boldsymbol{\Omega}^{-1}}\big| \le \|\boldsymbol{\theta}_{*,>k} - \boldsymbol{\theta}^s_{*,>k}\|_2\|\boldsymbol{\theta}_{*,>k}\|_2 \le 2C_\theta k^{-2\rho} \,,$$

and further

$$\omega^2_{k+1}\|\boldsymbol{\theta}_{*,>k} - \boldsymbol{\theta}^s_{*,>k}\|^2_{\boldsymbol{\Omega}^{-2}} \le 2\|\boldsymbol{\theta}_{*,>k}\|^2 + 2\|\boldsymbol{\theta}^s_{*,>k}\|^2 \le 4C_\theta k^{-2\rho} \,. \tag{44}$$

Therefore,

$$R_{n,m}(\alpha, \lambda) \le \alpha^2 R^{\text{ex}}_{\text{su}}(\infty) + Ck_1^{-2(\mu\wedge\rho)} + (k_1 + c_\#) \cdot s_{n,m}(\alpha) + 2\Delta_{k_1} \,.$$

*Proof of claim* $(a)$. The stated assumption on $\Delta_k$ imply that (eventually adjusting the constant $C$):

$$R_{n,m}(\alpha, \lambda) \le \alpha^2 R^{\text{ex}}_{\text{su}}(\infty) + Ck_1^{-2(\mu\wedge\rho)} + (k_1 + c_\#) \cdot s_{n,m}(\alpha) \,.$$

We now select $\lambda_*(\alpha)$ so that $k_1 \asymp s_{n,m}(\alpha)^{-1+\beta}$ where $\beta = 2(\mu \wedge \rho)/(1 + 2(\mu \wedge \rho))$. (this is possible for all $n, m$ large enough under the assumption on $\omega_k$), to A straightforward calculation yields:

$$R_{n,m}(\alpha, \lambda_*(\alpha)) \leq \alpha^2 R_{\mathsf{su}}^{\mathsf{ex}}(\infty) + C \cdot s_{n,m}(\alpha)^\beta \,,$$

which proves claim $(a)$.

*Proof of Claim* $(b)$. We choose $\omega_k = k^\mu$, $\theta_{*,k} = k^{-\rho'-1/2}$, $\theta_{*,k}^s = \theta_{*,k} + a_k k^{-\rho-1/2}$, with $a_k \sim \mathsf{Unif}(\{-A, +A\})$. We will choose $A \leq 1$ a sufficiently small numerical constant. Note that, for $\mu > 2\rho + 1/2$

$$\Delta_k = k^{-\mu} \left| \sum_{\ell=1}^{k} a_\ell \ell^{\mu-2\rho-1} \right| \leq CAk^{-\mu+\varepsilon} \left| \sum_{\ell=1}^{k} \ell^{2\mu-4\rho-2} \right|^{1/2} \leq CAk^{-2\rho-1/2+\varepsilon'} \,,$$

where, for any $\varepsilon > 0$, the first inequality holds with probability at least $1/2$ for all $k > k_0(\varepsilon)$. We can therefore select the $a_\ell$, so that $\Delta_k \leq C'' A k^{-2\rho-\varepsilon}$ for some $C'' < \infty$.

Following the calculation at point $(a)$ decompose the bias term as

$$B_{n,m} = \sum_{k=1}^{\infty} \left( \frac{1}{1 + \lambda\omega_k} \right)^2 \left[ \alpha^2 (\theta_{*,k} - \theta_{*,k}^s)^2 + (\lambda\omega_k)^2 \theta_{*,k}^2 \right] + 2\alpha E_{n,m} \,,$$

$$E_{n,m} := \sum_{k=1}^{\infty} \left( \frac{1}{1 + \lambda\omega_k} \right)^2 (\theta_{*,k} - \theta_{*,k}^s) \lambda\omega_k \theta_{*,k} \,.$$

Note that $|E_{n,m}| \leq \delta_{k_1} + \Delta_{k_1} \leq CAk_1^{-2(\mu\wedge\rho)}$. Therefore

$$B_{n,m} - \alpha^2 \|\boldsymbol{\theta}_* - \boldsymbol{\theta}_*^s\|^2$$

$$\geq \sum_{k=1}^{\infty} \left( \frac{\lambda\omega_k}{1 + \lambda\omega_k} \right)^2 \theta_{*,k}^2 - \alpha^2 \sum_{k=1}^{\infty} \left[ 1 - \left( \frac{1}{1 + \lambda\omega_k} \right)^2 \right] (\theta_{*,k} - \theta_{*,k}^s)^2 - CAk_1^{-2(\mu\wedge\rho)}$$

$$\geq \frac{1}{4\omega_{k_1+1}^2} \|\boldsymbol{\theta}_{*,\leq k_1}\|_{\boldsymbol{\Omega}^2}^2 + \frac{1}{4} \|\boldsymbol{\theta}_{*,>k_1}\|^2 - \frac{A}{4\omega_{k_1+1}} \|\boldsymbol{\theta}_{*,\leq k_1}\|_{\boldsymbol{\Omega}}^2 - \frac{A}{4} \|\boldsymbol{\theta}_{*,>k_1}\|^2 - CAk_1^{-2(\mu\wedge\rho)}$$

$$\geq C k_1^{-2(\mu\wedge\rho)} \,.$$

By a similar calculation, we also obtain

$$V_{n,m} \geq C k_1 \,,$$

and therefore

$$R_{n,m}(\alpha, \lambda) \geq \alpha^2 R_{\mathsf{su}}^{\mathsf{ex}}(\infty) + C k_1^{-2(\mu\wedge\rho)} + C k_1 \cdot s_{n,m}(\alpha) \,.$$

The proof is completed by minimizing over $k_1$.

## D  Analysis of the nonparametric model: Proofs for Section 3.2

This appendix is devoted to proving Theorem 2. Recall that this is established within the white noise model of Eq. (14), which we copy here for the readers' convenience

$$\mathrm{d}Y = f_*(\boldsymbol{x})\,\mathrm{d}\boldsymbol{x} + \frac{\sigma}{\sqrt{n}}\mathrm{d}B(\boldsymbol{x}) \,, \tag{45}$$

The adaptation of the estimator (13) to this continuous setting is given explicitly below

$$\hat{f}_{n,m,\alpha} = \arg\min_f \left\{ (1-\alpha)\|Y - f\|_2^2 + \alpha\|Y_s - f\|_2^2 + \lambda\|f\|_{p,2}^2 \right\}. \tag{46}$$

The proof of Theorem 2 is based on a reduction to a suitable 'sequence model' via the Fourier transform, defined as

$$\theta(\boldsymbol{q}) := \int_{[0,1]^d} f(\boldsymbol{x})\,e^{-\iota\langle \boldsymbol{q}, \boldsymbol{x}\rangle}\,\mathrm{d}\boldsymbol{x} \,, \tag{47}$$

for $\boldsymbol{q} \in \mathcal{Q}_d := \{2\pi\boldsymbol{q} : \boldsymbol{q} \in \mathbb{Z}^d\}$, where $\iota = \sqrt{-1}$. The inverse Fourier transform is defined as

$$f(\boldsymbol{x}) = \frac{1}{(2\pi)^d} \sum_{\boldsymbol{q} \in \mathcal{Q}_d} \theta(\boldsymbol{q})\, e^{\iota\langle \boldsymbol{q}, \boldsymbol{x}\rangle} \,. \tag{48}$$

We let $\theta_*$, $\theta_{*,s}$, and $\hat{\theta}_{\lambda,p,n,m,\alpha}$ respectively denote the Fourier transform of $f_*$, $f_{*,s}$, and $\hat{f}_{\lambda,p,n,m,\alpha}$. The Fourier transforms of the observations are given by

$$\hat{Y}(\boldsymbol{q}) = \theta_*(\boldsymbol{q}) + \frac{\sigma}{\sqrt{n}}\, G(\boldsymbol{q})\,, \qquad \hat{Y}_s(\boldsymbol{q}) = \theta_{*,s}(\boldsymbol{q}) + \frac{\sigma_s}{\sqrt{m}}\, G_s(\boldsymbol{q})\,, \tag{49}$$

where $G(\boldsymbol{q})$ and $G_s(\boldsymbol{q})$ are i.i.d. standard Gaussian. It then follows that

$$\hat{\boldsymbol{\theta}}_{n,m}(\alpha) = \arg\min_{\boldsymbol{\theta}} \left\{ (1-\alpha)\|\hat{\boldsymbol{Y}} - \boldsymbol{\theta}\|_2^2 + \alpha\|\hat{\boldsymbol{Y}}_s - \boldsymbol{\theta}\|_2^2 + \lambda\|\boldsymbol{\theta}\|_{p,2}^2 \right\}\,. \tag{50}$$

where we abuse the notation to define

$$\|\boldsymbol{\theta}\|_{p,2}^2 := \sum_{\boldsymbol{q} \in \mathcal{Q}_d} c_{p,\boldsymbol{q}}\, |\theta(\boldsymbol{q})|^2 \,. \tag{51}$$

with $c_{p,\boldsymbol{q}} := 1 + \|\boldsymbol{q}\|^{2r}$. Minimizing (50) we get

$$\hat{\theta}_{n,m}(\boldsymbol{q};\alpha) = \frac{1}{1 + \lambda c_{p,\boldsymbol{q}}} \left[ (1-\alpha)\,\hat{Y}(\boldsymbol{q}) + \alpha\,\hat{Y}_s(\boldsymbol{q}) \right]\,. \tag{52}$$

Taking the inverse Fourier transform and plugging it into the excess risk formula we get

$$R(\hat{f}_{n,m,\alpha}) = \sum_{\boldsymbol{q} \in \mathcal{Q}_d} \frac{1}{(1 + \lambda c_{p,\boldsymbol{q}})^2} \left[ \alpha(\theta_{*,s} - \theta_*)(\boldsymbol{q}) \right.$$
$$\left. + \lambda c_{p,\boldsymbol{q}}\theta_*(\boldsymbol{q}) \right]^2 + V_{n,m} \sum_{\boldsymbol{q} \in \mathcal{Q}_d} \frac{1}{(1 + \lambda c_{p,\boldsymbol{q}})^2}\,, \tag{53}$$

where

$$V_{n,m} := (1-\alpha)^2 \frac{\sigma^2}{n} + \alpha^2 \frac{\sigma_s^2}{m}\,. \tag{54}$$

The convexity of $x \to x^2$ implies

$$(a+b)^2 = \left( \gamma\frac{a}{\gamma} + (1-\gamma)\frac{b}{1-\gamma} \right)^2 \leq \frac{a^2}{\gamma} + \frac{b^2}{1-\gamma} \tag{55}$$

for $\gamma \in (0,1)$ and therefore we can upper bound the first sum in (53) by taking $\gamma = 1/(1+\delta)$ for any $\delta > 0$, which yields

$$R(f_{n,m,\alpha}) \leq (1+\delta)\alpha^2\|\boldsymbol{\theta}_{*,s} - \boldsymbol{\theta}_*\|_2^2 + \frac{1+\delta}{\delta} \sum_{\boldsymbol{q} \in \mathcal{Q}_d} \left( \frac{\lambda c_{p,\boldsymbol{q}}}{1 + \lambda c_{p,\boldsymbol{q}}} \right)^2 |\theta_*(\boldsymbol{q})|^2 + V_{n,m} \sum_{\boldsymbol{q} \in \mathcal{Q}_d} \frac{1}{(1 + \lambda c_{p,\boldsymbol{q}})^2} \,. \tag{56}$$

## D.1 Proof of Theorem 2

We now upper bound the first sum above. We note that, defining $q_0$ via $\lambda c_r(q_0) = 1$ (with an abuse of notation $c_r(t) = 1 + t^{2r}$), whence $q_0 \geq (\lambda/2)^{-1/2r}$ for all $\lambda < 1$:

$$\sum_{\boldsymbol{q} \in \mathcal{Q}_d} \left( \frac{\lambda c_{p,\boldsymbol{q}}}{1 + \lambda c_{r,\boldsymbol{q}}} \right)^2 \cdot |\theta_*(\boldsymbol{q})|^2 \leq \sum_{\boldsymbol{q} \in \mathcal{Q}_d, \|\boldsymbol{q}\|_2 \leq q_0} \lambda^2 c_r(\boldsymbol{q})^2 |\theta_*(\boldsymbol{q})|^2 + \sum_{\boldsymbol{q} \in \mathcal{Q}_d, \|\boldsymbol{q}\|_2 > q_0} |\theta_*(\boldsymbol{q})|^2$$

$$\leq \lambda^2 \max_{\|\boldsymbol{q}\|_2 \leq q_0} \frac{c_r(\boldsymbol{q})^2}{c_s(\boldsymbol{q})} \sum_{\boldsymbol{q} \in \mathcal{Q}_d, \|\boldsymbol{q}\|_2 \leq q_0} c_s(\boldsymbol{q})|\theta_*(\boldsymbol{q})|^2 + \max_{\|\boldsymbol{q}\|_2 > q_0} \frac{1}{c_s(\boldsymbol{q})} \sum_{\boldsymbol{q} \in \mathcal{Q}_d, \|\boldsymbol{q}\|_2 > q_0} c_s(\boldsymbol{q})|\theta_*(\boldsymbol{q})|^2$$

$$\overset{(a)}{\leq} \lambda^2 \max_{\|\boldsymbol{q}\|_2 \leq q_0} \frac{c_r(\boldsymbol{q})^2}{c_s(\boldsymbol{q})} \max_{\|\boldsymbol{q}\|_2 > q_0} \frac{1}{c_s(\boldsymbol{q})}$$

$$\leq \lambda^2 \max\left(1, \frac{c_r(q_0)^2}{c_s(q_0)}\right) + \frac{1}{c_s(q_0)}$$

$$\leq C \max(\lambda^2, \lambda^{p/r}) + C\lambda^{p/r} \leq C\lambda^{2 \wedge (p/r)}\,,$$

where in $(a)$ we used the fact that $\|f_*\|_{2,p}^2 = \sum_{\boldsymbol{q}} c_s(\boldsymbol{q})|\theta_*(\boldsymbol{q})|$. Letting $C_i(d)$ be constants depending on $d$, we have

$$
\begin{aligned}
\sum_{\boldsymbol{q} \in \mathcal{Q}_d} \frac{1}{(1 + \lambda c_{r,\boldsymbol{q}})^2} &\leq C_1(d) \int_{\mathbb{R}^d} \frac{1}{(1 + \lambda c_{r,\boldsymbol{q}})^2} \, \mathrm{d}\boldsymbol{q} \\
&\leq C_1(d) \int_{\mathbb{R}^d} \frac{1}{(1 + \lambda \|\boldsymbol{q}\|^{2r}))^2} \, \mathrm{d}\boldsymbol{q} \\
&\leq C_2(d) \int_0^\infty \frac{t^{d-1}}{(1 + \lambda t^{2r})^2} \mathrm{d}t \\
&\leq C_2(d) \int_0^{\lambda^{-1/2r}} t^{d-1} \, \mathrm{d}t + C_2(d)\lambda^{-2} \int_{\lambda^{-1/2r}}^\infty t^{d-1-4r} \, \mathrm{d}t \, .
\end{aligned}
$$

For convergence we requite $r > d/4$, in which case

$$
\sum_{\boldsymbol{q} \in \mathcal{Q}_d} \frac{1}{(1 + \lambda c_{r,\boldsymbol{q}})^2} \leq C_4(d)\lambda^{-d/2r} \, . \tag{57}
$$

# E   Analysis of high-dimensional regression: Proofs for Section 3.4

## E.1   Auxiliary definition for Theorem 3

Our characterization is given in terms of a variational principle. For $\delta, \delta_s \in (0, \infty)$, define $\mathscr{R}(\cdot; \alpha) : \mathbb{R}_{\geq 0}^3 \to \mathbb{R}$ via

$$
\begin{aligned}
\mathscr{R}(\xi, \xi_\perp, \omega; \alpha) := &-\omega\sqrt{\rho^2 + \rho_s^2} + \rho\sqrt{\delta(\tau^2 + \sigma^2)} + \rho_s\sqrt{\delta_s(\tau_s^2 + \sigma_s^2)} \tag{58} \\
&- \frac{\delta\rho^2}{2(1-\alpha)} - \frac{\delta_s\rho_s^2}{2\alpha} + \frac{\lambda}{2}\left(\xi^2 + \xi_\perp^2 + \omega^2\right),
\end{aligned}
$$

where $\tau, \tau_s$ are defined by

$$
\tau^2 := (\xi - r)^2 + \xi_\perp^2 + \omega^2 \, , \tag{59}
$$
$$
\tau_s^2 := (\xi - r_s \cos\gamma)^2 + (\xi_\perp - r_s \sin\gamma)^2 + \omega^2 \, , \tag{60}
$$

and $\rho = \overline{\rho}/\sqrt{1 + t^2}$, $\rho_s = \overline{\rho}t/\sqrt{1 + t^2}$, with $\overline{\rho}$ solving the polynomial equation

$$
\overline{\rho}^2 = \frac{\delta(\tau^2 + \sigma^2)}{\left(\delta/(1-\alpha) + \omega/\overline{\rho}\right)^2} + \frac{\delta_s(\tau_s^2 + \sigma_s^2)}{\left(\delta_s/\alpha + \omega/\overline{\rho}\right)^2} \, , \tag{61}
$$

and $t$ is given by

$$
t = \frac{\omega + \delta\overline{\rho}/(1-\alpha)}{\omega + \delta_s\overline{\rho}/\alpha} \cdot \sqrt{\frac{\delta_s(\tau_s^2 + \sigma_s^2)}{\delta(\tau^2 + \sigma^2)}} \, . \tag{62}
$$

Theorem 3 states that the asymptotics of the test error is determined by the minimizer of $\mathscr{R}$.

## E.2   Proof of Theorem 3

The proof is based on Gordon Gaussian comparison inequality [Gor85, Ver18], and follow a standard route, see e.g. [TOH15, TAH18, MM21]. We will limit ourselves to outlining the main steps of the calculation. Throughout, we consider the case $\varepsilon_0 > 0$, $\delta + \delta_s > 1$ because the other one ($\varepsilon_0 = 0$ and $\delta, \delta_s > 1$) is analogous and less interesting.

We begin by rewriting the ridge cost function in terms of a Lagrangian

$$
\widehat{R}_{n,m}(\boldsymbol{\theta}; \alpha) = \max_{\boldsymbol{u} \in \mathbb{R}^n} \max_{\boldsymbol{u}^s \in \mathbb{R}^m} \widehat{L}_{n,m}(\boldsymbol{\theta}, \boldsymbol{u}, \boldsymbol{u}^s; \alpha) \, , \tag{63}
$$

$$
\begin{aligned}
\widehat{L}_{n,m}(\boldsymbol{\theta}, \boldsymbol{u}, \boldsymbol{u}^s; \alpha) :=& \langle \boldsymbol{u}, \boldsymbol{X}(\boldsymbol{\theta} - \boldsymbol{\theta}_*)\rangle + \langle \boldsymbol{u}^s, \boldsymbol{X}^s(\boldsymbol{\theta} - \boldsymbol{\theta}_{*,s})\rangle - \langle \boldsymbol{u}, \boldsymbol{\varepsilon}\rangle - \langle \boldsymbol{u}^s, \boldsymbol{\varepsilon}^s\rangle \tag{64} \\
&- \frac{n\|\boldsymbol{u}\|_2^2}{2(1-\alpha)} - \frac{m\|\boldsymbol{u}^s\|_2^2}{2\alpha} + \frac{\lambda}{2}\|\boldsymbol{\theta}\|_2^2 \, .
\end{aligned}
$$

Let $\Delta(\boldsymbol{\theta}, \boldsymbol{u}, \boldsymbol{u}^s) := \|\boldsymbol{u}\|_2 \|\boldsymbol{\theta} - \boldsymbol{\theta}_*\|_2 G + \|\boldsymbol{u}^s\|_2 \|\boldsymbol{\theta} - \boldsymbol{\theta}_{*,s}\|_2 G_s.$, where $G, G_s$ are independent standard normal random variables, independent of $\boldsymbol{X}, \boldsymbol{X}^s$. By Gordon's inequality [Gor85], we can compare the Gaussian process $\widehat{L}_{n,m}(\boldsymbol{\theta}, \boldsymbol{u}, \boldsymbol{u}^s; \alpha) + \Delta(\boldsymbol{\theta}, \boldsymbol{u}, \boldsymbol{u}^s)$ to

$$\widehat{L}^G_{n,m}(\boldsymbol{\theta}, \boldsymbol{u}, \boldsymbol{u}^s; \alpha) := \|\boldsymbol{u}\| \langle \boldsymbol{g}, \boldsymbol{\theta} - \boldsymbol{\theta}_* \rangle + \|\boldsymbol{\theta} - \boldsymbol{\theta}_*\| \langle \boldsymbol{h}, \boldsymbol{u} \rangle + \|\boldsymbol{u}^s\| \langle \boldsymbol{g}^s, \boldsymbol{\theta} - \boldsymbol{\theta}_{*,s} \rangle + \|\boldsymbol{\theta} - \boldsymbol{\theta}_{*,s}\| \langle \boldsymbol{h}, \boldsymbol{u}^s \rangle \tag{65}$$

$$- \langle \boldsymbol{u}, \boldsymbol{\varepsilon} \rangle - \langle \boldsymbol{u}^s, \boldsymbol{\varepsilon}^s \rangle - \frac{n\|\boldsymbol{u}\|_2^2}{2(1-\alpha)} - \frac{m\|\boldsymbol{u}^s\|_2^2}{2\alpha} + \frac{\lambda}{2} \|\boldsymbol{\theta}\|_2^2 \,.$$

Next we define the orthonormal vectors

$$\boldsymbol{v}_* := \frac{\boldsymbol{\theta}_*}{\|\boldsymbol{\theta}_*\|_2}, \qquad \boldsymbol{v}_*^\perp := \frac{\boldsymbol{P}_{\boldsymbol{\theta}_*}^\perp \boldsymbol{\theta}_{*,s}}{\|\boldsymbol{P}_{\boldsymbol{\theta}_*}^\perp \boldsymbol{\theta}_{*,s}\|_2}, \tag{66}$$

where $\boldsymbol{P}_{\boldsymbol{\theta}_*}^\perp = \boldsymbol{I} - \boldsymbol{P}_{\boldsymbol{\theta}_*} := \boldsymbol{I} - \boldsymbol{v}_* \boldsymbol{v}_*^\mathsf{T}$ is the projector orthogonal to $\boldsymbol{\theta}_*$. We then decompose

$$\boldsymbol{\theta} = \xi \boldsymbol{v}_* + \xi_\perp \boldsymbol{v}_*^\perp + \boldsymbol{\theta}^\perp \,, \tag{67}$$

where $\langle \boldsymbol{v}_*, \boldsymbol{\theta}^\perp \rangle = \langle \boldsymbol{v}_*^\perp, \boldsymbol{\theta}^\perp \rangle = 0$, and define $\omega := \|\boldsymbol{\theta}^\perp\|_2$. Defining $\tau^2 = \|\boldsymbol{\theta} - \boldsymbol{\theta}_*\|_2^2$, $\tau_s^2 = \|\boldsymbol{\theta} - \boldsymbol{\theta}_{*,s}\|_2^2$, Eq. (60) follows.

With these notations, and letting $\hat{\sigma}^2 = \|\tau\boldsymbol{h} + \boldsymbol{\varepsilon}\|_2^2/n - \tau^2$, $\hat{\sigma}_s^2 = \|\tau_s \boldsymbol{h}^s + \boldsymbol{\varepsilon}^s\|_2^2/m - \tau_s^2$, we get

$$\widehat{\mathscr{L}}^G_{n,m}(\boldsymbol{\theta}, \rho, \rho_s; \alpha) := \max_{\boldsymbol{u}, \boldsymbol{u}^s} \left\{ \widehat{L}^G_{n,m}(\boldsymbol{\theta}, \boldsymbol{u}, \boldsymbol{u}^s; \alpha) : \|\boldsymbol{u}\| = \frac{\rho}{\sqrt{d}}, \|\boldsymbol{u}^s\| = \frac{\rho_s}{\sqrt{d}} \right\}, \tag{68}$$

$$\widehat{\mathscr{L}}^G_{n,m}(\boldsymbol{\theta}, \rho, \rho_s; \alpha) = \frac{\rho}{\sqrt{d}} \langle \boldsymbol{g}, \boldsymbol{\theta} - \boldsymbol{\theta}_* \rangle + \frac{\rho_s}{\sqrt{d}} \langle \boldsymbol{g}^s, \boldsymbol{\theta} - \boldsymbol{\theta}_{*,s} \rangle + \rho\sqrt{\delta(\tau^2 + \hat{\sigma}^2)} + \rho_s\sqrt{\delta_s(\tau_s^2 + \hat{\sigma}_s^2)} \tag{69}$$

$$- \frac{\delta\rho^2}{2(1-\alpha)} - \frac{\delta_s\rho_s}{2\alpha} + \frac{\lambda}{2}\left(\xi^2 + \xi_\perp^2 + \omega^2\right).$$

We finally decompose $\boldsymbol{g} = \boldsymbol{g}_\| + \boldsymbol{g}_\perp$ where $\boldsymbol{g}_\| \in \mathsf{span}(\boldsymbol{v}_*, \boldsymbol{v}_*^\perp)$ and $\boldsymbol{g}_\| \perp \mathsf{span}(\boldsymbol{v}_*, \boldsymbol{v}_*^\perp)$, and similarly for $\boldsymbol{g}_s$, and define

$$\mathscr{L}^G_{n,m}(\xi, \xi_\perp, \omega, \rho, \rho_s; \alpha) := \min_{\boldsymbol{\theta}} \left\{ \widehat{\mathscr{L}}^G_{n,m}(\boldsymbol{\theta}, \rho, \rho_s; \alpha) : \boldsymbol{\theta} = \xi \boldsymbol{v}_* + \xi_\perp \boldsymbol{v}_*^\perp + \boldsymbol{\theta}^\perp, \ \ \|\boldsymbol{\theta}^\perp\| = \omega \right\}. \tag{70}$$

Defining $\iota$ via $\|\rho\boldsymbol{g}_\perp/\sqrt{n} + \rho_s\boldsymbol{g}_{s,\perp}/\sqrt{m}\| = (1+\iota)\sqrt{\rho^2 + \rho_s^2}$, we obtain

$$\mathscr{L}^G_{n,m}(\xi, \xi_\perp, \omega, \rho, \rho_s; \alpha) = -(1+\iota)\sqrt{\rho^2 + \rho_s^2} \cdot \omega + \Delta + \rho\sqrt{\delta(\tau^2 + \hat{\sigma}^2)} + \rho_s\sqrt{\delta_s(\tau_s^2 + \hat{\sigma}_s^2)} \tag{71}$$

$$- \frac{\delta\rho^2}{2(1-\alpha)} - \frac{\delta_s\rho_s^2}{2\alpha} + \frac{\lambda}{2}\left(\xi^2 + \xi_\perp^2 + \omega^2\right),$$

where $\Delta$ is the contribution of the perpendicular components. Simple concentration estimates imply that for any $\varepsilon > 0$ there exist $c(\varepsilon) > 0$ such that

$$\mathbb{P}\big(|\hat{\sigma} - \sigma| \le \varepsilon\sqrt{\tau^2 + \sigma^2}, |\hat{\sigma}_s - \sigma_s| \le \varepsilon\sqrt{\tau_s^2 + \sigma_s^2}\big) \ge 1 - e^{-c(\varepsilon)n}, \tag{72}$$

$$\mathbb{P}\big(\Delta \le \sqrt{(\rho^2 + \rho_s^2)(\xi^2 + \xi_\perp^2)}\big) \ge 1 - e^{-c(\varepsilon)n}, \tag{73}$$

$$\mathbb{P}\big(|\iota| \le \varepsilon\big) \ge 1 - e^{-c(\varepsilon)n}. \tag{74}$$

We can then estimate $\mathscr{L}^G_{n,m}(\xi, \xi_\perp, \omega, \rho, \rho_s; \alpha)$ by

$$\mathscr{L}^G(\xi, \xi_\perp, \omega, \rho, \rho_s; \alpha) = -\sqrt{\rho^2 + \rho_s^2} \cdot \omega + \rho\sqrt{\delta(\tau^2 + \sigma^2)} + \rho_s\sqrt{\delta_s(\tau_s^2 + \sigma_s^2)} \tag{75}$$

$$- \frac{\delta\rho^2}{2(1-\alpha)} - \frac{\delta_s\rho_s^2}{2\alpha} + \frac{\lambda}{2}\left(\xi^2 + \xi_\perp^2 + \omega^2\right),$$

Differentiating with respect to $\rho$ and $\rho_s$ and setting the derivatives to 0 yields $\rho = \overline{\rho}/\sqrt{1+t^2}$, $\rho_s = \overline{\rho}t/\sqrt{1+t^2}$, with $\overline{\rho}, t$ given by Eqs. (61), (62). By computing second derivatives, one obtain that this is a local maximum. Since $\mathscr{L}^G(\xi, \xi_\perp, \omega, \rho, \rho_s; \alpha) \to -\infty$ as $\rho^2 + \rho_s^2 \to \infty$, the maximum over $\rho, \rho_s$ is either achieved at this point or at the boundary $\{\rho = 0\} \cup \{\rho_s = 0\}$. By checking the signs of partial derivatives along this boundary, the only other possibility is $\rho = \rho_s = 0$.

For economy of notation, write $F(\rho, \rho_s) := \mathscr{L}^G(\xi, \xi_\perp, \omega, \rho, \rho_s; \alpha)$. For any unit vector $\boldsymbol{v} = (v_1, v_2) \geq 0$, the directional derivative is

$$\nabla_{\boldsymbol{v}} F(\boldsymbol{r})\big|_{\boldsymbol{r}=0} = -\omega + v_1 \sqrt{\delta(\tau^2 + \sigma^2)} + v_2 \sqrt{\delta_s(\tau_s^2 + \sigma_s^2)}$$
$$\geq \omega\big[-1 + v_1\sqrt{\delta} + v_2\sqrt{\delta_s}\big].$$

By maximizing over the direction, we see that $\boldsymbol{v}$ can be chosen so that $\nabla_{\boldsymbol{v}} F(\boldsymbol{0}) \geq \omega[-1 + \sqrt{\delta + \delta_s}]$. Hence $\rho = \rho_s = 0$ cannot be the global aximum for $\delta + \delta_s > 1$.

Hence, we get

$$\mathscr{R}(\xi, \xi_\perp, \omega; \alpha) = \max_{\rho, \rho_s \geq 0} \mathscr{L}^G(\xi, \xi_\perp, \omega, \rho, \rho_s; \alpha). \tag{76}$$

We further note that, for fixed $\rho, \rho_s > 0$, the function $(\xi, \xi_\perp, \omega) \mapsto \mathscr{L}^G(\xi, \xi_\perp, \omega, \rho, \rho_s; \alpha)$ is jointly strictly convex for $\lambda > 0$. Hence $(\xi, \xi_\perp, \omega) \mapsto \mathscr{R}(\xi, \xi_\perp, \omega; \alpha)$ is also strictly convex for $\lambda > 0$. Therefore, it has a unique minimizer, which we denote by $(\xi^*, \xi_\perp^*, \omega^*)$. Proceeding as in [MM21], we obtain the following result.

**Proposition E.1.** *Under the assumptions of Proposition 3, for any $\varepsilon, \varepsilon_0 > 0$ there exists $c = c(\varepsilon, \varepsilon_0) > 0$ such that, if $\alpha \in [\varepsilon_0, 1 - \varepsilon_0]$ (letting $\boldsymbol{P}^\perp := \boldsymbol{I} - \boldsymbol{v}_* \boldsymbol{v}_*^\mathsf{T} - \boldsymbol{v}_*^\perp(\boldsymbol{v}_*^\perp)^\mathsf{T}$)*

$$\mathbb{P}\Big\{\big|\langle \boldsymbol{v}_*, \hat{\boldsymbol{\theta}}_{n,m}\rangle - \xi^*\big| \leq \varepsilon,\ \big|\langle \boldsymbol{v}_*^\perp, \hat{\boldsymbol{\theta}}_{n,m}\rangle - \xi_\perp^*\big| \leq \varepsilon,\ ,\big|\|\boldsymbol{P}^\perp \hat{\boldsymbol{\theta}}_{n,m}\| - \omega^*\big| \leq \varepsilon\Big\} \geq 1 - 2\,e^{-cn}. \tag{77}$$

In particular, the last proposition implies (a weaker form of) Theorem 3 whereby the supremum is taken over a finite net. Namely for $\eta > 0$, we define

$$N(\varepsilon_0, \eta) := [\varepsilon_0, 1 - \varepsilon_0] \cap \eta\mathbb{Z}.$$

Recalling that, in the present case, $R(\hat{\boldsymbol{\theta}}) = \|\hat{\boldsymbol{\theta}} - \boldsymbol{\theta}\|_2^2$, we obtain (after adjusting the constant $c$) we have therefore:

$$\mathbb{P}\Big(\max_{\alpha \in N(\varepsilon_0, \eta)} \big|R(\hat{\boldsymbol{\theta}}_{n,m}(\alpha)) - \mathscr{R}_{\text{test}}(\alpha)\big| \geq \varepsilon\Big) \geq 1 - 2\,e^{-cn}. \tag{78}$$

Finally, let $\boldsymbol{X}_+ \in \mathbb{R}^{(m+n)\times d}$ be the matrix obtained by stacking $\boldsymbol{X}$ and $\boldsymbol{X}_s$. Given constants $C_1, C_2, C_3$, define the good event

$$\mathcal{G} := \Big\{C_1 n \leq \lambda_{\min}(\boldsymbol{X}_+^\mathsf{T}\boldsymbol{X}_+) \leq \lambda_{\max}(\boldsymbol{X}_+^\mathsf{T}\boldsymbol{X}_+) \leq C_2 n; \|\boldsymbol{X}^\mathsf{T}\boldsymbol{y}\| \leq C_3 n,\ \|\boldsymbol{X}_s^\mathsf{T}\boldsymbol{y}_s\| \leq C_3 n\Big\} / \tag{79}$$

By a standard bound on eigenvalues of Wishart matrices [Ver18], for $\delta + \delta_s > 1$, we can choose $C_1, C_2, C_3$ such that

$$\mathbb{P}(\mathcal{G}) \geq 1 - 2e^{-cn}. \tag{80}$$

Further on $\mathcal{G}$, $\boldsymbol{\theta}_{n,m}(\alpha)$ is bounded (in $\ell_2$ norm, and Lipschitz continuous in $\alpha$). As a consequence, for a sufficiently large constant $L$,

$$\mathbb{P}\Big(\big|R(\hat{\boldsymbol{\theta}}_{n,m}(\alpha_1)) - R(\hat{\boldsymbol{\theta}}_{n,m}(\alpha_2))\big| \leq L|\alpha_1 - \alpha_2| \forall \alpha_1, \alpha_2 \in [\varepsilon_0, 1 - \varepsilon_0]\Big) \geq 1 - 2e^{-cn}. \tag{81}$$

The claim follows by using this estimate together with Eq. (78).

# F    Datasets information

- Imdb reviews datatset:
    - Paper:  [MDP$^+$11]
    - Link
    - Licence: Creative Commons Attribution-NonCommercial-ShareAlike 3.0 International License
- Rotten Tomatoes reviews:
    - Paper: [PL05]
    - Link
    - Data has been scraped from the publicly available website https://www.rottentomatoes.com as of 2020-10-31.
    - Licence: CC0: Public Domain
- Goodreads bookreviews
    - Papers: [WMNM19], [WM18]
    - Link
    - License: Unknown
- CIFAR10 and CIFAR100:
    - Paper: [Kri09]
    - Link
    - License: Unknown
- TCGA Pan-Cancer Clinical Data
    - Link
    - Publicaly availbale, free to use
    - Licence: None

# G    Compute resources information

We ran all experiments on a single machine with 2 RTX 4090 GPUs and a 24-core Intel Xeon E5 CPU. All experiments completed in less than 24 hours.

