# OpenReview forum: "Scaling laws for learning with real and surrogate data"
_NeurIPS.cc/2024/Conference — NeurIPS 2024 poster_

### Official Review · Reviewer_yPKz · 2024-06-29

**Soundness:** 3
**Presentation:** 2
**Contribution:** 2
**Rating:** 6
**Confidence:** 3

**Summary:**

This work addresses the challenge of augmenting limited data with more accessible surrogate data to improve generalization. The authors proposed a weighted ERM approach for integrating surrogate data into training and analyzed its performance under various statistical models. It was shown that incorporating surrogate data provably reduces test error, even when the surrogate data is far from the original ones. The paper introduced a scaling law that predicts the optimal weighting scheme and the amount of surrogate data to use, supported by both theories and experiments across different domains.

**Strengths:**

- The effect of weighted ERM on generalization is theoretically analyzed on various statistical models.
- The discussion on the connection with the Stein paradox through the toy Gaussian mean estimation example is insightful and provides a nice intuition for why incorporating surrogate data can be beneficial.
- The theoretical results are well-supported by experiments in different domains.

**Weaknesses:**

Weighted ERM is a simple and arguably the most intuitive way to incorporate surrogate data. This is a strength rather than a weakness on its own. However, given the ubiquity of such methods, the discussion on related works could be more comprehensive.
- For example, in addition to synthetic data from generative models, data augmentations also seem to fall into the category of surrogate data considered in the paper. The results on sample complexities of data augmentation methods could be relevant.
- Another potentially related topic is distributionally robust optimization. Despite the different optimization problems, the idea of finding a near-optimal weight for different groups of samples shares a similar high-level goal.
- Instead of "introducing weighted ERM", the contribution of this work seems to be more about providing a theoretical underpinning for the effectiveness of weighted ERM with an insight on the choice of weight. This could be made clearer in the paper.

**Questions:**

- In Eq (4), is there any reason for using $\approx$ instead of $\asymp$?
- In Theorem 1 and 2, how to interpret $\beta$ intuitively, the difficulty of the learning problem?
- In the Sobolev class example (line 161), what's the particular reason for assuming $n=Q^d$? Is this essential or just for simplicity?

**Limitations:**

- Some limitations are listed in the Weaknesses section.
- It would be helpful to provide more intuitions for Theorem 3 given the complexity of the notations and the distinct form of the result in the asymptotic regime (cf. the clear sample complexities in Theorems 1 and 2). Remark 3.3 on the connection with the scaling law provides some insights but doesn't seem to be clear enough.

---

> ### Author Rebuttal · Authors · 2024-08-05
>
> Essentially all weaknesses pointed by the referee concern literature review.
> We will expand the comparison with literature.
>
> Regarding the specific suggestions provided:
> 1. Data augmentation. We feel this is a significantly different setting. In data augmentation, the new data is typically obtained by perturbations/transformations of the original one. As such, the dependence between original and surrogate data need to be carefully modeled.
> 2. Distributionally robust optimization. This is indeed an interesting connection that is worth exploring.
> 3. We agree that  “introducing weighted ERM” is not accurate. We will replace it with “investigating weighted ERM” or similar.
> 4. We agree that the content of Theorem 3 is not easy to parse. We will complement it with some simplified/approximate expressions that are easier to parse and interpret.
>
> Questions:
>
> Q1: The symbol $\asymp$ indicates equivalence up to (possibly large) multiplicative constants. In several of our mathematical examples (and in simulations) constants seem to be close to 1 (possibly with some small additive error). We use $\approx$ to indicate that we interpret the scaling law in this stronger sense.
>
> Q2: $\beta$ is a scaling exponent that quantifies how the test error scales with the sample size. Scaling laws have been widely popular among AI practitioners and in those cases this exponent is fitted to empirical data. In these examples we compute the scaling exponent explicitly.
>
> Q3: Indeed, we assume $n=Q^d$ for mathematical simplicity. It is standard to assume such “regular designs” in nonparametric regression because this simplifies the analysis, without changing the final result.

---

> > ### Comment · Reviewer_yPKz · 2024-08-10
> >
> > Many thanks for the clarifications. My concerns are addressed. Conditioned on a better presentation and a more comprehensive literature review, I will raise my score and vote for acceptance.

---

> > > ### Author Response · Authors · 2024-08-12
> > >
> > > Thank you, we will do our best to improve the presentation and comparison with earlier work, as recommended.

---

### Official Review · Reviewer_3jgb · 2024-07-10

**Soundness:** 2
**Presentation:** 3
**Contribution:** 3
**Rating:** 4
**Confidence:** 4

**Summary:**

This work investigates the effects of augmenting training datasets with lower quality data, under a weighted ERM scheme. It is shown that introducing this surrogate data to optimally weighted training can improve predictive performance on the original data distribution, as measured by test error, even when the surrogate data is unrelated to the dataset of interest. A corresponding scaling law is derived, giving insight into both the optimal weighting and the amount of surrogate data that can be used. Numerical examples are provided.

**Strengths:**

The paper is generally well written.
A substantial numerical study is provided.

**Weaknesses:**

The theory of this paper is developed for a squared loss function. However, most of the numerics are done on classification tasks without mention of this discrepancy. This should be addressed, with justification provided if possible.
The role of the regularizer is barely mentioned, however it appears to play a crucial role in the relatively clean form of the relevant quantities given in the result statements. Analysis of sensitivity of the results to the regularization parameters would seem important and be appreciated. Further, growth estimates on its spectrum do not match the usual ridge regression setup (i.e. setting $\Omega$ to be the identity) in a way that's independent of parameter dimension. This makes the assumption highly artificial, and somewhat redundant (compared to just using the largest eigenvalue), and impacts the statement of the theorems.
Clear discussion on the rationale for a scaling law and its specific form is not given.

**Questions:**

Can you please address the above weaknesses?

**Limitations:**

While the phenomena is framed as analogous to a regularization technique, I do not feel that adequate care is taken to ensure that non-expert readers are not encouraged to add junk data to their training regimes without adequate care being taken.

---

> ### Author Rebuttal · Authors · 2024-08-05
>
> We agree that it would be important to generalize our theory to classification loss. At the same time, a substantial number of results suggest that the two settings are not as different in high dimension. Among others
> 1. Vidya Muthukumar, Adhyyan Narang, Vignesh Subramanian, Mikhail Belkin, Daniel Hsu, and Anant Sahai, Classification vs regression in overparameterized regimes: Does the loss function matter?, Journal of Machine Learning Research 22 (2021),
> 2. Montanari, A., Ruan, F., Sohn, Y. and Yan, J., 2019. The generalization error of max-margin linear classifiers: Benign overfitting and high dimensional asymptotics in the overparametrized regime.
>
> These papers show that the risk of max margin classification is either identical (paper 1) or very similar (paper 2) to that of least square regression in high dimension.
> We also would like to point out that some of our simulations are for square loss and some for classification loss. The results are qualitatively very similar.
>
> We also agree that investigating the sensitivity to the regularization value is an important research question. At the same time, we would like to point out that:
> 1. Theorems 1 and 2 keep holding (up to constants)  if the regularization parameter is changed by a multiplicative constant.
> 2. Theorem 3 captures the entire dependence on the regularization parameter. We did not explore this dependence in the plots presented in the manuscript, but we can do it  straightforwardly (it is just a matter of changing a parameter in a function that evaluate the formula).
>
> The setting in Section 3.1 is really motivated by nonparametric regression and other nonparametric settings (eg  the Sobolev regression problem of Section 3.2). In these cases, the theta_k are Fourier or wavelet coefficients and $\Omega$ encodes the prior smoothness information about the signal. For instance, cubic smoothing splines correspond to the case $\Omega_k$ proportional to $k^4$.
>
> The reviewer mentions that the paper does not take “adequate care to ensure that non-expert readers” do not “add junk data to their training”.
> We would like to emphasize that, to the contrary, this is exactly the problem that we address in our paper.  If the weight \alpha is optimized through a consistent procedure (eg cross-validation), then the resulting learning procedure is very robust to the use of bad surrogate data.

---

> > ### Comment · Reviewer_3jgb · 2024-08-11
> > **Response**
> >
> > I thank the authors for their clarification. I do realize that these losses behave similarly, however in a paper such as this I think it should be addressed when presenting theory and should be explicitly mentioned in the paper. The authors have briefly commented regarding the regularization parameter, setting of Theorem 3.1, and my concerns about partitioners blindly using dummy data. However, they have not provided any indication on whether they intend to address these concerns in their paper, and if so, detailed discussion on how the changes to the document, its framing, and experiments will be made. Without this, I cannot adjust my score.

---

> > > ### Author Response · Authors · 2024-08-12
> > >
> > > Thanks for the comment. Concretely, we plan the following changes.
> > > To address the generalization to other loss functions:
> > > 1. We will add a discussion of classification loss, referring to the papers mentioned in our rebuttal. We will draw explicit consequence of these results for the surrogate data setting.
> > > 2. We will plot an empirical comparison of classification and regression losses. For this plot, we will consider one example with binary labels, and fit either cross-entropy loss or least square loss, plotting the resulting test error versus alpha in the two cases.
> > > 3. We have mathematical results on classification loss for under-parametrized settings in which the sample size is much larger than the number of parameters. These are currently given in Appendix B. We will instantiate these general results for classification, and state these consequences in the main text.
> > >
> > > To address readers blindly adding bad surrogate data:
> > > 1. We will add a new subsection to Section 1, to discuss the practical use of the scaling law. (This will expand the paragraph beginning on line 93.)
> > > 2. In this subsection we will outline an explicit flowchart of how practitioners can use the proposed weighting scheme and scaling law.
> > > 3. We will add a figure demonstrating the evolution of test error with added surrogate data using our approach, and a naive approach (comparing for “good” and “bad” surrogate data). This figure should clarify that if ‘bad’ surrogate data are added to the training in a naive way, it can dramatically hurt the model test error. On the other hand, if the same data is added with weight that is optimized via cross-validation, then this effect is significantly mitigated and potentially reversed (“bad” data can help).
> > > 4. The same figure will be used to illustrate the robustness of the method with respect to changes in the regularization parameter.
> > >
> > > Finally, we will add motivations for the setting of Theorem 3.1 by connecting it to classical nonparametric regression models.

---

> > > > ### Comment · Reviewer_3jgb · 2024-08-14
> > > > **Response**
> > > >
> > > > Thanks for your response. I have raised my score as a result of these proposed changes.

---

> ### Comment · Area_Chair_wp1A · 2024-08-11
> **discussion**
>
> Dear Reviewer 3jgb,
>
> Thank you very much for submitting your review report. The author(s) have posted responses to your review. Could you kindly provide comments on whether your concerns have been adequately addressed?
>
> AC

---

### Official Review · Reviewer_Fovm · 2024-07-10

**Soundness:** 4
**Presentation:** 4
**Contribution:** 4
**Rating:** 9
**Confidence:** 3

**Summary:**

This study addresses the challenge of computational scenarios where true data are scarce, and surrogate data are available to assist in building statistical models. The authors provide novel theoretical and empirical insights demonstrating that training models with both true and surrogate data, with appropriated selected coefficient $\alpha$, could improve empirical risk reduction on test sets, even when the surrogate data is unrelated.

**Strengths:**

- The scaling law claim is clearly presented and supported by theoretical results across various contexts, including Gaussian sequence model, non-parametric regression, low-dimensional asymptotic, and high-dimensional linear regression. This breadth of application suggests the claim's novelty and relevance.

- The link established between training with surrogate data and the Stein paradox is both novel and intriguing, potentially influencing future work in data augmentation, multi-fidelity modeling, and out-of-distribution training.

- Theoretical and empirical results are robust and well-articulated, lending significant credibility and depth to the findings.

**Weaknesses:**

- The discussion is limited to certain loss functions, specifically $\lVert \theta_\ast - \theta\rVert^2$ and $\lVert f_\ast - f\rVert^2_{L^2}$. Extending these results to include more common loss functions like L-1/L-inf norms or classification-driven losses could broaden the applicability of the findings.

- Some notations are unclear or undefined, which could hinder understanding for readers unfamiliar with the symbols or concepts. For example, the symbol $\asymp$ used in Line 152 and $\Lambda(\alpha)$ in Line 207 require definitions.

- Some minor errors:
    - Line 50 introduces the test error $R_\text{test}$ but this subscript is not used in the following context, e.g. Equation (4). Please make sure the test error notations are consistent;
    - Line 226: $R_\ast=R(\hat\theta_{0, \infty}(0))$ should be $R_\ast=R(\hat\theta_{\infty, 0}(0))$

**Questions:**

- Equation (4) appears to lack a variable that quantifies the "closeness" between the surrogate and true data, such as the $\gamma$ parameter defined in Theorem 3. Could the authors clarify the absence of such a metric in this model formulation?

Note: I realize that $R_\text{su}^\text{ex}(\infty)$ can represent the deviation from surrogate data to true data (indirectly). But let's keep this in case others share similar questions.

**Limitations:**

The limitation is included and no negative societal impact should be considered.

---

> ### Author Rebuttal · Authors · 2024-08-05
>
> 1. Thanks for the positive feedback!
>
> 2. We agree that studying classification and other losses would be an important next step.
>
> 3. Thanks for pointing out the typos. We will correct them in the final version. We will also add the definitions of the symbols/concepts pointed out by the reviewer.
>
> 4. The “closeness” enters implicitly in the first term on the right-hand side of Eq (4), namely the excess population error of training on (infinitely many) surrogate samples. In this formula we assume that the population error of training on (infinitely many) original  samples is 0. Hence this term is non-negative and equal to 0 if and only if surrogate distribution coincides with the original one.

---

> > ### Comment · Reviewer_Fovm · 2024-08-09
> >
> > Thank you for the clarification. Good luck!

---

### Official Review · Reviewer_XfMG · 2024-07-16

**Soundness:** 2
**Presentation:** 1
**Contribution:** 2
**Rating:** 3
**Confidence:** 3

**Summary:**

This is a technical report on predicting training loss for a model trained on a weighted combination of real (in-distribution) and surrogate (out-of-distribution) data. The report proposes a parametric function for predicting how the training loss scales with the amount of real and surrogate data. The report first conducts theoretical analysis with a number of standard models on stylized cases, such as Gaussian sequence model, high-dimensional linear regression, etc., and derivated conditions for the proposed scaling relationship to hold. The report conducts empirical validation of the proposed scaling law on a range of classic tasks such as Sentiment analysis for the IMDB dataset, ResNet for CIFAR-10 classification, etc., and shows the training loss consistent with the proposed scaling function.

**Strengths:**

The paper adopts the format of a technical report rather than a typical research paper. The structure of the report is clear.

The problem is clearly introduced and formulated at the beginning of the paper.

Abundant theoretical analyses are provided and a range of models and cases are considered.

Experiments are diverse.

**Weaknesses:**

I'm not sure whether this technical report format is suitable for a NeurIPS paper. The lack of context or background is not a critical problem, though the lack of context may make it less accessible to broader audiences. What I found most concerning is the complete lack of comparison with existing works and a serious literature review.

The practice of data augmentation or/and using synthetic data from generative models/simulation, etc. is prevalent in the field and has a long history. This also seems not also the first paper to propose the idea of weighted training with real and surrogate data.

It is essential for any paper contributing to this topic to first
1. organize and list previous efforts on this topic;
2. introduce the current progress and research landscape; identify and justify there exists a valid research gap;
3. how this paper moves forward compared to previous efforts.

All of these are missing from this paper. Section 1.3 simply listed a number of potentially relevant references without commenting on each one or explaining how they relate to or are different from this paper. It is not actually informative.

---
The problem definition does not appear rigorous. Though problem formulation and notations come early on in the paper, many are not clearly introduced or defined.

Some vague arguments mismatch with the math-heavy writing style of the paper. For example, in the abstract, the authors emphasized

> "Integrating surrogate data can significantly reduce the test error on the original distribution. Surprisingly, this can happen even when the surrogate data is unrelated to the original ones. "

The word 'unrelated' makes it a very strong argument. Surprisingly, I did not find it clearly defined in the main paper.

Actually, I don't think learning with unrelated surrogate data will always reduce model loss. Imagine training the model with mislabeled data. It will only degrade the validation performance. My guess is what the authors are trying to say might be "training the model with not directly related data may also contribute positively due to regularization effect or knowledge transfer". But this is well-known to the community. So I am confused.

**Questions:**

See Weaknesses.

**Limitations:**

The work does not have a standalone discussion for its limitations.

---

> ### Author Rebuttal · Authors · 2024-08-05
>
> We agree: we will expand the related-work discussion.
>
> At the same time, part of this criticism is unfair. The distinction between “technical report” and “research paper” depends on the subcommunity. Mathematical papers tend to be more technical and less focused on positioning.
>
>
> Review: ‘The word 'unrelated' makes it a very strong argument. Surprisingly, I did not find it clearly defined in the main paper.’
>
> Response: The abstract and introduction try to convey our results in an intuitive way.
> This sentence means that the surrogate data have a distribution very different from the original one.
> In Section 2 we give a very simple mathematical example in which this claim is formalized and formally proven. Subsequent sections provide more sophisticated examples.
>
> So the claim is both rigorously stated two pages after it is introduced, rigorously proven, and, judging from the referee report, very surprising.
>
> Review: Actually, I don't think learning with unrelated surrogate data will always reduce model loss.
>
> Response: Actually Section 2 already gives a toy example in which adding surrogate data (and proper data-driven weighting) will yield a non-zero improvement, regardless of the distance between original and surrogate data!
>
> In practice, this improvement can be negligibly small, when the distribution of surrogate data is very different from the original one. However, we find that the improvement is often non-negligible in practice. Further, a data-driven weighing strategy yields significant improvements even in cases where the common practice of naively mixing real and surrogate data with equal weight hurts rather than helps.
>
> Review:  The work does not have a standalone discussion for its limitations.
>
> Response: Section 5 is a standalone discussion of the paper’s limitations.

---

> ### Comment · Area_Chair_wp1A · 2024-08-11
> **discussion**
>
> Dear Reviewer XfMG,
>
> Thank you very much for submitting your review report. The author(s) have posted responses to your review. Could you kindly provide comments on whether your concerns have been adequately addressed?
>
> AC

---

> > ### Comment · Reviewer_XfMG · 2024-08-12
> > **Thanks for the responses.**
> >
> > Thanks for the responses. I have seriously re-read the manuscript, attempted derivations myself, and read additional references.
> >
> > I admit that some of my previous reviews are inaccurate (on the evaluation loss and training with "unrelated data"). The quoted theories are indeed interesting.
> >
> > I like the idea of this paper. I feel the limitations are these stylized cases, generalized from Stein's phenomenon, still have gaps from practical use cases. For example, the paper will greatly improve its impact and attractiveness to readers beyond theory people if it can connect to commonly used practices such as regularization, data augmentation, etc.
> >
> > It is unclear whether the currently proposed scaling laws are practically useful. For example, a number of papers exist studying the problem of mixing data from different distributions to improve model performance, providing scaling functions with high accuracy and applying them to broad use cases including foundation models.
> > E.g.,
> > [Model performance scaling with multiple data sources]
> > [Performance scaling via optimal transport: Enabling data selection from partially revealed sources]
> > [Data mixing laws: Optimizing data mixtures by predicting language modeling performance]
> >
> > Do the authors think the proposed methods are practically competitive compared with these works?
> >
> > My current opinion is [weak reject] for the presentation issue outweighs the merit of the theoretical results.

---

> > > ### Author Response · Authors · 2024-08-12
> > >
> > > As witnessed by the references provided by the reviewers, adding surrogate data to training is a practice of great current interest in itself. This is different from data augmentation, which uses transformations of the original data, and from regularization, which does not use surrogate data. We will emphasize the relation to and distinction from  these lines of work in our revision.
> > >
> > > The referee also mentions three related works. All of these papers are very different and cannot be compared to ours.
> > >
> > > The most important difference is all of these papers use vanilla ERM instead of weighted ERM. As a consequence, if the surrogate data sample is sufficiently large, the effect of surrogate data will drown the original data, and the resulting model will perform poorly. In other words, if we compare our approach to theirs in a setting in which the surrogate dataset is significantly larger than the original one, our approach will outperform theirs by a multiplicative factor that can be arbitrarily large.
> > > (Of course, the naive solution to this problem is to add only a small fraction of surrogate data, but this is suboptimal.)
> > >
> > > Overcoming this fundamental problem is the very starting point of our work. Hence our work takes the next step beyond what is accomplished in these papers.
> > >
> > > We remark the following additional differences:
> > > 1. Hashimoto, T., 2021, July. Model performance scaling with multiple data sources. This paper derives a low-dimensional scaling result for data mixtures. However, the settings studied are such that the exponent is always equal to one.
> > > 2. Kang, F., Just, H.A., Sahu, A.K. and Jia, R., 2024. Performance scaling via optimal transport: Enabling data selection from partially revealed sources.
> > > This paper is entirely empirical. It assumes without mathematical justification that the error is an affine function of the Optimal Transport distance. Coefficients are fitted to this postulated relation without providing insights into how the scaling behavior depends on the mixture proportion. The only theorem assumes that the postulated scaling law holds exactly, which is obviously unrealistic.
> > > In conclusion, the real problem addressed in this paper is actually very different from ours. Given a certain postulated law, they optimize the proportion (which they do by gradient descent). Our work is about deriving a correct scaling law.
> > > 3. Ye, J., Liu, P., Sun, T., Zhou, Y., Zhan, J. and Qiu, X., 2024. ‘Data mixing laws: Optimizing data mixtures by predicting language modeling performance’.
> > > This paper was posted on arxiv on March 24. As such, it is concurrent or follow-up work. Also, our work implies that the “mixing law” they suggest does not hold in any of the  models that we can analyze mathematically.  (The law they propose in Eq (1) is exponential in a linear combination of the proportions, while we prove a different relationship.)

---

> > > > ### Comment · Reviewer_XfMG · 2024-08-13
> > > > **Thanks for the responses**
> > > >
> > > > Thanks for the responses.
> > > >
> > > > Adding surrogate data, by the idea of Stein's example, suggests improving the quality of estimation for a random variable by leveraging other sources of randomness, related or not. For that purpose, the effect of adding regularization or data augmentation seems also to align with the idea. Why we cannot model these effects with the same model for surrogate data?  Can we treat regularization as an independent source of randomness or consider augmented data as a related random variable?
> > > >
> > > > As for mixing data, why is "vanilla ERM" different from "weighted ERM for modeling purposes? For example, is training on [80% real data mixed with 20% surrogate data with vanilla ERM] the same as training on [50% real data mixed with 50% surrogate data with an 8:2 weighted ERM]?

---

> > > > > ### Author Response · Authors · 2024-08-13
> > > > >
> > > > > Regarding regularization.  The effect of regularization can be viewed as a special type of surrogate data, but there are very important differences that we also emphasized in our response to the area chair. These differences are subtle and require careful analysis, which is what we do in the paper.
> > > > > 1. The surrogate sample size m is finite. The effect of finite m in the scaling laws we derive (Eq (4), Eq (10), Eq (11), etc) is important, and would not be modeled by treating the surrogate data as regularization: the first error term in the square brackets in Eq. (4) would disappear. On the other hand,it is very important to capture this effect. In practice, the scaling law (4) can be used to decide how much surrogate data to acquire, and this is only possible if we model the effect of finite m.
> > > > > 2. The regularization effect produced by surrogate data depends of course on the distribution of the surrogate data itself. In particular, while a regularization typically shrinks the parameters of the model towards 0, surrogate data shrink it towards a value theta** that is determined by the data distribution. Also, the form of this shrinkage effect depends in subtle ways on the data distribution.
> > > > > 3. Finally, the value of this ‘center’ theta** is of course very important, and can significantly change the behavior of the weighted ERM scheme. In practical settings, we cannot construct a regularizer which contains a reasonable guess of theta** with a meaningful guess without surrogate data.
> > > > > Finally, augmented data can be regarded as consisting of samples dependent on the original samples. This of course can be modeled mathematically. The difficulty is in analyzing and obtaining non-trivial results in such a model. We do not do it in this paper, and in fact we are not aware of any works doing it in a general rigorous way.
> > > > >
> > > > >
> > > > > Regarding weighting. This is a great question. In synthesis, the two schemes proposed by the referee are equivalent only for what concerns the expected (or population) risk. However, analysis at the level of expected risk is only accurate when the number of original and surrogate examples are very large or infinite. But in this extreme case adding surrogate data is already pointless. In more practical scenarios, where the number of original examples is small, we can achieve a lot more with weighted ERM than vanilla ERM.
> > > > > This can be seen already in Eq. (1).
> > > > > A practical illustration.
> > > > > Consider the case where we have 400 original examples and 1,000,000 surrogate examples.
> > > > > Suppose we want to give (on average) a weight of 80% to original and 20% to surrogate.
> > > > > With weighted ERM we can use all 1,000,000 surrogate examples and assign the surrogate dataset a weight of 0.2 to it and weight 0.8 to the original dataset.
> > > > > In the “vanilla ERM” alternative, we must limit ourselves to 100 surrogate examples. This is a very small number and the statistical noise introduced by the surrogate examples may be dominating the signal they provide.
> > > > > Overall,  adding surrogate examples in a vanilla ERM fashion may hurt the model performance. On the other hand adding a large number of surrogate examples with a weight 0.2 will help the model if surrogate data and the original data distributions are not too far in expectation.

---

### Official Review · Reviewer_zfeC · 2024-07-17

**Soundness:** 4
**Presentation:** 4
**Contribution:** 3
**Rating:** 8
**Confidence:** 4

**Summary:**

In the context of linear regression, this paper proposes to complement training data with surrogate data (generated / synthesized, or from an unrelated task / domain). For an optimal combination of real and surrogate data, new scaling laws are derived which show that that surrogate data can can help reduce the text error.

**Strengths:**

The paper is very well-written. The context and motivations are clearly presented. The paper makes nontrivial contribution to understanding commonplace practice in training large models in modern AI era: namely, the use of synthetic  or surrogate data.

**Weaknesses:**

NaN

**Questions:**

- In practice, how is the optimal value of the weighting paramater $\alpha_*$ to be set ? Is there a consistent estimator for it which only depends on the input training data ? Without such a prescription, the proposed scheme might fall short of practical usefulness.

- My understanding is that the proof of the main result uses the Gordon comparison theorem. If a nontrivial covariance matrix for the features $x$ is introduced, major modifications might be required, or an entirely different approach (e.g via RMT) might be required. I'd be glad if the the authors can comment on this.

- I wonder what this work could tell us about the "model collapse" phenomenon [1,2,3], where a model trained on synthetic data eventually breaks down (becomes biased towards a trivial / useless estimator).

[1] Shumailov et al. "The Curse of Recursion: Training on Generated Data Makes Models Forget" (2023)
[2]  Alemohammad et al. "Self-Consuming Generative Models Go MAD" (2023)
[3] Dohmatob et al. "A Tale of Tails: Model Collapse as a Change of Scaling Laws" (ICML 2024)

**Limitations:**

The setup considered is extremely: linear regression with Gaussian data with no covariance structure (i.e isotropic covariates)

---

> ### Author Rebuttal · Authors · 2024-08-05
>
> In our paper we provide two recommendations for selecting $\alpha_*$
> 1. Compute the error on a validation split from the original data for various values of $\alpha$, and optimize among those. (This can be improved using cross validation)
> 2. Use the scaling law expression, with free parameter fitted to empirical error at $\alpha=0$ and $\alpha=1$.
> The first technique can be proved to be consistent under standard assumptions. The second one is computationally faster. Of course, we can think of intermediate strategies as well.
> We will emphasize these recommendations.
>
> Indeed, the result for high-dimensional ridge regression uses Gordon inequality. This approach can be generalized to cover cases with non-identity covariance at the price of becoming technically heavier. (See, e.g. Celentano, M., Montanari, A. and Wei, Y., 2023. The lasso with general gaussian designs with applications to hypothesis testing. The Annals of Statistics, 51(5), pp.2194-2220.)
>
> The connection to model collapse is indeed worth pursuing. One interesting remark is that there exist weighting schemes that will prevent model collapse.

---

> > ### Comment · Reviewer_zfeC · 2024-08-13
> > **Discussion**
> >
> > My concerns have mostly been addressed. Thanks. I'll keep my current score.

---

### Comment · Area_Chair_wp1A · 2024-08-11
**request for clarification**

Dear Author(s),

I am curious about the following points. Could you please provide some comments? Thank you very much.

1. Would the theoretical analysis still hold if we were to replace the surrogate data loss term with an l2 penalty centered at θ** (whether carefully designed or arbitrarily selected)? If not, could you please elaborate on the challenges this would pose for the proof? If it would, considering that this substitution could significantly simplify the problem, could you provide insights on why the use of surrogate data remains essential?

2. I notice that Theorem 3 characterizes the concentration of the risk to R_test(α). By the way, could 'Λ(α)' possibly be a typo? Additionally, I didn't see a definition for R(θ). Would the concentration weaken or perhaps improve if α were to fall outside this interval [ε0, 1−ε0]?

---

> ### Author Response · Authors · 2024-08-12
>
> Thanks for your questions:
>
> a. The analysis of this type of regularization (l2 penalty centered at theta**) could be carried out in the same setting as the one we do. Indeed, it can sometimes be viewed as a special case of the analysis we do. However, there are three important differences:
>
> 1. We consider the surrogate sample size m to be finite. The effect of finite m in the scaling laws we derive (Eq (4), Eq (10), Eq (11), etc) is important, and would not be modeled by the setting proposed by the AE: the first error term in the square brackets in Eq. (4) would disappear. On the other hand, we believe it is very important to capture this effect. In practice, the scaling law (4) can be used to decide how much surrogate data to acquire, and this is only possible if we model the effect of finite m.
>
> 2. The regularization effect produced by surrogate data depends of course on the distribution of the surrogate data itself. While thinking of it as an l2 penalty can provide useful intuition, it is not generally correct even in very simple models. For instance, in the Gaussian sequence model of Section 3.1, if the model is heteroscedastic, then the regularization will be a weighted l2, depending on the noise distribution. Similarly for our other theoretical results. These examples can be treated as corollaries of our results, and we plan to add them to our revision.
>
> 3. Finally, the choice of theta** is of course very important, and can significantly change the behavior of the weighted ERM scheme. While in very simple models and/or theoretical work, we can pretend that we have a good guess, in large high-dimensional models, there is typically no practical way to come up with a meaningful guess without surrogate data.
>
> b. Thank you very much for these points. Indeed the $\Lambda(\alpha)$ is a typo: this should be $R_{test}(\alpha)$. The uniformity interval $[\epsilon_0,1-\epsilon_0]$ can be replaced by $[0,1]$ when $\delta,\delta_s>1$. This is also confirmed visually in Figure 4. If $\delta$ or $\delta_s<1$, then a `phase transition’ can occur at $\alpha=0$ or $\alpha=1$, which is why we focus on the interval $[\epsilon_0,1-\epsilon_0]$. While this phase transition phenomenon is quite interesting, we do not have space to elaborate on it in this submission.
> We know however that when this phase transition occurs, the test error is typically higher at $\alpha=0$ or $\alpha=1$. Therefore, from a practical point of view, there is no loss of generality in focusing on $[\epsilon_0,1-\epsilon_0]$.
> Finally, the test error $R_{test}(\theta)$ is defined in line 54, and for the specific case at hand in line 199.

---

> > ### Comment · Area_Chair_wp1A · 2024-08-14
> > **thank you very much!**
> >
> > Thank you very much for the clarification.

---

### Decision · Program_Chairs · 2024-09-25

**Decision:**

Accept (poster)

**Comment:**

This paper addresses the challenge of augmenting limited real data with surrogate data, particularly in scenarios where high-quality data is scarce or expensive to obtain. The authors propose a weighted empirical risk minimization (ERM) approach to integrate surrogate data into training processes. The primary contribution is the introduction of a new scaling law that optimizes the weighting scheme for combining real and surrogate data, validated both theoretically and empirically. The key finding is that integrating surrogate data, even when unrelated to the original data, can significantly reduce test error, a result linked to Stein's paradox.
The paper is technically sound, offering both theoretical analysis and empirical validation across various domains.

Reviewer XfMG gave the lowest score of 3, primarily citing concerns about the literature review, notation clarity, practical utility, and format. After reviewing the discussion chain, I believe these comments are not entirely fair.

Reviewer 3jgb gave the second-lowest score of 4, with concerns centered on the theory-simulation gap, the need for a more detailed discussion of regularization, and the assumptions underlying the theorems. I believe these issues can be addressed with minor revisions and should not prevent the paper from being published.

The other scores are 6, 8, and 9, very high indeed.